# Training Nonlinear Transformers for Chain-of-Thought Inference: A Theoretical Generalization Analysis

**Hongkang Li[1], Songtao Lu[2,\*], Pin-Yu Chen[3], Xiaodong Cui[3], Meng Wang[1,†]**
[1]Rensselaer Polytechnic Institute, [2]The Chinese University of Hong Kong, [3]IBM Research

## Abstract

Chain-of-Thought (CoT) is an efficient prompting method that enables the reasoning ability of large language models by augmenting the query using multiple examples with multiple intermediate steps. Despite the empirical success, the theoretical understanding of how to train a Transformer to achieve the CoT ability remains less explored. This is primarily due to the technical challenges involved in analyzing the nonconvex optimization on nonlinear attention models. To the best of our knowledge, this work provides the first theoretical study of training Transformers with nonlinear attention to obtain the CoT generalization capability so that the resulting model can inference on unseen tasks when the input is augmented by examples of the new task. We first quantify the required training samples and iterations to train a Transformer model towards CoT ability. We then prove the success of its CoT generalization on unseen tasks with distribution-shifted testing data. Moreover, we theoretically characterize the conditions for an accurate reasoning output by CoT even when the provided reasoning examples contain noises and are not always accurate. In contrast, in-context learning (ICL), which can be viewed as one-step CoT without intermediate steps, may fail to provide an accurate output when CoT does. These theoretical findings are justified through experiments.

## 1 Introduction

Transformer-based large-scale foundation models, such as GPT-3 (Brown et al., 2020), GPT-4 (OpenAI, 2023), LLaMa (Touvron et al., 2023a;b), and Sora (Liu et al., 2024), have demonstrated remarkable success across various tasks, including natural language processing (Brown et al., 2020; Touvron et al., 2023b), multimodal learning (OpenAI, 2023; Radford et al., 2021), and image/video generation (OpenAI, 2023; Liu et al., 2024). What is more surprising is that large language models (LLMs) demonstrate reasoning ability through the so-called "Chain-of-Thought" (CoT) method (Wei et al., 2022). The objective is to let a pre-trained LLM generate $K$ steps of reasoning given input query $\boldsymbol{x}_{query}$ without any fine-tuning. To achieve that, the input $\boldsymbol{x}_{query}$ is augumented with $l$ examples $\{\boldsymbol{x}_i, \{\boldsymbol{y}_{i,j}\}_{j=1}^K\}_{i=1}^l$ of a certain $K$-step reasoning task, where each $\boldsymbol{x}_i$ is the input with $\boldsymbol{y}_{i,j}$ as the $j$-th reasoning step, and $\boldsymbol{y}_{i,K}$ is the final output. A pre-trained model then takes the resulting augmented input, referred to as a *prompt*, and outputs the corresponding reasoning steps $\{\boldsymbol{z}_j\}_{j=1}^K$ for $\boldsymbol{x}_{query}$, or simply outputs $\boldsymbol{z}_K$. CoT can be viewed as an extended and more intelligent method than the previous in-context learning (ICL) method, where only input-label pairs $\{\boldsymbol{x}_i, \boldsymbol{y}_{i,K}\}_{i=1}^l$ are augumented in the prompt to predict $\boldsymbol{z}_K$ with the pre-trained model.

Inspired by the outstanding empirical performance of CoT in arithmetic reasoning (Wang et al., 2023; Zhang et al., 2023b; Wang & Zhou, 2024), symbolic reasoning (Zhang et al., 2023b; Zhou et al., 2023), and commonsense reasoning (Wang et al., 2023; Wang & Zhou, 2024), there have been some recent works (Li et al., 2023c; Feng et al., 2023; Li et al., 2024d; Yang et al., 2024; Wen et al., 2024) on the theoretical understanding of CoT. These works investigate CoT from the perspective of expressive power, i.e., they construct the Transformer architecture that is proven to have the CoT ability. They also demonstrate empirically that supervised training on pairs of CoT

---

*This work was done when Prof. Songtao Lu was at IBM Research.
†Corresponding author. Email: wangm7@rpi.edu.

prompts and corresponding outputs can lead to models with CoT ability. However, none of these results theoretically address the question of why a Transformer can obtain generalization-guaranteed CoT ability by training from data with gradient-based methods. Meanwhile, another line of research (Zhang et al., 2023a; Huang et al., 2023; Wu et al., 2023; Li et al., 2024a) aims to unveil the reasons behind the ICL ability of Transformers through characterizing the training dynamics of a Transformer in the supervised setting. These analyses are specifically applicable to ICL. Therefore, a theoretical question still remains less explored, i.e.,

*Why can a Transformer be trained to generalize on multi-step reasoning tasks via CoT?*

## 1.1 Major Contributions

Following Li et al. (2023c); Feng et al. (2023); Li et al. (2024d); Yang et al. (2024); Wen et al. (2024), we train the model in a supervised setting using prompt and label pairs. This paper provides the first theoretical analysis of the training dynamics of nonlinear Transformers to achieve CoT ability. We prove that the learned model has guaranteed CoT ability for new tasks with distribution shifts from the training tasks, even when there exist noisy and erroneous context examples in the prompt. We theoretically characterize the required number of training samples and iterations needed to train a desirable model and the number of context examples required for successful CoT reasoning with a generalization guarantee. Moreover, we provide a theoretical explanation for why CoT outperforms ICL in some cases. Our main technical contributions are as follows:

1. **A quantitative analysis of how the training can enable the CoT ability**: We theoretically analyze the training dynamics on a one-layer single-head attention-only Transformer and quantify the required number of context examples in each training sample, the total number of training samples, and the number of training iterations needed to acquire CoT ability. We illustrate that the CoT ability results from the property that the attention values of the learned model are concentrated on testing context examples with the same input patterns as the testing query during each reasoning step.

2. **A quantitative analysis of how context examples affect CoT performance**: We characterize the required number of context examples in the testing prompt for successful CoT reasoning when noise and error exist in contexts. Our quantitative bounds are consistent with the intuition that more accurate context examples and more similar examples to the query improve CoT accuracy.

3. **A theoretical characterization of why CoT outperforms ICL**: We provide a quantitative analysis of the requirements for successful ICL reasoning with our studied trained model. We show that successful ICL requires an additional condition that the prompt has a dominant number of correct input-label examples, while the success of CoT does not depend on this condition. This can be viewed as one of the possible reasons why CoT outperforms ICL.

## 1.2 Related works

**Expressive power of CoT** Li et al. (2023c) proves the existence of a Transformer that can learn a multi-layer perceptron (MLP). They interpret CoT as first filtering important tokens and then making predictions by ICL. They also establish the required number of context examples for a desired prediction with the constructed Transformer. Feng et al. (2023); Li et al. (2024d); Merrill & Sabharwal (2024) show that Transformers with CoT are more expressive than Transformers without CoT. Yang et al. (2024); Wen et al. (2024) show the superiority of standard Transformers in some reasoning tasks compared with recurrent neural networks and linear Transformers.

**Theoretical analysis of ICL** As a simplified one-step version of CoT, ICL has gained much attention from the theoretical community. Garg et al. (2022); Akyürek et al. (2023); Bai et al. (2023); Guo et al. (2023) demonstrate that Transformers are expressive to conduct many machine learning algorithms in context. Akyürek et al. (2023); Von Oswald et al. (2023); Ahn et al. (2023); Cheng et al. (2023); Ding et al. (2024) especially show the existence of Transformers to implement gradient descent and its variants with different input prompts. Zhang et al. (2023a); Huang et al. (2023); Wu et al. (2023); Li et al. (2024a) explore the training dynamics and generalization of ICL on single-attention Transformers. Cui et al. (2024); Chen et al. (2024) provably show the superiority of multi-head attention over single-head attention to achieve ICL ability.

**Training and Generalization of Transformers** There have been several recent works about the optimization and generalization analysis of Transformers. Jelassi et al. (2022); Li et al. (2023d); Oymak et al. (2023); Li et al. (2023a;b; 2024b); Luo (2023); Huang et al. (2024); Zhang et al. (2024) study

the generalization of one-layer Transformers by assuming spatial association, semantic/contextual structure, or the majority voting of tokens in the data. Oymak et al. (2023); Tarzanagh et al. (2023b;a); Tian et al. (2023a;b); Li et al. (2024c); Ildiz et al. (2024); Nichani et al. (2024); Makkuva et al. (2024b) investigate the training dynamics or loss landscape of Transformers for the next token prediction by assuming infinitely long input sequences, causal structure/Markov Chain of data, or a proper prediction head. Deora et al. (2023); Chen & Li (2024) analyze the optimization and generalization of multi-head attention networks.

## 2 PROBLEM FORMULATION

We study the problem of learning and generalization of $K$-steps reasoning tasks. Each task $f = f_K \circ \cdots f_2 \circ f_1$ is a composition of functions $\{f_i\}_{i=1}^K$ and outputs labels $z_1, z_2, \cdots, z_K$ for the input $x_{query}$. During the $k$-th reasoning step, $k \in [K]$, the label is $z_k = f_k(z_{k-1})$, where $z_0 := x_{query}$.

### 2.1 TRAINING TO ACQUIRE THE CHAIN-OF-THOUGHT ABILITY

Following theoretical analysis (Feng et al., 2023; Li et al., 2024d; Wen et al., 2024) and empirical works like process supervision (Lightman et al., 2024), we first investigate the training on a Transformer model to obtain the CoT ability in evaluating new data and tasks. It is a supervised learning setting on pairs of prompts and labels. Different from the testing prompt that includes examples and only $x_{query}$, the training prompt includes multiple $K$-steps reasoning examples and a $(k-1)$-step reasoning of $x_{query}$ for any $k$ in $[K]$, and the label for this prompt is $z_k$. Specifically,

**Training Prompt and Label for CoT.** For every prompt and output pair from a task $f = f_K \circ \cdots f_2 \circ f_1$, we construct a prompt $P$ that include the query input $z_{k-1}$ by prepending $l_{tr}$ reasoning examples and the first $k-1$ steps of the reasoning query. The prompt $P$ of the query input $z_{k-1}$ is formulated as:
$$P = (E_1, E_2, \cdots, E_{l_{tr}}, Q_k) \in \mathbb{R}^{2d_{\mathcal{X}} \times (l_{tr}K+k)},$$
$$\text{where } E_i = \begin{pmatrix} x_i & y_{i,1} & \cdots & y_{i,K-1} \\ y_{i,1} & y_{i,2} & \cdots & y_{i,K} \end{pmatrix}, \; Q_k = \begin{pmatrix} z_0 & z_1 & \cdots & z_{k-2} & z_{k-1} \\ z_1 & z_2 & \cdots & z_{k-1} & 0 \end{pmatrix}, i \in [l_{tr}], \quad (1)$$

where $E_i$ is the $i$-th context example, and $Q_k$ is the first $k$ steps of the reasoning query for any $k$ in $[K]$. We have $y_{i,k} = f_k(y_{i,k-1})$ and $z_k = f_k(z_{k-1})$ for $i \in [l_{tr}]$, $k \in [K]$ with a notation $y_{i,0} := x_i$. Let $p_s$ and $p_{query}$ be the $s$-th column and the last column of $P$, respectively, for $s \in [l_{tr}K + k - 1]$. $x_i, y_{i,k}, z_j \in \mathbb{R}^{d_{\mathcal{X}}}$ for $i \in [l_{tr}]$ and $j, k \in [K]$. We respectively call $x_i$ and $y_{i,k}$ *context* inputs and outputs of the $k$-th step of the $i$th context example. For simplicity of presentation, we denote $z$ as the label of $P$, which is indeed $z_k$ for (1). All the notations are summarized in Table 3 in Appendix.

The **learning model** is a single-head, one-layer attention-only Transformer. We consider positional encoding $\{c_k\}_{k=1}^K \in \mathbb{R}^{2d_{\mathcal{X}}}$. Following theoretical works (Jelassi et al., 2022; Huang et al., 2024; Ildiz et al., 2024), we add the positional encoding to each $p_i$ by $\tilde{p}_i = p_i + c_{(i \mod K)}$, $i \in [K(l_{tr}+1)]$. $\tilde{p}_{query}$ is also defined by adding the corresponding $c_k$ to $p_{query}$. Mathematically, given a prompt $P$ defined in (1) with $\text{len}(P)$ (which is at most $K(l_{tr}+1)$) denoting the number of columns, it can be written as
$$F(\Psi; P) = \sum_{i=1}^{\text{len}(P)-1} W_V \tilde{p}_i \cdot \text{softmax}((W_K \tilde{p}_i)^\top W_Q \tilde{p}_{query}), \quad (2)$$

where $W_Q, W_K \in \mathbb{R}^{m \times (2d_{\mathcal{X}})}$, $W_V \in \mathbb{R}^{d_{\mathcal{X}} \times (2d_{\mathcal{X}})}$ are the embedding matrices for queries, keys, and values, respectively. $\Psi := \{W_Q, W_K, W_V\}$ is the set of all model weights[1]. Typically, $m > 2d_{\mathcal{X}}$. Here, $\text{softmax}((W_K \tilde{p}_i)^\top W_Q \tilde{p}_{query}) = e^{(W_K \tilde{p}_i)^\top W_Q \tilde{p}_{query}} / \sum_{j=1}^{\text{len}(P)-1} e^{(W_K \tilde{p}_j)^\top W_Q \tilde{p}_{query}}$.

The **training problem** to enhance the reasoning capability solves the empirical risk minimization,
$$\min_{\Psi} R_N(\Psi) := \frac{1}{N} \sum_{n=1}^N \ell(\Psi; P^n, z^n), \quad (3)$$

using $N$ prompt and label pairs $\{P^n, z^n\}_{n=1}^N$. For the $n$-th sample, $x_{query}^n$ and the context input $x_i^n$ are all sampled from an unknown distribution $\mathcal{D}$, the training task $f^n$ is sampled from $\mathcal{T}$, $k$ is randomly selected from 1 to $K$, and $P^n$ is constructed following (1). The loss function is squared loss, i.e., $\ell(\Psi; P^n, z^n) = 1/2 \cdot \|z^n - F(\Psi; P^n)\|^2$, where $F(\Psi; P^n)$ is defined in (2).

[1]We focus on a one-layer single-head Transformer motivated by recent advancements and current state in Transformer and CoT analysis. Please see Appendix B.1 for discussion.

## 2.2 Training Algorithm

For simplicity of analysis, we let $\boldsymbol{W} = \boldsymbol{W}_K^\top \boldsymbol{W}_Q$ and $\boldsymbol{W}_V = (\boldsymbol{0}_{d_\mathcal{X} \times d_\mathcal{X}} \ \boldsymbol{I}_{d_\mathcal{X}}) \in \mathbb{R}^{d_\mathcal{X} \times (2d_\mathcal{X})}$ as (Jelassi et al., 2022; Huang et al., 2023; Zhang et al., 2023a; Huang et al., 2024). Let $\{\boldsymbol{c}_k\}_{k=1}^K$ be a set of orthonormal vectors. The model is trained using stochastic gradient descent (SGD) with step size $\eta$ with batch size $B$, summarized in Algorithm 1 in Appendix C. Each entry of $\boldsymbol{W}^{(0)}$ is generated from $\mathcal{N}(0, \xi^2)$ for a tiny $\xi > 0$. $\boldsymbol{W}_V$ is fixed during the training. The fraction of prompts with $\boldsymbol{z}_{k-1}$ as the query input is $1/K$ by uniform sampling for any $k \in [K]$ in each batch.

## 2.3 Chain-of-Thought Inference

We then consider another $K$-steps reasoning task $f \in \mathcal{T}'$, whose target is to predict labels $\{\boldsymbol{z}_k\}_{k=1}^K$ given the input query $\boldsymbol{x}_{query}$. $\mathcal{T}'$ is the set of testing tasks, and $\mathcal{T}' \neq \mathcal{T}$.

**Testing Prompt for CoT.** The testing prompt $\boldsymbol{P}$ is composed of $l_{ts}$ ($\leq l_{tr}$) context examples of $K$ steps plus a query, which is constructed as

$$\boldsymbol{P} = (\boldsymbol{E}_1, \boldsymbol{E}_2, \cdots, \boldsymbol{E}_{l_{ts}}, \boldsymbol{p}_{query}) \in \mathbb{R}^{(2d_\mathcal{X}) \times (l_{ts}K+1)}, \boldsymbol{p}_{query} = (\boldsymbol{x}_{query}^\top, \boldsymbol{0}^\top)^\top, \quad (4)$$

where $\boldsymbol{E}_i$ follows the form in (1) for $i \in [l_{ts}]$.

We follow the CoT-I/O scheme formulated in (Li et al., 2023c; Feng et al., 2023; Li et al., 2024d; Yang et al., 2024; Park et al., 2024) as the inference method. Specifically, for a $K$-step CoT with $l_{ts}$ examples on a certain $f \in \mathcal{T}'$, given the testing prompt $\boldsymbol{P}$ defined in (4), let $\boldsymbol{P}_1 = \boldsymbol{P}$ and $\boldsymbol{P}_0$ be the first $K \cdot l_{ts}$ columns of $\boldsymbol{P}$. When we use CoT prompting for prediction in the $k$-th step, we first generate the output $\boldsymbol{v}_k$, $k \in [K]$ via greedy decoding by feeding the $k$-th step prompt $\boldsymbol{P}_k$ to the trained model $\Psi$ obtained from (3). The greedy decoding scheme means outputting the most probable token from the discrete set $\mathcal{Y}$ of all possible outputs, as stated in (5).

$$\boldsymbol{v}_k = \arg\min_{\boldsymbol{u} \in \mathcal{Y}} \frac{1}{2} \|F(\Psi; \boldsymbol{P}_k) - \boldsymbol{u}\|^2, \text{ (greedy decoding)} \quad (5)$$

Then, we use the output $\boldsymbol{v}_k$ to update $\boldsymbol{P}_k$ and use $\boldsymbol{v}_k$ as the query input to form the input prompt $\boldsymbol{P}_{k+1}$ for the next step, which is computed as

$$\boldsymbol{P}_k = (\boldsymbol{P}_{k-1} \ \boldsymbol{q}_k) \in \mathbb{R}^{(2d_\mathcal{X}) \times (Kl_{ts}+k)}, \ \boldsymbol{P}_{k+1} = (\boldsymbol{P}_k \ \boldsymbol{q}_{k+1}) \in \mathbb{R}^{(2d_\mathcal{X}) \times (Kl_{ts}+k+1)},$$
$$\text{where } \boldsymbol{q}_k = (\boldsymbol{v}_{k-1}^\top \ \boldsymbol{v}_k^\top)^\top, \ \boldsymbol{q}_{k+1} = (\boldsymbol{v}_k^\top \ \boldsymbol{0}^\top)^\top, \quad (6)$$

where $\boldsymbol{q}_k$ is the $k$-th step reasoning column for the query. The model finally outputs $\boldsymbol{v}_1, \cdots, \boldsymbol{v}_K$ as CoT result for query $\boldsymbol{x}_{query}$ by (5). The CoT process is summarized in Algorithm 2 of Appendix C.

When $K \geq 2$, following (Li et al., 2023c; Feng et al., 2023; Li et al., 2024d; Yang et al., 2024), the **CoT generalization error** given the testing query $\boldsymbol{x}_{query}$, the testing data distribution $\mathcal{D}'$, and the labels $\{\boldsymbol{z}_k\}_{k=1}^K$ on a $K$-steps testing task $f \in \mathcal{T}'$ is defined as

$$\bar{R}_{CoT, \boldsymbol{x}_{query} \sim \mathcal{D}', f \in \mathcal{T}'}^f (\Psi) = \mathbb{E}_{\boldsymbol{x}_{query} \sim \mathcal{D}'} \left[ \frac{1}{K} \sum_{k=1}^K \mathbb{1}[\boldsymbol{z}_k \neq \boldsymbol{v}_k] \right], \quad (7)$$

which measures the average error between the output and the label of each reasoning step. A zero CoT generalization error indicates correct generations in all $K$ steps.

## 2.4 In-Context Learning Inference

The ICL inference on a $K$-steps reasoning task $f \in \mathcal{T}'$ only predicts the final-step label by perpending examples of input and label pairs before the query. ICL can be viewed as a one-step CoT without intermediate steps. Here, we evaluate the ICL performance of the trained model.

**Testing Prompt for ICL.** Mathematically, ICL is implemented by constructing a prompt $\boldsymbol{P}$ as below,

$$\boldsymbol{P} = (\boldsymbol{E}_1, \cdots, \boldsymbol{E}_{l_{ts}}, \boldsymbol{p}_{query}), \text{where } \boldsymbol{p}_{query} = \begin{pmatrix} \boldsymbol{x}_{query} \\ \boldsymbol{0} \end{pmatrix}, \boldsymbol{E}_i = \begin{pmatrix} \boldsymbol{x}_i & \boldsymbol{0} & \cdots & \boldsymbol{0} \\ \boldsymbol{y}_{i,K} & \boldsymbol{0} & \cdots & \boldsymbol{0} \end{pmatrix} \quad (8)$$

$\boldsymbol{P} \in \mathbb{R}^{(2d_\mathcal{X}) \times (l_{ts}K+1)}$, $\boldsymbol{E}_i \in \mathbb{R}^{(2d_\mathcal{X}) \times K}$ for $i \in [l_{ts}]$. Note that in the ICL setting, $\boldsymbol{E}_i$ only has input $\boldsymbol{x}_i$ and the $K$-step output $\boldsymbol{y}_{i,K}$ but does not include any intermediate labels. We pad zeros in $\boldsymbol{E}_i$ so that its dimension is the same as $\boldsymbol{E}_i$ in (1) for the inference with the same model as for CoT. The ICL output is $\boldsymbol{v} = \arg\min_{\boldsymbol{u} \in \mathcal{Y}} \frac{1}{2} \|F(\Psi; \boldsymbol{P}) - \boldsymbol{u}\|^2$, following (5). The **ICL generalization error** is

$$\bar{R}_{ICL, \boldsymbol{x}_{query} \sim \mathcal{D}', f \in \mathcal{T}'}^f (\Psi) = \mathbb{E}_{\boldsymbol{x}_{query} \sim \mathcal{D}'} [\mathbb{1}[\boldsymbol{z}_K \neq \boldsymbol{v}]], \quad (9)$$

which measures the error between the one-step reasoning output and the final step label.

## 3   THEORETICAL RESULTS

We first summarize the main theoretical insights in Section 3.1. Then, we introduce the formulation of data and tasks in Section 3.2. Sections 3.3, 3.4, and 3.5, respectively characterize the training analysis of the Transformer and generalization using CoT and ICL with the trained model.

### 3.1   MAIN THEORETICAL INSIGHTS

We consider the setup that the model is trained using samples generated from tasks in $\mathcal{T}$ that operate on $M$ orthonormal training-relevant (TRR) patterns, while both CoT and ICL are evaluated on tasks in $\mathcal{T}'$ that operate on $M'$ orthonormal testing-relevant (TSR) patterns. We consider the general setup that the context examples in the prompt for CoT and ICL testing are both noisy, i.e., TSR patterns with additive noise, and partially inaccurate, i.e., the reasoning in some examples contains incorrect steps. Our main insights are as follows.

**P1. Training Dynamics of Nonlinear Transformer towards CoT**. We theoretically analyze the training dynamics on a one-layer single-head attention-only Transformer to acquire the CoT generalization ability and characterize the required number of training samples and iterations. Theorem 1 shows that to learn a model with guaranteed CoT ability, the required number of context examples in each training sample and the total number of training samples/iterations are linear in $\alpha^{-1}$ and $\alpha^{-2}$, respectively, where $\alpha$ is the fraction of context examples with inputs that share the same TRR patterns as the query. This is consistent with the intuition that the CoT performance is enhanced if more context examples are similar to the query. Moreover, the attention values of the learned model are proved to be concentrated on testing context examples that share similar input TSR patterns as the testing query during each of the reasoning steps (Proposition 1), which is an important property that leads to the success of the CoT generalization.

**P2. Guaranteed CoT Generalization**. To achieve zero CoT error on tasks in $\mathcal{T}'$ and data based on TSR patterns that contain a non-trivial component in the span of TRR patterns with the learned model, Theorem 2 shows that the required number of context examples, where noise and errors are present, for task $f$ in the testing prompt is proportional to $(\alpha' \tau^f \rho^f)^{-2}$. Here, $\alpha'$ is the fraction of context examples with inputs that share the same TSR patterns as the query. $\tau^f$ in $(0,1)$ measures the fraction of accurate context examples, and a larger constant $\rho^f$ in $(0,1)$ reflects a higher reasoning accuracy in each step of the examples. This result formally characterizes the intuition that more accurate context examples and more similar examples to the query improve the CoT accuracy.

**P3. CoT outperforms ICL**. In Theorem 3, We theoretically show that the required number of testing context examples for ICL to be successful has a similar form to that for CoT in Theorem 2, but with an additional requirement (Condition 1) that the fraction of correct input-label examples in the testing prompt must be dominant. Because not all testing cases satisfy this requirement, our result provides one explanation for why CoT sometimes outperforms ICL.

### 3.2   THE FORMULATION OF DATA AND TASKS

**Training data and tasks**: Consider $M$ training-relevant (TRR) patterns $\boldsymbol{\mu}_1, \boldsymbol{\mu}_2, \cdots, \boldsymbol{\mu}_M$, which form an orthonormal set $\mathcal{M} = \{\boldsymbol{\mu}_i\}_{i=1}^M$. $M = \Theta(d), M \leq d$. $(\boldsymbol{\mu}_i^\top, 0_{d_{\mathcal{X}}}^\top)^\top \perp \boldsymbol{c}_k$ for $i \in [M'], k \in [K]$.

Every training prompt $\boldsymbol{P}$ in (1) contains the query and training examples from the same training task $f$ in the set of training tasks $\mathcal{T}$. Specifically, each training task $f$ is a composition of $K$ functions $f = f_K \circ \cdots \circ f_2 \circ f_1$ where each function $f_k$ belongs to a function set $\mathcal{F}$. The $k$-th step label of the query is $\boldsymbol{z}_k = f_k(\boldsymbol{z}_{k-1})$ given the $k$-th step input $\boldsymbol{z}_{k-1}$ with $\boldsymbol{z}_k \in \mathcal{M}, k \in [K]$. Moreover, the $k$-th step label of the $i$-th ($i \in [l_{tr}]$) context example is $\boldsymbol{y}_{i,k} = f_k(\boldsymbol{y}_{i,k-1})$ given the $k - 1$th step input $\boldsymbol{y}_{i,k-1}, k \in [K]$ with $\boldsymbol{x}_i, \boldsymbol{y}_{i,k} \in \mathcal{M}$, where $\boldsymbol{y}_{i,0} := \boldsymbol{x}_i$.[2] We assume that $f_k(\boldsymbol{x}) \neq f_{k'}(\boldsymbol{x}')$ if and only if either $\boldsymbol{x} \neq \boldsymbol{x}'$ or $f_k \neq f_{k'}$.

**Training prompt**: Consider a training prompt $\boldsymbol{P}$ on task $f \in \mathcal{T}$ defined in (1) with the query input $\boldsymbol{z}_{k-1}, k \in [K]$. Let $\alpha \in (0, 1 - c]$ for some constant $c > 0$[3] denote the fraction of context examples with input sharing the same TRR pattern as the query input.

---

[2] The formulation of $f$ is motivated by recent theoretical works on model training or ICL with Transformers. Please see Appendix B.2 for details.

[3] This is to prevent the trivial case that the model only learns the positional encoding but not the TRR patterns when $\alpha$ becomes arbitrarily close to 1.

**Testing task and query**: Consider $M'$ testing-relevant (TSR) patterns $\boldsymbol{\mu}'_1, \boldsymbol{\mu}'_2, \cdots, \boldsymbol{\mu}'_M$, which form an orthonormal set $\mathcal{M}' = \{\boldsymbol{\mu}'_i\}_{i=1}^{M'}$. $M' \leq M$. We also have $\boldsymbol{\mu}'_i \perp \boldsymbol{c}_k$ for $i \in [M'], k \in [K]$. Let $\mathcal{T}'$ denote the set of testing tasks, which all operate on patterns in $\mathcal{M}'$ rather than $\mathcal{M}$ in training tasks in $\mathcal{T}$. Every testing task $f = f_K \circ \cdots f_2 \circ f_1 \in \mathcal{T}'$ is a composition of $K$ functions. The reasoning for the testing query is considered to be *noiseless* and *accurate*. That means,

$$\boldsymbol{z}_k \in \mathcal{M}' \text{ for all } k \in \{0\} \cup [K], \text{ and } \boldsymbol{z}_k = f_k(\boldsymbol{z}_{k-1}), \boldsymbol{z}_0 = \boldsymbol{x}_{query}.$$

**Testing prompt**: We consider the general setup that testing examples are *noisy* and *erroneous*. By noisy examples, we mean all inputs and outputs of each step are noisy versions of TSR patterns, i.e.,

$$\boldsymbol{x}_i, \boldsymbol{y}_{i,k} \in \{\boldsymbol{b} \in \mathbb{R}^d | \boldsymbol{b} = \boldsymbol{\mu}'_j + \boldsymbol{\delta}, j \in [M'], \boldsymbol{\delta} \perp \mathcal{M}', \|\boldsymbol{\delta}\| \leq \sqrt{2}/2\}, \tag{10}$$

with noise $\boldsymbol{\delta} \neq 0$ for $i \in [K l_{ts}^f], k \in [K]$. Denote TSR $: \mathbb{R}^d \mapsto \mathbb{Z}^+$ as a function that outputs the index of the TSR pattern of the noisy input. We consider the case that at least an $\alpha'$ fraction of context examples where the TSR pattern of the input $\boldsymbol{y}_{s,1}, s \in [l_{ts}^f]$ is the same as $\boldsymbol{x}_{query}$.

By erroneous examples, we mean that the reasoning steps in test examples may contain errors. To formally model this, we define the **step-wise transition matrices** $\{\boldsymbol{A}_k^f\}_{k=1}^K \in \mathbb{R}^{M' \times M'}$ such that $\boldsymbol{A}_k^f$ represents the reasoning probabilities of step $k$ in test examples. Specifically, there exists some constant $\rho^f$ in $(0,1)$ such that for all $s \in [l_{ts}^f], k \in [K]$, the $i,j$-th entry of $\boldsymbol{A}_k^f$ satisfies

$$A_{k(i,j)}^f = \Pr(\text{TSR}(\boldsymbol{y}_{s,k}) = j | \text{TSR}(\boldsymbol{y}_{s,k-1}) = i),$$
$$\text{and } A_{k(i,j^*)}^f \geq 1/(1-\rho^f) \cdot A_{k(i,j)}^f, \forall j \in [M'], \text{ where } \boldsymbol{\mu}'_{j^*} = f_k(\boldsymbol{\mu}'_i), \tag{11}$$

Note that (11) characterizes a general case in inference that for any given $k$, in the $k$-th reasoning step of the test example, the $k$-th step output is a noisy version of the true label with the highest probability, which guarantees that the examples are overall informative in the $k$-th step. This requirement is intuitive because otherwise, these examples would overall provide inaccurate information on the $k$-th step reasoning. Moreover, (11) models the general case that, with some probability, the $k$-step reasoning is inaccurate in the examples. $\rho^f$ is referred to as the **primacy** of the step-wise transition matrices. $\rho^f$ reflects the difference in the probability of correct reasoning and incorrect reasoning in each step, and a larger $\rho^f$ indicates a larger probability of accurate reasoning.

Let $\boldsymbol{B}^f = \prod_{k=1}^K \boldsymbol{A}_k^f$ be the $K$**-step transition matrix**. Then $B_{(i,j)}^f$ is the probability that the $K$-th step output is a noisy version of $\boldsymbol{\mu}'_j$, when the input is a noisy version of $\boldsymbol{\mu}'_i$ in the testing example. We similarly define $\rho_o^f$ in $(0,1)$ as the primacy of $\boldsymbol{B}^f$, where

$$B_{(i,j^*)}^f \geq 1/(1-\rho_o^f) \cdot B_{(i,j)}^f, \ \forall j \in [M'], \ j^* = \arg\max_{j \in [M']} B_{(i,j)}^f. \tag{12}$$

**Example 1.** *Consider a simple two-step inference example with $K = 2$, $\boldsymbol{\mu}'_1$, $\boldsymbol{\mu}'_2$ as the TSR pattern, and $\boldsymbol{\delta} = 0$ in inputs and outputs of every step, as shown in Figure 1. The black solid arrows denote the correct inference process, where $f_1(\boldsymbol{\mu}'_1) = \boldsymbol{\mu}'_1$, $f_1(\boldsymbol{\mu}'_2) = \boldsymbol{\mu}'_2$, $f_2(\boldsymbol{\mu}'_1) = \boldsymbol{\mu}'_2$, and $f_2(\boldsymbol{\mu}'_2) = \boldsymbol{\mu}'_1$. Hence, $\boldsymbol{\mu}'_1 \to \boldsymbol{\mu}'_1 \to \boldsymbol{\mu}'_2$ and $\boldsymbol{\mu}'_2 \to \boldsymbol{\mu}'_2 \to \boldsymbol{\mu}'_1$ are two inference **trajectories** under the function $f$. The testing examples contain errors and follow the transition matrices $\boldsymbol{A}_1^f$ and $\boldsymbol{A}_2^f$ (brown dashed arrows). We let $\boldsymbol{A}_1^f = \begin{pmatrix} 0.6 & 0.4 \\ 0.4 & 0.6 \end{pmatrix}$, $\boldsymbol{A}_2^f = \begin{pmatrix} 0.4 & 0.6 \\ 0.8 & 0.2 \end{pmatrix}$, which results in $\boldsymbol{B}^f = \begin{pmatrix} 0.56 & 0.44 \\ 0.64 & 0.36 \end{pmatrix}$.*

### 3.3 THE SAMPLE COMPLEXITY ANALYSIS OF THE TRAINING STAGE

We first characterize the convergence and the testing performance of the model during the training stage with sample complexity analysis in Theorem 1.

**Theorem 1.** *For any $\epsilon > 0$, when (i) the number of context examples in every training sample is*

$$l_{tr} \geq \Omega(\alpha^{-1}), \tag{13}$$

*(ii) the number of iterations satisfies*

$$T \geq \Omega(\eta^{-1}\alpha^{-2}K^3 \log \frac{K}{\epsilon} + \eta^{-1}MK(\alpha^{-1} + \epsilon^{-1})), \tag{14}$$

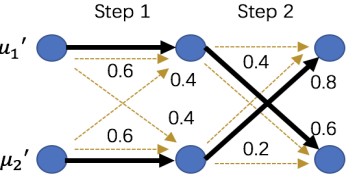

Figure 1: An example of a two-step inference

*and (iii) the training tasks and samples are selected such that every TRR pattern is equally likely in every inference step and in each training batch[4] with batch size $B \geq \Omega(\max\{\epsilon^{-2}, M\} \cdot \log M)$, the step size $\eta < 1$ and $N = BT$ samples, then with a high probability, the returned model guarantees*

$$\mathbb{E}_{\boldsymbol{x}_{query} \in \mathcal{M}, f \in \mathcal{T}} \left[ \ell(\Psi; \boldsymbol{P}, \boldsymbol{z}) \right] \leq \mathcal{O}(\epsilon). \tag{15}$$

Theorem 1 indicates that with long enough training prompts and a sufficient number of iterations and samples for training, a one-layer Transformer can achieve a diminishing loss of $\mathcal{O}(\epsilon)$ on data following the same distribution as training examples. The results indicate that (i) the required number of context examples is proportional to $\alpha^{-1}$; (ii) the required number of iterations and samples increases as $M$ and $\alpha^{-2}$ increases. As a sanity check, these bounds are consistent with the intuition that it will make the training stage more time- and sample-consuming if the number of TRR patterns increases or the fraction of prompt examples that share the same TRR pattern as the query decreases.

## 3.4 CoT generalization guarantee

In this section, we first define two quantities, $\tau^f$, and $\tau_o^f$ for each testing task $f \in \mathcal{T}'$ based on the formulation of testing data and tasks in Section 3.2. These two quantities are used to characterize the CoT and ICL generalization in Theorems 2 and 3, respectively.

**Definition 1.** *For $f = f_K \circ \cdots f_1 \in \mathcal{T}'$, we define the **min-max trajectory transition probability** as:*

$$\tau^f = \min_{i \in [M']} \prod_{k=1}^{K} A_{k(TSR(f_{k-1} \circ \cdots f_0(\boldsymbol{\mu}_i')), TSR(f_k \circ \cdots f_0(\boldsymbol{\mu}_i')))}^f, \text{ where } f_0(\boldsymbol{\mu}_i') := \boldsymbol{\mu}_i', \forall i \in [M'], \tag{16}$$

*which measures the minimum probability, over all the initial TSR patterns, of the $K$-step reasoning trajectory that has the highest probability over all $K$-step trajectories. We also define the **min-max input-label transition probability** as*

$$\tau_o^f = \min_{i \in [M']} \max_{j \in [M']} B_{i,j}^f, \tag{17}$$

*which measures the minimum probability, over all the initial TSR patterns, of the output that has the highest probability over outputs.*

For instance, in Example 1 after (12), $\tau^f = \min\{0.36, 0.48\} = 0.36$, $\tau_o^f = \min\{0.56, 0.64\} = 0.56$.

**Theorem 2** (CoT generalization). *Given a trained model, the training process of which satisfies conditions (i) to (iii) in Theorem 1, then as long as*

*(iv) each TSR pattern $\boldsymbol{\mu}_j'$ in the orthonormal set $\{\boldsymbol{\mu}_j'\}_{j=1}^{M'}$ satisfies*

$$\boldsymbol{\mu}_j' = \boldsymbol{\lambda}_j + \tilde{\boldsymbol{\mu}}_j \tag{18}$$

*where $\boldsymbol{\lambda}_j \perp span(\boldsymbol{\mu}_1, \cdots, \boldsymbol{\mu}_M)$, $\tilde{\boldsymbol{\mu}}_j \in span(\boldsymbol{\mu}_1, \cdots, \boldsymbol{\mu}_M)$, and $\|\tilde{\boldsymbol{\mu}}_j\| \geq \Theta((\log \epsilon^{-1})^{-1})$, and (v) the number of testing examples for any $f \in \mathcal{T}'$ is*

$$l_{ts}^f \geq \Omega((\alpha' \tau^f \rho^f)^{-2} \log M), \tag{19}$$

*we have $\bar{R}_{CoT, \boldsymbol{x}_{query} \in \mathcal{M}', f \in \mathcal{T}'}^f(\Psi) = 0$.*

**Remark 1.** *Theorem 2 proves that a trained one-layer Transformer can generate all $K$-steps reasoning correctly by CoT for a new task $f$ in $\mathcal{T}'$ with two additional conditions. Condition (iv) means that each TSR pattern in the task set $\mathcal{T}'$ is the summation of a component that belongs to the span of the TRR patterns and a component that is perpendicular to the span.*

*Condition (v) indicates that, to achieve the desired CoT accuracy, the number of context examples should be proportional to $\alpha'^{-2}$, $\rho_s^{f-2}$, and $\tau_s^{f-2}$, meaning it decreases as $\alpha'$, $\rho_s^f$, or $\tau_s^f$ increase. It can be interpreted as follows, if the number of context examples remains fixed, an increase in $\alpha'$, $\rho_s^f$, or $\tau_s^f$ results in improved CoT accuracy. This aligns with intuition, because $\alpha'$ represents the fraction of examples similar to the query, and $\rho^f$ and $\tau^f$ reflect the accuracy of the reasoning steps in the context examples.*

---

[4]Our analysis assumes that the whole set of $\mathcal{M}$ is achievable uniformly in each step and training batch. This condition is to ensure a balanced gradient update among all TRR patterns, as used in (Li et al., 2024a) for ICL.

### 3.5 ICL Generalization and Comparison with CoT

Because only input-label pairs are used as context examples for ICL, the input-label pairs in context examples should be accurate overall to be informative about the task. We formulate this requirement as Condition 1.

**Condition 1.** *For the testing task $f = f_K \circ \cdots \circ f_1 \in \mathcal{T}'$, we have that for any $i \in [M']$,*

$$TSR(f(\boldsymbol{\mu}'_i)) = \arg \max_{j \in [M']} B^f_{(i,j)}. \tag{20}$$

Condition 1 requires that in a context example, if the input TSR is $\boldsymbol{\mu}'_i$, then $f(\boldsymbol{\mu}'_i)$ is the output TSR pattern with the highest probability over all TSR patterns. Note that (11) indicates that, for every $k$ and $i$, when $\boldsymbol{\mu}'_i$ is the $k$-th step input, $f_k(\boldsymbol{\mu}'_i)$ is the step-$k$ output with the highest probability over all TSR patterns. However, (11) does not necessarily imply (20). In Example 1, given the input $\boldsymbol{\mu}'_1$, although the inference trajectory $\boldsymbol{\mu}'_1 \to \boldsymbol{\mu}'_1 \to \boldsymbol{\mu}'_2$ under $f$ has the highest probability over all 2-step trajectories, $\boldsymbol{\mu}'_1$ has the higher probability to be the final output than the correct output $\boldsymbol{\mu}'_2$ by the two-step transition matrix $\boldsymbol{B}^f$, thus violating Condition 1.

Our result of the ICL generalization is stated as follows.

**Theorem 3** (ICL generalization). *Given a trained model, the training process of which satisfies conditions (i) to (iii) of Theorem 1 and (18), for the testing task $f \in \mathcal{T}'$,*

*Case A.  if Condition 1 does not hold, then $\bar{R}^f_{ICL, \boldsymbol{x}_{query} \in \mathcal{M}', f \in \mathcal{T}'}(\Psi) \geq \Omega(1)$, no matter how large the number of training samples $l^f_{ts}$ is;*

*Case B.  if Condition 1 holds, then $\bar{R}^f_{ICL, \boldsymbol{x}_{query} \in \mathcal{M}', f \in \mathcal{T}'}(\Psi) = 0$, provided that*

$$l^f_{ts} \geq \Omega((\alpha' \tau^f_o \rho^f_o)^{-2} \log M). \tag{21}$$

**Remark 2** (Comparison between CoT and ICL). *Theorem 3(a) formally states that, Condition 1 is necessary for a successful ICL generalization. Because Condition 1 is not required for CoT generalization, CoT performs better than ICL if Condition 1 fails[5]. Theorem 3(b) characterizes that when Condition 1 holds, a desired ICL generalization needs a testing prompt length linear in $\alpha'^{-2}$, $\rho^{f-2}_o$, and $\tau^{f-2}_o$ for the testing task $f \in \mathcal{T}'$. This result is the counterpart of the requirement (19) for the CoT generalization, indicating that more context examples with the same TSR pattern as the query and more accurate context examples improve ICL generalization.*

Ref. Li et al. (2023c) also shows the advantage of CoT over ICL to learn MLP functions, but in a different setting from ours, where our studied tasks operate on patterns. More importantly, this paper characterizes the CoT and ICL performance theoretically when the testing task has a distribution shift from training tasks (TRR patterns to TSR patterns), and the testing examples contain errors, while Li et al. (2023c) only empirically evaluates the CoT and ICL performance with noisy examples.

## 4 THE MECHANISM OF CoT AND THE PROOF SKETCH

### 4.1 TRANSFORMERS IMPLEMENT CoT BY ATTENDING TO THE MOST SIMILAR EXAMPLES EVERY STEP

We characterize the key mechanism of a properly trained one-layer Transformer to implement CoT on a $K$-steps reasoning task via training dynamics analysis of the attention layer, as demonstrated in Figure 2. This is different from the mechanism study in (Li et al., 2023c; Feng et al., 2023) by constructing a model that can conduct CoT. We have the following proposition for the trained model.

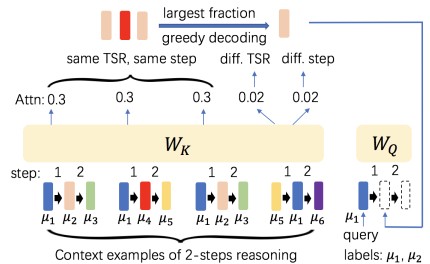

**Proposition 1.** *Let $\mathcal{S}^*_k$ denote the index set of the context columns of the testing prompt $\boldsymbol{P}$ in (4) that (a) correspond to the $k$-th step in a context example and (b) share the*

Figure 2: Concentration of attention weights for CoT inference.

---

[5]Our insight of the comparison between CoT and ICL still holds when we evaluate CoT generalization only by the final step output. This is because a successful CoT generalization in Theorem 2 on all reasoning steps already ensures a satisfactory CoT generalization on the final step.

*same TSR pattern in the $k$-th input as the $k$-th input $\boldsymbol{v}_{k-1}$ of the query, $k \in [K]$. Given a trained model that satisfies conditions (i) to (iii) of Theorem 1 and (18) and (19) after $T$ iterations, we have*

$$\sum_{i \in \mathcal{S}_k^*} softmax(\tilde{\boldsymbol{p}}_i^\top \boldsymbol{W}^{(T)} \tilde{\boldsymbol{q}}_k) \geq 1 - \epsilon, \text{ where } \tilde{\boldsymbol{p}}_i = \boldsymbol{p}_i + \boldsymbol{c}_{(i \mod K)}, \tilde{\boldsymbol{q}}_k = \boldsymbol{q}_k + \boldsymbol{c}_k, \quad (22)$$

*with $\boldsymbol{q}_k$ defined in (6). Moreover, for any $f \in \mathcal{T}'$, the $k$-th step output $\boldsymbol{v}_k$ given $\boldsymbol{x}_{query} = \boldsymbol{\mu}_i'$ satisfies,*

$$\boldsymbol{v}_k = f_k \circ \cdots \circ f_1(\boldsymbol{\mu}_i'). \quad (23)$$

Proposition 1 first illustrates that, when conducting the $k$-th step reasoning of the query for any $k \in [K']$, the trained model assigns dominant attention weights on the prompt columns that are also the $k$-th step reasoning of examples and share the same TSR pattern in the $k$-th step input as the query. Then, given a sufficient number of testing context examples by (19), it is ensured that the fraction of the correct TSR pattern is the largest in the output of each step by (11). Subsequently, the generation by greedy decoding (5) is correct in each step, leading to a successful CoT generalization.

### 4.2 An Overview of the Proof

The technical challenges of the proof are concentrated on Theorem 1, where the property of the trained model is derived. The proof of Theorem 1 is built upon three Lemmas, which characterize the **two stages of the training dynamics**, i.e., Transformers first attend to tokens with the same step as the query and then, among them, further concentrate on tokens that share the same TSR pattern as the query. Specifically, Lemmas 3 and 4 show that if a training prompt $\boldsymbol{P}$ includes the first $k$ steps of the reasoning query, then the attention weights on columns of $\boldsymbol{P}$ with a different step from the query decrease to be close to zero in the first stage. Lemma 5 computes the gradient updates in the second stage, where the attention weights on columns in $\boldsymbol{P}$ that correspond to the same step and have the same TRR pattern as the query gradually become dominant. Theorem 1 unveils this training process by showing the required number of training iterations and sample complexity.

To prove Theorem 2, we first compute the required number of context examples for the new task $f \in \mathcal{T}'$ so that by concentration inequalities, the number of context examples with accurate TSR is larger than examples with inaccurate TSR patterns in all $K$ reasoning steps with high probability. Then, by the correlation between TRR and TSR patterns (18), we also show that the trained Transformer can attend to context columns with the same TSR pattern as the query. Therefore, the model can make the correct generation in each step. Theorem 3 follows a similar proof idea to Theorem 2, with the difference that the trained model predicts output directly from the input query following $\boldsymbol{B}^f$ instead of $\boldsymbol{A}_k^f, k \in [K]$ in CoT. Therefore, Condition 1 is required for the success of ICL generalization.

## 5 Numerical Experiments

**Data Generation and Model setup.** We use synthetic data generated following Sections 2 and 3.2. Let $d_{\mathcal{X}} = 30$, $M = 20$, $M' = 10$, $\alpha = 0.4$. We consider 3-steps tasks for training and testing, i.e., $K = 3$. A reasoning task $f$ is generated by first sampling a set of numbers of permutations $\{p_i\}_{i=1}^M$ with $p_i \in [M]$ and then let $f_k(\boldsymbol{\mu}_{p_i}) = \boldsymbol{\mu}_{p_{((i+k) \mod M)}}$ for $i \in [M], k, j \in [K]$. The testing noise level is set to be 0.2 for any examples and $f \in \mathcal{T}'$. The learning model is a one-layer single-head Transformer defined in (2) or a three-layer two-head Transformer. We set $\tau^f = 0.5$, $\rho^f = 0.8$, $\alpha' = 0.8$ for CoT testing if not otherwise specified.

**Experiments on the generalization of CoT.** We first verify the required number of context examples for a desired CoT generalization on a one-layer Transformer. We investigate the impact of $\alpha', \tau^f$, and $\rho^f$ by varying one and fixing the other two. Figure 3 illustrates that more testing examples are needed when $\alpha', \tau^f$, or $\rho^f$ is small, which verifies the trend of the lower bound of $l_{ts}^f$ in (19).

**Experiments on the generalization of ICL and a comparison with CoT.** We then verify the ICL generalization with the trained model. We vary $\tau_o^f$ and $\rho_o^f$ by changing $\tau^f$ and $\rho^f$. Figure 3 indicates that more testing examples are required when $\alpha', \tau_o^f$, or $\rho_o^f$ is small, which is consistent with our bound in (21). We then consider the case where $\tau_o^f = 0.4$ and $\rho_o^f = 0.1$ so that the generated testing prompt may not satisfy Condition 1 depending on the specific choices of $A_k^f$'s. Figure 5 shows that when Condition 1 holds, the ICL testing error decreases if the number of contexts increases. However, when Condition 1 fails, the ICL testing error remains large, irrespective of the number of contexts.

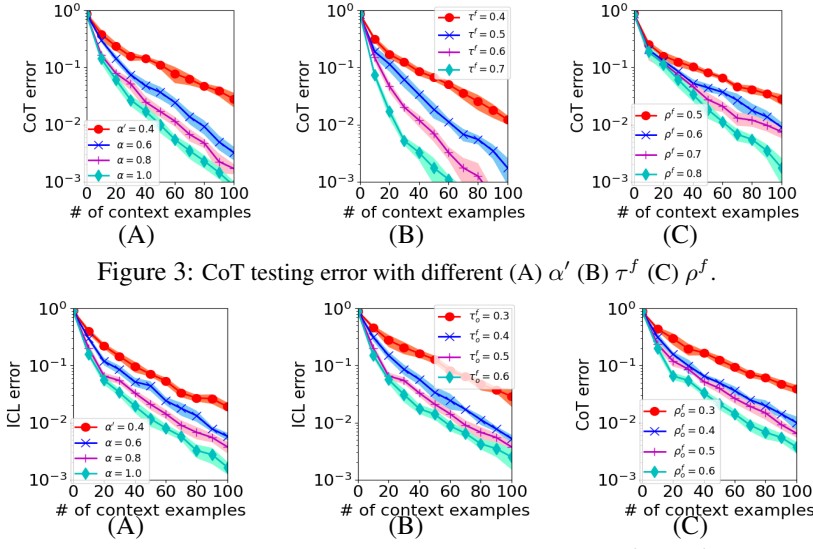

Figure 3: CoT testing error with different (A) $\alpha'$ (B) $\tau^f$ (C) $\rho^f$.

Figure 4: ICL testing error with different (A) $\alpha'$ (B) $\tau_o^f$ (C) $\rho_o^f$.

**Experiments on the training dynamics of CoT.** In Figure 6, we compute the total attention weights on four types of testing context columns along the training, which are contexts with the same (or different) TSR pattern and in the same (or different) step as the query. The result shows that the attention weights on contexts that share the same TSR pattern and in the same step as the query increase along the training and converge to around 1. This verifies the mechanism formulated in (22). Meanwhile, Figure 6 also justifies the two-stage training dynamics proposed in Section 4.2, where we add a

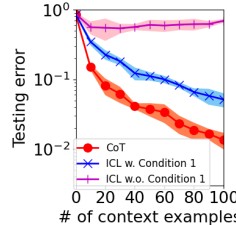 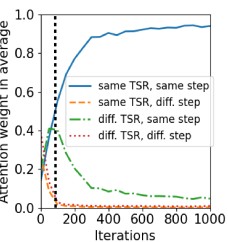

Figure 5: Comparison between CoT and ICL w./w.o. Condition 1

Figure 6: Training dynamics of Transformers for CoT

black vertical dashed line to demonstrate the stage transition boundary. We observe that the attention weights on context columns with a different step, i.e., the red and yellow curves, decrease to zero in the first stage. Then, the attention weights on contexts with the same TSR pattern and the same step as the query, i.e., the blue curve, increase to 1 in the second stage. We also justify the attention mechanism of CoT on a three-layer two-head Transformer with a two-step reasoning task. Figure 7 shows that there exists at least one head in each layer of the Transformer that implements CoT as characterized in Proposition 1. This indicates that the CoT mechanism we characterize on one-layer Transformers can be extended to multi-layer multi-head Transformers.

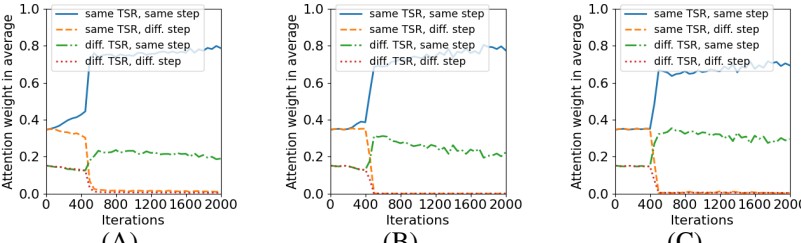

Figure 7: Training dynamics of Transformers. (A) Layer 1, Head 2 (B) Layer 2 Head 2 (C) Layer 3 Head 2.

## 6 CONCLUSION, LIMITATIONS, AND FUTURE WORKS

This paper theoretically analyzes the training dynamics of Transformers with nonlinear attention, together with the CoT generalization ability of the resulting model on new tasks with noisy and partially inaccurate context examples. We quantitatively characterize and compare the required conditions for the success of CoT and ICL. Although based on a simplified Transformer model and reasoning tasks operating on patterns, this work deepens the theoretical understanding of the CoT mechanism. Future directions include designing efficient prompt-generating methods for CoT and analyzing LLM reasoning on a more complicated data model.

ACKNOWLEDGMENTS

This work was supported by National Science Foundation(NSF) #2430223, Army Research Office (ARO) W911NF-25-1-0020, and the Rensselaer-IBM Future of Computing Research Collaboration (http://airc.rpi.edu). We also thank all anonymous reviewers for their constructive comments.

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

# APPENDIX

## A    EXPERIMENTS ON REAL-WORLD DATA

We consider a simple arithmetic task that outputs $((A_1 o_1 A_2) o_2 A_3) o_3 A_4$ given $A_1, A_2, A_3, A_4$ chosen from integers from 0 to 9 as the input, where $o_1, o_2, o_3 \in O = \{+, -, \times\}$. The CoT output follows the format of $A_1 o_1 A_2 = S_1, S_1 o_2 A_3 = S_2, S_2 o_3 A_4 = S_3$ and will be evaluated by whether all the three steps are correct for the query as (7). ICL directly outputs $S_3$, and the performance is evaluated by the prediction accuracy of $S_3$ as (9). In the following experimental settings, the accuracy is computed on 50 prompts. Each prompt contains three context examples. The inference model is GPT-4 (OpenAI, 2023).

**An increasing number of erroneous examples hurts the CoT generalization.**   To model the errors in the context examples in the testing prompt, we replace $o_3$ with one operation $\hat{o}_3$ from $O \backslash o_3$ in the presentation of some of the context examples in the testing prompt. Note that the output values $S_3$ are still correctly computed from $S_3 = S_2 o_3 A_4$. Table 1 shows that when the total number of testing examples is fixed to be three, with the increasing number of incorrect examples, the testing accuracy decreases. This is consistent with Remark 1 for Theorem 2.

| # of incorrect examples | 0 | 1 | 2 | 3 |
|---|---|---|---|---|
| CoT accuracy | 100% | 100% | 56% | 0% |

Table 1: The accuracy with different numbers of incorrect examples for CoT. Errors in presenting $o_3$.

**CoT is more robust to erroneous examples with implementation error than ICL.**   In this setting, the error in a context examples is introduced by replacing $o_1$ with one operation $\hat{o}_1$ randomly and independently selected from $O \backslash o_1$. Hence, $S_1 = A_1 \hat{o}_1 A_2$, and the successive computation are based on the wrongly computed $S_1$. The results in Table 2 shows that when two incorrect examples exist, CoT performs better than ICL, which justifies Remark 2 for Theorem 3.

| # of incorrect examples | 0 | 1 | 2 |
|---|---|---|---|
| CoT accuracy | 100% | 100% | 100% |
| ICL accuracy | 100% | 100% | 60% |

Table 2: The accuracy with different numbers of incorrect examples for CoT and ICL. Errors in implementing $o_1$.

## B    ADDITIONAL DISCUSSIONS

### B.1    THE MOTIVATION TO STUDY ONE-LAYER SINGLE-HEAD TRANSFORMERS

The reasons we study one-layer single-head attention-only nonlinear Transformers in this work are as follows.

First, it is much more challenging to theoretically analyze the training dynamics and generalization of multi-layer/head Transformers. This is because the loss landscape for multi-layer/head Transformers is highly nonlinear and non-convex due to the interactions between multiple nonlinear functions. The simplified data helps to characterize the gradient updates in different directions for different patterns and steps. Non-orthogonal data make the updates less separable for different inputs, which is more challenging to analyze.

Second, the state-of-the-art theoretical works (Li et al., 2023a; 2024a; Huang et al., 2023; Makkuva et al., 2024a; Ildiz et al., 2024) on optimization and generalization also focus mainly on one-layer Transformers. No existing works study the optimization and generalization of CoT even for one-layer Transformers. Therefore, we plan to focus on the one-layer analysis to obtain more theoretical insights. We leave the theoretical analysis of the multi-layer case as future works.

Third, although we admit the gap between theory and practice, our theory still makes contributions under our settings. Our work is the first one to investigate the optimization and generalization of CoT and characterize the conditions when CoT is better than ICL. We establish the required number of context examples for a successful CoT in terms of how informative and erroneous the prompt is.

We also implement experiments on the attention mechanism for three-layer two-head Transformers on two-step reasoning tasks. Please see Figure 7 for details. The findings of all three layers are generally consistent with Proposition 1 for the single-layer single-head case, which indicates that the CoT mechanism we characterize on one-layer Transformers can be extended to multi-layer multi-head Transformers.

## B.2 THE MOTIVATION OF THE DATA AND TASK FORMULATION

There are several reasons for using such data formulation.

First, our data formulation of orthogonal patterns, on which the function is based, is widely used in the state-of-the-art theoretical study of model training or ICL on language and sequential data [(Tian et al., 2023a; Huang et al., 2023; Li et al., 2024a; Chen et al., 2024). For example, (Huang et al., 2023; Li et al., 2024a) study ICL on regression or classification tasks, which also use orthogonal patterns as data. Sections 2.1 and 2.2 in (Chen et al., 2024) consider learning n-gram data in ICL by formulating transitions between orthogonal patterns. Section 3 of (Tian et al., 2023a) also assume orthogonal patterns in Transformer model training, and the generation comes from the orthogonal pattern set. The data formulation we use is consistent with the existing theoretical works.

Second, based on this formulation, one can characterize the gradient updates in different directions for different patterns and steps. This enables us to distinguish the impact of different patterns and steps in the convergence analysis of CoT using Transformers. Non-orthogonal data make the model updates less separable for different inputs, which is more challenging to analyze. Moreover, we would like to mention that during the inference, the tokens in testing prompts contain noises as defined in Equation 10. This makes the tokens of different TSR patterns not orthogonal to each other and relaxes our orthogonality condition to some degree.

## B.3 THE DISCUSSION OF POSITIONAL ENCODING

The positional encoding (PE) we use is simplified for theoretical analysis. The formulation of PE we use is motivated by (Huang et al., 2024; Nichani et al., 2024), where each token is added with a PE represented by orthogonal vectors. These works formulate the distribution of the PE to be related to the structure of the data, such as patch-wise association (Huang et al., 2024), and sparse token selection (Nichani et al., 2024). Likewise, we follow their intuition to make the PE vary in different steps of our reasoning tasks so that the Transformer can distinguish different steps when making inferences for the query.

Our analysis can be extended to study more general PEs with additional technical work in the future. One possible direction is studying the family of periodic and separable PE. For example, the absolute PE proposed by (Vaswani et al., 2017) considers PE as a sinusoid, which is periodic. Such analysis can be made by relaxing the "orthogonality" of PE vectors to a certain "separability" between PE vectors.

We also conduct experiments on a three-layer single-head Transformer with the standard PE proposed in Section 3.5 of (Vaswani et al., 2017) for our problem. Figure shows that the blue curve increases to be the largest along the training, which means the attention weights on example steps that share the same TSR pattern and the same step as the query. This indicates that the CoT mechanism of using standard PE is the same as the one proposed in Proposition 1 in our paper. One might note that the scores of the blue curve are not as high as Figure 6 in our paper. We guess the reason why the distinction in attention values is more significant in our PE may be the additional orthogonality of our PE and the property that its period is the same as the reasoning length. Nevertheless, the strong similarity between the results on standard PE and our used PE shows the practical significance of our analysis.

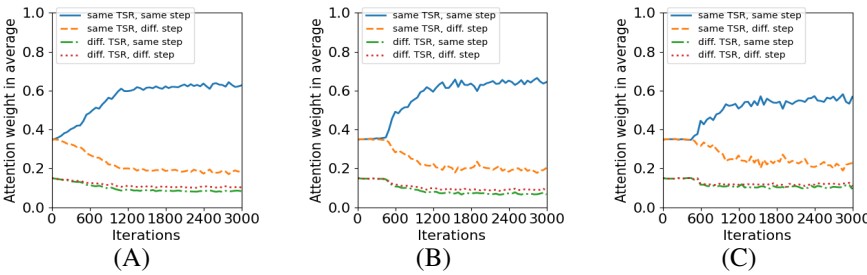

Figure 8: CoT mechanism with standard PE of (A) Layer 1 (B) Layer 2 (C) Layer 3.

## C  ALGORITHMS

We first present the training algorithm introduced in Section 2.2.

---

**Algorithm 1** Training with Stochastic Gradient Descent (SGD)

---

1: **Hyperparameters:** The step size $\eta$, the number of iterations $T$, batch size $B$.
2: **Initialization:** Let $\boldsymbol{W} = \boldsymbol{W}_K^\top \boldsymbol{W}_Q$ and $\boldsymbol{W}_V = (\boldsymbol{0}_{d_\mathcal{X} \times d_\mathcal{X}} \ \boldsymbol{I}_{d_\mathcal{X}} \ \boldsymbol{0}_{d_\mathcal{X} \times d_\varepsilon})$. Each entry of $\boldsymbol{W}^{(0)}$ is generated from $\mathcal{N}(0, \xi^2)$ for a small constant $\xi > 0$. $\boldsymbol{W}_V$ and $\boldsymbol{a}$ are fixed during the training.
3: **Training by SGD:** For each iteration, we independently sample $\boldsymbol{x}_{query} \sim \mathcal{D}$, $f \in \mathcal{T}_{tr}$ to form a batch of training prompt and labels $\{\boldsymbol{P}^n, z^n\}_{n \in \mathcal{B}_t}$ as introduced in Section 3.2. Each TRR pattern is sampled equally likely in each batch. For each $t = 0, 1, \cdots, T - 1$

$$\boldsymbol{W}^{(t+1)} = \boldsymbol{W}^{(t)} - \eta \cdot \frac{1}{B} \sum_{n \in \mathcal{B}_t} \nabla_{\boldsymbol{W}^{(t)}} \ell(\Psi^{(t)}; \boldsymbol{P}^n, \boldsymbol{z}^n). \tag{24}$$

4: **Output:** $\boldsymbol{W}^{(T)}$.

---

We then summarize the algorithm of the CoT inference introduced in Section 2.3 as follows.

---

**Algorithm 2** Inference with Chain-of-Thought (CoT)

---

1: **Input**: $\boldsymbol{z}_0 = \boldsymbol{v}_0 = \boldsymbol{x}_{query}$, $\boldsymbol{P}_0$, and $\boldsymbol{P}_1$.
2: **for** $k = 1, \cdots, K - 1$, **do**

  Compute $\boldsymbol{v}_k$ by greedy decoding in (5). Then update $\boldsymbol{P}_k$ and $\boldsymbol{P}_{k+1}$ by (6). $\qquad$ (25)

3: **end for**
4: **Output**: $\boldsymbol{v}_1, \boldsymbol{v}_2, \cdots, \boldsymbol{v}_{K-1}$, and $\boldsymbol{v}_K$ by (5).

---

## D  PRELIMINARIES

We first summarize the notations we use in this paper in Table 3.

**Lemma 1** (Multiplicative Chernoff bounds, Theorem D.4 of (Mohri et al., 2018))**.** *Let $X_1, \cdots, \boldsymbol{X}_m$ be independent random variables drawn according to some distribution $\mathcal{D}$ with mean $p$ and support included in $[0, 1]$. Then, for any $\gamma \in [0, \frac{1}{p} - 1]$, the following inequality holds for $\hat{p} = \frac{1}{m} \sum_{i=1}^m X_i$:*

$$\Pr(\hat{p} \geq (1 + \gamma)p) \leq e^{-\frac{mp\gamma^2}{3}}, \tag{26}$$

$$\Pr(\hat{p} \leq (1 - \gamma)p) \leq e^{-\frac{mp\gamma^2}{2}}. \tag{27}$$

**Definition 2** ((Vershynin, 2010))**.** *We say $X$ is a sub-Gaussian random variable with sub-Gaussian norm $K > 0$, if $(\mathbb{E}|X|^p)^{\frac{1}{p}} \leq K\sqrt{p}$ for all $p \geq 1$. In addition, the sub-Gaussian norm of $X$, denoted $\|X\|_{\psi_2}$, is defined as $\|X\|_{\psi_2} = \sup_{p \geq 1} p^{-\frac{1}{2}} (\mathbb{E}|X|^p)^{\frac{1}{p}}$.*

**Lemma 2** (Vershynin (2010) Proposition 5.1, Hoeffding's inequality)**.** *Let $X_1, X_2, \cdots, X_N$ be independent centered sub-gaussian random variables, and let $K = \max_i \|\boldsymbol{X}_i\|_{\psi_2}$. Then for every $\boldsymbol{a} = (a_1, \cdots, a_N) \in \mathbb{R}^N$ and every $t \geq 0$, we have*

$$\Pr\left(\Big|\sum_{i=1}^N a_i X_i\Big| \geq t\right) \leq e \cdot \exp\left(-\frac{ct^2}{K^2 \|\boldsymbol{a}\|^2}\right), \tag{28}$$

Table 3: Summary of Notations

| Notations | Annotation |
|---|---|
| $\boldsymbol{x}_i, \boldsymbol{y}_{i,k}, \boldsymbol{x}_{query}, \boldsymbol{z}_k$ | $\boldsymbol{x}_i$ is the input to the first step of a reasoning example. $\boldsymbol{y}_{i,k}$ is the $k$-th step output label of $\boldsymbol{x}_i$. $\boldsymbol{x}_{query}$ is the query input. $\boldsymbol{z}_k$ the $k$-th step output label of $\boldsymbol{x}_{query}$. $k \in [K]$. |
| $\boldsymbol{P}, \boldsymbol{p}_{query}, \boldsymbol{E}_i, \boldsymbol{Q}_k, \boldsymbol{v}_k$ | $\boldsymbol{P}$ is a training or testing prompt that consists of multiple training or testing examples and a query. The last column of $\boldsymbol{P}$ is denoted by $\boldsymbol{p}^n_{query}$, which is the query of $\boldsymbol{P}$. $\boldsymbol{E}_i$ is the $i$-th context example of $\boldsymbol{P}$. $\boldsymbol{Q}_k$ is the first $k$ steps of the reasoning query. $k \in [K]$. $\boldsymbol{v}_k$ is the $k$-th step generation by CoT. $k \in [K]$. |
| $\boldsymbol{c}_i, \tilde{\boldsymbol{p}}_i, \tilde{\boldsymbol{p}}_{query}$ | $\boldsymbol{c}_i$ is the positional encoding for the $i$-th column of the input sequence. $\tilde{\boldsymbol{p}}_i = \boldsymbol{p}_i + \boldsymbol{c}_i$, where $\boldsymbol{p}_i$ is the $i$-th column of $\boldsymbol{P}$. $\tilde{\boldsymbol{p}}_{query}$ is the $\boldsymbol{p}_i$ of the query column. |
| $F(\Psi; \boldsymbol{P}), \ell(\Psi; \boldsymbol{P}^n, \boldsymbol{z}^n)$ | $F(\Psi; \boldsymbol{P}^n)$ is the Transformer output for $\boldsymbol{P}$ with $\Psi$ as the parameter. $\ell(\Psi; \boldsymbol{P}^n, \boldsymbol{z}^n)$ is the loss function value given $\boldsymbol{P}^n$ and the corresponding label $\boldsymbol{z}^n$. |
| $\boldsymbol{\mu}_i \in \mathcal{M}, \boldsymbol{\mu}'_i \in \mathcal{M}', \text{TSR}(\cdot)$ | $\boldsymbol{\mu}_i$ is the $i$-th training-relevant (TRR) pattern for $i \in [M]$. $\boldsymbol{\mu}'_i$ is the $i$-th testing-relevant (TSR) pattern for $i \in [M']$. $\mathcal{M}$ and $\mathcal{M}'$ are the set of TRR and TSR patterns, respectively. $\text{TSR}(\cdot)$ is a function that outputs the index of the TSR pattern of the noisy input. |
| $f_k, f$ | $f$ is the task function with $f = f_K \circ \cdots f_2 \circ f_1$ for a $K$-steps reasoning. $f_k$ is the $k$-th step task function. |
| $\mathcal{T}, \mathcal{T}', \mathcal{D}, \mathcal{D}'$ | $\mathcal{T}$ is the distribution of training tasks, while $\mathcal{T}'$ is the distribution of testing tasks. $\mathcal{D}$ is the training data distribution. $\mathcal{D}'$ is the testing data distribution. |
| $\alpha, \alpha'$ | $\alpha$ (or $\alpha'$) is the fraction of context examples with input sharing the same TRR (or TSR) pattern as the query. |
| $\boldsymbol{A}^f_k, \boldsymbol{B}^f_k$ | $\boldsymbol{A}^f_k$ is the step-wise transition matrix at the $k$-th step for the task $f$, $k \in [K]$. $\boldsymbol{B}^f_k$ is the $K$-steps transition matrix of the task $f$. |
| $\tau^f, \tau^f_o, \rho^f, \rho^f_o$ | $\tau^f$ is the min-max trajectory transition probability for task $f$. $\tau^f_o$ is the min-max input-label transition probability for task $f$. $\rho^f$ and $\rho^f_o$ are primacy of the step-wise transition matrices and the $K$-steps transition matrix, respectively. |
| $\mathcal{S}^*_k$ | The index set of context columns of the prompt that correspond to the $k$-th step of the example and share the same TSR pattern in the $(k-1)$-th output as the $(k-1)$-th output $\boldsymbol{v}_{k-1}$ of the query. |
| $p_n(t)$ | $p_n(t)$ is the summation of attention weights on context columns that share the same TRR/TSR pattern and in the same step as the query. |
| $\mathcal{B}_b$ | $\mathcal{B}_b$ is the SGD batch at the $b$-th iteration. |
| $l_{tr}$ | $l_{tr}$ is the universal number of training context examples. |
| $l^f_{ts}$ | $l_{ts}$ is the number of testing context examples of the task $f$. |
| $\mathcal{O}(), \Omega(), \Theta()$ | We follow the convention that $f(x) = O(g(x))$ (or $\Omega(g(x))$, $\Theta(g(x))))$ means that $f(x)$ increases at most, at least, or in the order of $g(x)$, respectively. |
| $\gtrsim, \lesssim$ | $f(x) \gtrsim g(x)$ (or $f(x) \lesssim g(x)$ ) means that $f(x) \geq \Omega(g(x))$ (or $f(x) \lesssim \mathcal{O}(g(x))$). |

where $c > 0$ is an absolute constant.

**Definition 3.** *Define that for $\tilde{\boldsymbol{p}}_i$ that shares the same TRR/TSR pattern and in the same step as the query,*

$$p_n(t) = \sum_i softmax(\tilde{\boldsymbol{p}}^n_i{}^\top \boldsymbol{W}^{(t)} \tilde{\boldsymbol{p}}^n_{query}). \tag{29}$$

**Lemma 3.** *Given the SGD training scheme described in Section 2.2, $B \geq \Omega(M \log M)$, and $l_{tr} \geq \Omega(\alpha^{-1})$, we have the following results. When $\mathcal{O}(\eta^{-1} \alpha^{-2} K^3 \log \frac{K}{\epsilon}) \geq t \geq 1$, for any $\boldsymbol{p}$ as a*

*column of context examples in (1), we have*

$$\tilde{\boldsymbol{p}}^{\top} \eta \frac{1}{B} \sum_{n \in \mathcal{B}_b} \frac{\partial \ell(\Psi; \boldsymbol{P}^n, \boldsymbol{z}^n)}{\partial \boldsymbol{W}^{(t)}} \tilde{\boldsymbol{p}}$$

$$\leq \frac{\eta}{B} \sum_{n \in \mathcal{B}_b} \left( \frac{1}{KM} (1 - p_n(t))^2 (-4 p_n(t)(1 + \frac{\alpha^2}{K^2}) + \frac{\alpha^2}{K^2}(1 + \frac{2(K-1)}{K})) - \frac{\alpha^2}{K^3}(1 - p_n(t))^2 \right).$$

$$(30)$$

*For any $\tilde{\boldsymbol{p}}'$ that shares the same TRR pattern and a different positional encoding as $\tilde{\boldsymbol{p}}$, we have*

$$\frac{\eta}{B} \sum_{n \in \mathcal{B}_b} \left( \frac{1}{KM}(-4 - (3K-2)(1 - p_n(t))(1 + \frac{\alpha^2}{K^2}))p_n(t)(1 - p_n(t)) + \frac{\alpha^2}{K^3}(1 - p_n(t))^2 \right)$$

$$\leq \tilde{\boldsymbol{p}}'^{\top} \eta \frac{1}{B} \sum_{n \in \mathcal{B}_b} \frac{\partial \ell(\Psi; \boldsymbol{P}^n, \boldsymbol{z}^n)}{\partial \boldsymbol{W}^{(t)}} \tilde{\boldsymbol{p}}$$

$$\leq \frac{\eta}{B} \sum_{n \in \mathcal{B}_b} \left( \frac{1}{KM}(-4 - (3K-2)(1 - p_n(t))(1 + \frac{\alpha^2}{K^2}))p_n(t)(1 - p_n(t)) + \frac{1}{K} p_n(t)(1 - p_n(t))^2 \right.$$

$$\left. \cdot (1 + \frac{\alpha^2}{K^2}) \right).$$

$$(31)$$

*For any $\tilde{\boldsymbol{p}}'$ that shares a different TRR pattern but the same positional encoding as $\tilde{\boldsymbol{p}}$, we have*

$$\eta \cdot \frac{1}{B} \sum_{n \in \mathcal{B}_b} \left( \frac{1}{KM}(-\frac{\alpha^2}{K^2} + (K - 1 + \frac{(2K-1)\alpha^2}{K^2})p_n(t))(1 - p_n(t))^2 - (1 - p_n(t))^2 \frac{\alpha^2}{K^3} \right.$$

$$\left. + \frac{1}{K} \cdot (1 - p_n(t))^2(-p_n(t) + (1 - p_n(t))\frac{\alpha^2}{K^2}) \right)$$

$$\leq \tilde{\boldsymbol{p}}'^{\top} \eta \frac{1}{B} \sum_{n \in \mathcal{B}_b} \frac{\partial \ell(\Psi; \boldsymbol{P}^n, \boldsymbol{z}^n)}{\partial \boldsymbol{W}^{(t)}} \tilde{\boldsymbol{p}}$$

$$\leq \eta \cdot \frac{1}{B} \sum_{n \in \mathcal{B}_b} \left( \frac{1}{KM}(-\frac{\alpha^2}{K^2} + (K - 1 + \frac{(2K-1)\alpha^2}{K^2})p_n(t))(1 - p_n(t))^2 - (1 - p_n(t))^2 \frac{\alpha^2}{K^3} \right).$$

$$(32)$$

*For any $\tilde{\boldsymbol{p}}'$ that shares a different TRR pattern and a different positional encoding from $\tilde{\boldsymbol{p}}$, we have*

$$\eta \cdot \frac{1}{B} \sum_{n \in \mathcal{B}_b} \left( \frac{1}{KM} p_n(t)(1 - p_n(t))^2(1 + \frac{(2-K)\alpha^2}{K^2}) + (1 - p_n(t))^2 \cdot \frac{\alpha^2}{K^3} \right)$$

$$\leq \tilde{\boldsymbol{p}}'^{\top} \eta \frac{1}{B} \sum_{n \in \mathcal{B}_b} \frac{\partial \ell(\Psi; \boldsymbol{P}^n, \boldsymbol{z}^n)}{\partial \boldsymbol{W}^{(t)}} \tilde{\boldsymbol{p}}$$

$$\leq \eta \cdot \frac{1}{B} \sum_{n \in \mathcal{B}_b} \left( \frac{1}{KM} p_n(t)(1 - p_n(t))^2(2 - K + \frac{(2-K)\alpha^2}{K^2}) + (1 - p_n(t))^2 p_n(t)(1 + \frac{\alpha^2}{K^2}) \cdot \frac{1}{K} \right).$$

$$(33)$$

**Lemma 4.** *Given the SGD training scheme described in Section 2.2, $B \geq \Omega(M \log M)$, and $l_{tr} \geq \Omega(\alpha^{-1})$, and*

$$t \gtrsim T_1 := \eta^{-1} \alpha^{-2} K^3 \log \frac{K}{\epsilon}, \tag{34}$$

*we have that if $\boldsymbol{p}_{query}$ is in the $k$-th step,*

$$\sum_{i \in \mathcal{S}_{[K] \backslash k}} softmax(\tilde{\boldsymbol{p}}_i^{\top} \boldsymbol{W}^{(t)} \tilde{\boldsymbol{p}}_{query}) \leq \epsilon \tag{35}$$

*where $\mathcal{S}_{[K] \backslash k}$ means the index set of context columns that are not in the $k$-th step.*

**Lemma 5.** *Given the SGD training scheme described in Section 2.2, $B \geq \Omega(M \log M)$, and $l_{tr} \geq \Omega(\alpha^{-1})$, we have the following results. When $t \geq T_1 = \eta^{-1} \alpha^{-2} K^3 \log \frac{K}{\epsilon}$, for any $\boldsymbol{p}$ as a*

column of context examples in (1), we have

$$\tilde{\boldsymbol{p}}^{\top} \eta \frac{1}{B} \sum_{n \in \mathcal{B}_b} \frac{\partial \ell(\Psi; \boldsymbol{P}^n, \boldsymbol{z}^n)}{\partial \boldsymbol{W}^{(t)}} \tilde{\boldsymbol{p}} \leq -\frac{\eta}{2MB} \sum_{n \in \mathcal{B}_b} 4 p_n(t)(1 - p_n(t))^2. \tag{36}$$

*For any $\tilde{\boldsymbol{p}}'$ that shares the same TRR pattern and a different positional encoding as $\tilde{\boldsymbol{p}}$, we have*

$$\left| \tilde{\boldsymbol{p}'}^{\top} \eta \frac{1}{B} \sum_{n \in \mathcal{B}_b} \frac{\partial \ell(\Psi; \boldsymbol{P}^n, \boldsymbol{z}^n)}{\partial \boldsymbol{W}^{(t)}} \tilde{\boldsymbol{p}} \right| \leq \eta \epsilon. \tag{37}$$

*For any $\tilde{\boldsymbol{p}}'$ that shares a different TRR pattern but the same positional encoding as $\tilde{\boldsymbol{p}}$, we have*

$$\left| \tilde{\boldsymbol{p}'}^{\top} \eta \frac{1}{B} \sum_{n \in \mathcal{B}_b} \frac{\partial \ell(\Psi; \boldsymbol{P}^n, \boldsymbol{z}^n)}{\partial \boldsymbol{W}^{(t)}} \tilde{\boldsymbol{p}} \right| \leq \frac{\eta}{2BM} \sum_{n \in \mathcal{B}_b} p_n(b)(1 - p_n(b))^2. \tag{38}$$

*For any $\tilde{\boldsymbol{p}}'$ that shares a different TRR pattern and a different positional encoding from $\tilde{\boldsymbol{p}}$, we have*

$$\left| \tilde{\boldsymbol{p}'}^{\top} \eta \frac{1}{B} \sum_{n \in \mathcal{B}_b} \frac{\partial \ell(\Psi; \boldsymbol{P}^n, \boldsymbol{z}^n)}{\partial \boldsymbol{W}^{(t)}} \tilde{\boldsymbol{p}} \right| \leq \eta \epsilon. \tag{39}$$

## E    PROOF OF MAIN THEOREMS

### E.1    PROOF OF THEOREM 1

*Proof.* By the condition in Lemma 3, we have that

$$B \geq \Omega(M \log M). \tag{40}$$

We know that there exists gradient noise caused by imbalanced TRR patterns in each batch. Then, by Hoeffding's inequality (28),

$$\Pr \left( \left\| \frac{1}{|\mathcal{B}_b|} \sum_{n \in \mathcal{B}_b} \frac{\partial \ell(\Psi; \boldsymbol{P}^n, z^n)}{\partial \boldsymbol{W}} - \mathbb{E} \left[ \frac{\partial \ell(\Psi; \boldsymbol{P}^n, z^n)}{\partial \boldsymbol{W}} \right] \right\| \geq \left| \mathbb{E} \left[ \frac{\partial \ell(\Psi; \boldsymbol{P}^n, z^n)}{\partial \boldsymbol{W}} \right] \epsilon \right| \right)$$
$$\leq e^{-B\epsilon^2} \leq M^{-C}, \tag{41}$$

if $B \gtrsim \epsilon^{-2} \log M$. Therefore, we require

$$B \gtrsim \max\{\epsilon^{-2}, M\} \log M. \tag{42}$$

By Lemma 5 and Definition 3, for $\tilde{\boldsymbol{p}}_i^n$ that share the same TRR pattern and the same positional encoding of $\tilde{\boldsymbol{p}}_{query}^n$,

$$\frac{p_n(t+1)}{|\mathcal{S}_1^n|} = \text{softmax}(\tilde{\boldsymbol{p}}_i^{n\top} \boldsymbol{W}^{(t+1)} \tilde{\boldsymbol{p}}_{query}^n) \geq \frac{1}{l} \cdot \frac{1}{\frac{\alpha}{K} + (1 - \frac{1}{K}) \cdot \epsilon + (\frac{1}{K} - \frac{\alpha}{K}) e^{-u}}, \tag{43}$$

where by (161),

$$u \gtrsim \frac{\eta}{KM} \sum_{b=0}^{t} (1 - p_n(b))^2 p_n(b). \tag{44}$$

For $\tilde{\boldsymbol{p}}_i^n$ that only share the same positional encoding of $\tilde{\boldsymbol{p}}_{query}^n$,

$$\text{softmax}(\tilde{\boldsymbol{p}}_i^{n\top} \boldsymbol{W}^{(t+1)} \tilde{\boldsymbol{p}}_{query}^n) \geq \frac{1}{l} \cdot \frac{1}{\frac{\alpha}{K} e^u + (1 - \frac{1}{K}) \cdot \epsilon + (\frac{1}{K} - \frac{\alpha}{K})}. \tag{45}$$

Therefore, to make the attention weights between $\tilde{\boldsymbol{p}}_{query}^n$ and $\tilde{\boldsymbol{p}}_i^n$ that share the same TRR pattern and the same positional encoding dominant, we need a large enough $u$. When $1 - p_n(b) \geq \Omega(1)$, we have

$$t \leq T_2 := \eta^{-1} KM\alpha^{-1}. \tag{46}$$

When $1 - p_n(b) \leq O(1)$,

$$p_n(t+1) = \frac{e^u}{e^u + \frac{1 - \frac{\alpha}{K}}{\frac{\alpha}{K}}} \gtrsim 1 - \frac{1 - \frac{\alpha}{K}}{\frac{\alpha}{K}} e^{-u}, \tag{47}$$

and

$$1 - p_n(t+1) \geq \frac{1 - \frac{\alpha}{K}}{\frac{\alpha}{K} e^u + (1 - \frac{\alpha}{K})} \gtrsim \frac{1 - \frac{\alpha}{K}}{\frac{\alpha}{K}} e^{-u}. \tag{48}$$

Then, we prove that when $t$ is large enough, $u(t) \geq \frac{1}{2} \log \frac{\eta(1-\alpha)^2 t}{\alpha^2 KM}$. We show it by induction. Suppose that the conclusion holds when $t = t_0$, then

$$
\begin{aligned}
u(t+1) &\geq \frac{\eta}{KM} \sum_{b=0}^{t_0} (1 - p_n(b))^2 p_n(b) + \frac{\eta}{KM}(1 - p_n(t))^2 p_n(t) \\
&\geq \frac{1}{2} \log \frac{(K-\alpha)^2 t}{2\alpha^2 KM} + \frac{\eta}{KM}(1 - p_n(t))^2 p_n(t) \\
&\geq \frac{1}{2} \log \frac{\eta(K-\alpha)^2(t+1)}{\alpha^2 KM},
\end{aligned}
\tag{49}
$$

where the last step is by

$$\frac{1}{2} \log(1 + \frac{1}{t}) \leq \frac{1}{2t} \leq \frac{\eta}{KM} \cdot (\frac{K-\alpha}{\alpha})^2 e^{-\log \frac{\eta(K-\alpha)^2 t}{\alpha^2 KM}}. \tag{50}$$

To make $(1 - p_n(t))^2 < \epsilon$, we need

$$(\frac{K-\alpha}{\alpha})^2 e^{-2u} \leq \epsilon. \tag{51}$$

Then, we get

$$u \geq \frac{1}{2} \log \frac{1}{\epsilon} + \log \frac{K-\alpha}{\alpha}. \tag{52}$$

Therefore, by

$$\frac{1}{2} \log \frac{\eta t}{KM} + \log \frac{K-\alpha}{\alpha} \geq \frac{1}{2} \log \frac{1}{\epsilon} + \log \frac{K-\alpha}{\alpha}, \tag{53}$$

we finally obtain

$$t \geq T_3 := \eta^{-1} \epsilon^{-1} KM. \tag{54}$$

For $\tilde{\boldsymbol{p}}_i^n$ that shares the same TSR pattern as the query, we have that when $t = T_1$,

$$\tilde{\boldsymbol{p}}_i^{n\top} \boldsymbol{W}^{(t)} \tilde{\boldsymbol{p}}_{query}^n \geq \log \frac{1}{\epsilon}. \tag{55}$$

When $t = T_1 + T_2 + T_3$,

$$\tilde{\boldsymbol{p}}_i^{n\top} \boldsymbol{W}^{(t)} \tilde{\boldsymbol{p}}_{query}^n \geq \Theta(1) \cdot \log \frac{1}{\epsilon} = \Theta(\log \frac{1}{\epsilon}). \tag{56}$$

Then,

$$
\begin{aligned}
T &:= T_1 + T_2 + T_3 \\
&= \Theta(\eta^{-1} \alpha^{-2} K^3 \log \frac{K}{\epsilon} + \eta^{-1} MK(\alpha^{-1} + \epsilon^{-1})).
\end{aligned}
\tag{57}
$$

Therefore,

$$\mathbb{E}_{\boldsymbol{x}_{query} \sim \mathcal{D}, f \in \mathcal{T}} [\ell(\Psi; \boldsymbol{P}, \boldsymbol{z})] \leq \mathcal{O}(\epsilon). \tag{58}$$

$\square$

### E.2 PROOF OF THEOREM 2

*Proof.* We know that $\alpha'$ is the fraction of examples that share the same TSR pattern as the query. We need that in each step, the number of examples that share the same TSR pattern as the current step of the query is at least 1. Note that the probability of examples where each reasoning step produces the most probable output is

$$\prod_{k=1}^{K} A^f_{k(\text{TSR}(f_{k-1}\circ\cdots f_0(\boldsymbol{\mu}'_i)),\text{TSR}(f_k\circ\cdots f_0(\boldsymbol{\mu}'_i)))}, \text{ where } f_0(\boldsymbol{\mu}'_i) := \boldsymbol{\mu}'_i, \forall\, i \in [M'], \tag{59}$$

where the input to the first step has the TSR pattern $\boldsymbol{\mu}'_i$. Define $m_{k(i)}$ as the TSR pattern in the $k$-th step output of the $i$-th context example by the transition matrix defined in 11. Consider that the TSR pattern of the $k$-th step label of the testing query is $\boldsymbol{\mu}'_{q_k}$, which is also the most probable $k$-th step output of the $k$-th step of a certain $\boldsymbol{x}_i$ with $\text{TSR}(\boldsymbol{x}_i) = \text{TSR}(\boldsymbol{x}_{query}) = q_0$. Let the TSR pattern of another reasoning process, where for a certain first-step input $\boldsymbol{x}_i$ with $\text{TSR}(\boldsymbol{x}) = \text{TSR}(\boldsymbol{x}_{query}) = q_0$, the $k$-th step output is the most probable for $k \in [K']\backslash\{h\}$, while the $h$-th step output is the second probable. Denote the TSR pattern of the $k$-th step output of $\boldsymbol{x}_i$ following this process as $\boldsymbol{\mu}'_{u_k}$ with $u_0 = q_0$. By the Chernoff bound of Bernoulli distribution in Lemma 1, we can obtain

$$\Pr\left(\frac{1}{l_{ts}}\sum_{i=1}^{l_{ts}}\mathbb{1}[m_{k(i)} = \boldsymbol{\mu}'_{q_k}, \forall k \in [K']] \leq (1 - \rho_s^f/2)\alpha'\prod_{k=1}^{K'}A^f_{k(q_{k-1},q_k)}\right) \tag{60}$$

$$\leq e^{-l_{ts}(\rho_s^f)^2\alpha'\prod_{k=1}^{K'}A^f_{k(q_{k-1},q_k)}} = M^{-C},$$

and by Lemma 2,

$$\Pr\left(\frac{1}{l_{ts}}\sum_{i=1}^{l_{ts}}\mathbb{1}[m_{k(i)} = \boldsymbol{\mu}'_{u_k}, \forall k \in [K']] \geq (1 - \rho_s^f/2)\alpha'\prod_{k=1}^{K'}A^f_{k(q_{k-1},q_k)}\right)$$

$$\leq \Pr\left(\frac{1}{l_{ts}}\sum_{i=1}^{l_{ts}}\mathbb{1}[m_{k(i)} = \boldsymbol{\mu}'_{u_k}, \forall k \in [K']] \geq \alpha'\prod_{k=1}^{K'}A^f_{k(u_{k-1},u_k)} + t_0\right) \tag{61}$$

$$\leq e^{-l_{ts}t_0^2} = M^{-C},$$

for some $c \in (0,1)$ and $C > 0$, where the first step is by the definition of $\rho_s^f$ in (11), and

$$t_0 \lesssim \rho_s^f\alpha'\prod_{k=1}^{K'}A^f_{k(q_{k-1},q_k)}. \tag{62}$$

Hence, with a high probability,

$$l_{ts} \gtrsim \max\{(\rho_s^{f^2}\alpha'\prod_{k=1}^{K'}A^f_{k(q_{k-1},q_k)})^{-1}\log M, (\rho_s^f\alpha'\prod_{k=1}^{K'}A^f_{k(q_{k-1},q_k)})^{-2}\log M\}$$

$$\gtrsim (\rho_s^f\alpha'\prod_{k=1}^{K'}A^f_{k(q_{k-1},q_k)})^{-2}\log M, \tag{63}$$

such that the number of examples with the same TSR pattern as the query in each of the total $K$ steps is at least 1. To make the above condition hold for any TSR pattern of the intermediate step of the query, we need

$$l_{ts} \gtrsim \max_{q_k \in [M']}(\rho_s^f\alpha'\prod_{k=1}^{K'}A^f_{k(q_{k-1},q_k)})^{-2}\log M$$

$$= \max_{i \in [M']}(\rho_s^f\alpha'\prod_{k=1}^{K'}A^f_{k(\text{TSR}(f_{k-1}\circ\cdots f_0(\boldsymbol{\mu}'_i)),\text{TSR}(f_k\circ\cdots f_0(\boldsymbol{\mu}'_i)))})^{-2}\log M \tag{64}$$

$$= (\rho_s^f\alpha'\tau_s^f)^{-2}\log M.$$

Then, we show the CoT testing error is zero by induction. In the first step, consider $\boldsymbol{x}_i = \boldsymbol{\mu}_j + \boldsymbol{\delta}_i$ such that

$$\tilde{\boldsymbol{p}}_i = \begin{pmatrix} \boldsymbol{\mu}'_j \\ \boldsymbol{y}_{i,1} \end{pmatrix} + \begin{pmatrix} \boldsymbol{\delta}_i \\ \boldsymbol{0} \end{pmatrix} + \boldsymbol{c}_{i \mod K}. \tag{65}$$

Since that

$$(\boldsymbol{\delta}_i^\top, 0^\top) \boldsymbol{W}^{(0)} \tilde{\boldsymbol{p}}_i \lesssim \xi, \tag{66}$$

by that each entry of $\boldsymbol{W}^{(0)}$ follows $\mathcal{N}(0, \xi^2)$, and

$$(\boldsymbol{\delta}_i^\top, 0^\top) \frac{\eta}{B} \sum_{n \in \mathcal{B}_b} \sum_{b=0}^{T-1} \frac{\partial \ell(\Psi; \boldsymbol{P}^n, \boldsymbol{z}^n)}{\partial \boldsymbol{W}^{(b)}} \tilde{\boldsymbol{p}}_{query} = 0, \tag{67}$$

we have that for $\tilde{\boldsymbol{p}}_i$ that shares the same TSR pattern as the query,

$$\tilde{\boldsymbol{p}}_i^\top \boldsymbol{W}^{(T)} \tilde{\boldsymbol{p}}_{query}$$
$$= \tilde{\boldsymbol{p}}_i^\top (\boldsymbol{W}^{(0)} + \frac{\eta}{B} \sum_{n \in \mathcal{B}_b} \sum_{b=0}^{T-1} \frac{\partial \ell(\Psi; \boldsymbol{P}^n, \boldsymbol{z}^n)}{\partial \boldsymbol{W}^{(b)}}) \tilde{\boldsymbol{p}}_{query}$$
$$= ((\boldsymbol{\mu}'_j{}^\top, \boldsymbol{y}_{i,1}^\top) + \boldsymbol{c}_{i \mod K}^\top)(\boldsymbol{W}^{(0)} + \frac{\eta}{B} \sum_{n \in \mathcal{B}_b} \sum_{b=0}^{T-1} \frac{\partial \ell(\Psi; \boldsymbol{P}^n, \boldsymbol{z}^n)}{\partial \boldsymbol{W}^{(b)}}) \tilde{\boldsymbol{p}}_{query}. \tag{68}$$

Since that $\boldsymbol{\lambda}_j$ is orthogonal to all the $\boldsymbol{\mu}_i, i \in [M]$, we have similar conclusion for $\boldsymbol{\lambda}_j$ as $\boldsymbol{\delta}_i$, i.e.,

$$(\boldsymbol{\lambda}_j^\top, 0^\top) \boldsymbol{W}^{(0)} \tilde{\boldsymbol{p}}_i \lesssim \xi, \tag{69}$$

and

$$(\boldsymbol{\lambda}_j^\top, 0^\top) \frac{\eta}{B} \sum_{n \in \mathcal{B}_b} \sum_{b=0}^{T-1} \frac{\partial \ell(\Psi; \boldsymbol{P}^n, \boldsymbol{z}^n)}{\partial \boldsymbol{W}^{(b)}} \tilde{\boldsymbol{p}}_{query} = 0. \tag{70}$$

Let $\boldsymbol{\mu}'_j = \boldsymbol{\lambda}_j + \tilde{\boldsymbol{\mu}}_j = \boldsymbol{\lambda}_j + \sum_{i=1}^{M'} k_{j,i} \boldsymbol{\mu}_i$. Then, we have

$$\tilde{\boldsymbol{p}}_i^\top \boldsymbol{W}^{(T)} \tilde{\boldsymbol{p}}_{query}$$
$$= ((\boldsymbol{\lambda}_j^\top + \sum_{i=1}^{M'} k_{j,i} \boldsymbol{\mu}_i^\top, \boldsymbol{y}_{i,1}^\top) + \boldsymbol{c}_{i \mod K}^\top)(\boldsymbol{W}^{(0)} + \frac{\eta}{B} \sum_{n \in \mathcal{B}_b} \sum_{b=0}^{T-1} \frac{\partial \ell(\Psi; \boldsymbol{P}^n, \boldsymbol{z}^n)}{\partial \boldsymbol{W}^{(b)}})((\boldsymbol{\lambda}_j^\top$$
$$+ \sum_{i=1}^{M'} k_{j,i} \boldsymbol{\mu}_i^\top, \boldsymbol{0}^\top) + \boldsymbol{c}_1)^\top$$
$$= \sum_{i=1}^{M'} k_{j,i}^2 ((\boldsymbol{\mu}_i^\top, \boldsymbol{y}_{i,1}^\top) + \boldsymbol{c}_{i \mod K}^\top)(\boldsymbol{W}^{(0)} + \frac{\eta}{B} \sum_{n \in \mathcal{B}_b} \sum_{b=0}^{T-1} \frac{\partial \ell(\Psi; \boldsymbol{P}^n, \boldsymbol{z}^n)}{\partial \boldsymbol{W}^{(b)}})((\boldsymbol{\mu}_i^\top, \boldsymbol{0}^\top) + \boldsymbol{c}_1)^\top$$
$$+ \sum_{i \neq i'} k_{j,i} k_{j,i'} ((\boldsymbol{\mu}_i^\top, \boldsymbol{y}_{i,1}^\top) + \boldsymbol{c}_{i \mod K}^\top)(\boldsymbol{W}^{(0)} + \frac{\eta}{B} \sum_{n \in \mathcal{B}_b} \sum_{b=0}^{T-1} \frac{\partial \ell(\Psi; \boldsymbol{P}^n, \boldsymbol{z}^n)}{\partial \boldsymbol{W}^{(b)}})((\boldsymbol{\mu}_{i'}^\top, \boldsymbol{0}^\top) + \boldsymbol{c}_1)^\top$$
$$\geq C \cdot \Theta(\log \frac{1}{\epsilon}) - \Theta(\xi)$$
$$= \Theta(\log \frac{1}{\epsilon}), \tag{71}$$

where the second to last step is by Theorem 1. The last step holds if $C \geq \Theta(\log^{-1}(1/\epsilon))$. Since the gradient updates for different TRR patterns are very close to each other, we have that $\sum_{i \neq i'} |k_{j,i} k_{j,i'}| \leq 1$ and

$$\sum_{i \neq i'} k_{j,i} k_{j,i'} ((\boldsymbol{\mu}_i^\top, \boldsymbol{y}_{i,1}^\top) + \boldsymbol{c}_{i \mod K}^\top)(\boldsymbol{W}^{(0)} + \frac{\eta}{B} \sum_{n \in \mathcal{B}_b} \sum_{b=0}^{T-1} \frac{\partial \ell(\Psi; \boldsymbol{P}^n, \boldsymbol{z}^n)}{\partial \boldsymbol{W}^{(b)}})((\boldsymbol{\mu}_{i'}^\top, \boldsymbol{0}^\top) + \boldsymbol{c}_1)^\top$$
$$\lesssim \Theta(1) \cdot \frac{\tilde{\boldsymbol{p}}_s^\top \boldsymbol{W}^{(T)} \tilde{\boldsymbol{p}}_{query}}{\log \frac{1}{\epsilon}}, \tag{72}$$

where $\tilde{\boldsymbol{p}}_s$ shares the same TSR pattern and the same step as $\tilde{\boldsymbol{p}}_{query}$. Hence, for $\tilde{\boldsymbol{p}}_i$ that shares a different TSR pattern with $\tilde{\boldsymbol{p}}_{query}$,

$$\tilde{\boldsymbol{p}}_i^\top \boldsymbol{W}^{(T)} \tilde{\boldsymbol{p}}_{query} \lesssim \Theta(1). \tag{73}$$

Therefore, we can derive that

$$\sum_{i \in \mathcal{S}_1^*} \text{softmax}(\tilde{\boldsymbol{p}}_i^\top \boldsymbol{W}^{(T)} \tilde{\boldsymbol{p}}_{query}) \geq 1 - \epsilon, \tag{74}$$

where $\mathcal{S}_1^*$ is the set of the first step of examples that share the same TSR pattern as the query. Then, the first step leads to a correct prediction with zero testing error, since that $\max_{j \in [M']} A_{k(q_0,j)}$ is the largest to make the correct prediction for $\boldsymbol{x}_{query}$ if $\boldsymbol{x}_{query} = \boldsymbol{\mu}'_{q_0}$, i.e.,

$$\boldsymbol{v}_1 = f_1(\boldsymbol{\mu}'_{q_0}). \tag{75}$$

Suppose that the $k$-th step generates a zero testing error. Then, for the $k+1$-th step, we know that there exists $\boldsymbol{p}_j$ that shares the same TSR pattern as $\boldsymbol{v}_k$. Then, we can also derive that

$$\tilde{\boldsymbol{p}}_j^\top \boldsymbol{W}^{(T)}((\boldsymbol{v}_k^\top, \boldsymbol{0}^\top)^\top + \boldsymbol{c}_k^\top)^\top = \Theta(\log \frac{1}{\epsilon}), \tag{76}$$

and

$$\sum_{j \in \mathcal{S}_k^*} \text{softmax}(\tilde{\boldsymbol{p}}_j^\top \boldsymbol{W}^{(T)}((\boldsymbol{v}_{k-1}^\top \ \boldsymbol{v}_k^\top)^\top + \boldsymbol{c}_k^\top)^\top) \geq 1 - \epsilon. \tag{77}$$

Hence, the $k+1$-th also makes the correct prediction, i.e.,

$$\boldsymbol{v}_{k+1} = f_{k+1} \circ \cdots f_1(\boldsymbol{\mu}'_{q_0}), \tag{78}$$

where $\boldsymbol{\mu}'_{q_{k+1}}$ is the TSR pattern of the $k+1$-th step input. Therefore, we show that CoT makes the correct prediction in each step as well as in the final prediction, such that

$$\bar{R}^f_{CoT, \boldsymbol{x} \in \mathcal{M}', f \in \mathcal{T}'}(\Psi) = 0. \tag{79}$$

$\square$

### E.3 PROOF OF THEOREM 3

*Proof.* We know that the positional encodings are the same for the ICL inference in all examples. Hence, similar to (74), we can derive that

$$\sum_{i \in \mathcal{S}_K^*} \text{softmax}(\tilde{\boldsymbol{p}}_i^\top \boldsymbol{W}^{(T)} \tilde{\boldsymbol{p}}_{query}) \geq 1 - \epsilon, \tag{80}$$

where $\mathcal{S}_K^*$ is the set of the last step output of examples that share the same TSR pattern as the last step output of the query. For $\boldsymbol{x}_{query} = \boldsymbol{\mu}'_q, q \in [K']$, we know that the distribution of the corresponding label $\boldsymbol{y}$ of $\boldsymbol{x}$ with $\text{TSR}(\boldsymbol{x}) = q$ follows the $q$-th row the $K$-steps transition matrix $B^f$. Let $F(\Psi; \boldsymbol{P}) = \sum_{i=1}^{M'} \lambda_i^P \boldsymbol{\mu}'_i$. Hence, based on the output scheme of ICL as stated in Section 2.3, we have that

$$\boldsymbol{v} = \arg\min_{\boldsymbol{y} \in \mathcal{M}'} \frac{1}{2} \|F(\Psi; \boldsymbol{P}) - \boldsymbol{y}\|^2 = \boldsymbol{\mu}_{\arg\max_{i \in [M']} \lambda_i^P}. \tag{81}$$

Note that the probability of examples with the most probable final output with $\boldsymbol{\mu}'_q$ as the TSR pattern of the input is

$$B_{(q, \text{TSR}(f(\boldsymbol{\mu}'_q)))}. \tag{82}$$

To ensure that the number of examples with the same TSR pattern as the query that generates the most probable output is at least 1, we compute the following,

$$\Pr\left(\frac{1}{l_{ts}} \sum_{i=1}^{l_{ts}} \mathbb{1}[m_i = \boldsymbol{\mu}'_{q_1}] \leq (1 - \rho_o^f/2)\alpha' B_{(q, \text{TSR}(f(\boldsymbol{\mu}'_q)))}\right)$$
$$\leq e^{-l_{ts}\rho_o^{f2}\alpha' B_{(q, \text{TSR}(f(\boldsymbol{\mu}'_q)))}} = M^{-C}, \tag{83}$$

for some $c \in (0, 1)$ and $C > 0$ by the Chernoff bound of Bernoulli distribution in Lemma 1. Here, $m_i$ is defined as the TSR pattern in the final output of the $i$-th context example by the $K$-steps transition matrix defined in 12. The TSR pattern of the most probable output of the testing query is $\boldsymbol{\mu}'_{q_1}$. Similarly, let the TSR pattern of the second most probable output of the testing query be $\boldsymbol{\mu}'_{q_2}$. We also have

$$
\begin{aligned}
&\Pr\left(\frac{1}{l_{ts}}\sum_{i=1}^{l_{ts}}\mathbb{1}[m_i = \boldsymbol{\mu}'_{q_2}] \geq (1 - \rho_o^f/2)\alpha' B^f_{(q,q_1)}\right) \\
&\leq \Pr\left(\frac{1}{l_{ts}}\sum_{i=1}^{l_{ts}}\mathbb{1}[m_i = \boldsymbol{\mu}'_{q_2}] \geq \alpha' B_{(q,q_2)} + c \cdot \rho_o^f \alpha' B^f_{(q,q_1)}\right) \\
&\leq e^{-l_{ts}\rho_o^{f\,2}c^2\alpha' B_{(q,q_1)}} = M^{-C},
\end{aligned}
\tag{84}
$$

by Lemma 2 and (12) for some constant $c > 0$. Therefore, to make the number of examples with the same TSR pattern in the output as the label of the query be at least 1 for any TSR pattern of the query and the output be the most probable one, we need

$$
\begin{aligned}
l_{ts}^f &\gtrsim \max\{(\rho_o^{f\,2}\alpha' \min_{i \in [M']} B_{(i,\mathrm{TSR}(f(\boldsymbol{\mu}'_i)))})^{-1}\log M, (\rho_o^f\alpha' \min_{i \in [M']} B_{(i,\mathrm{TSR}(f(\boldsymbol{\mu}'_i)))})^{-2}\log M\} \\
&= (\rho_o^f\alpha'\tau_o^f)^{-2}\log M\}.
\end{aligned}
\tag{85}
$$

In addition, if Condition 1 holds such that the most probable output is the actual label, we can derive

$$
\bar{R}^f_{ICL, \boldsymbol{x} \in \mathcal{M}', f \in \mathcal{T}'}(\Psi) = 0.
\tag{86}
$$

When (85) holds but Condition 1 does not, we know that ICL still always produces the most probable output by the $K$-steps transition matrix, but such an output is not the label since Condition 1 fails. Hence,

$$
\bar{R}^f_{ICL, \boldsymbol{x} \in \mathcal{M}', f \in \mathcal{T}'}(\Psi) \geq \Omega(1).
\tag{87}
$$

When both Condition 1 and (85) do not hold, ICL can produce multiple possible outputs with a non-trivial probability, which is decided by the distribution of the prompt instead of the $K$-steps transition matrix. This can be seen from that (83) and (84) both do not hold since (85) fails. Then, ICL can produce both the most probable and the second most probable output with a constant probability. Let the TSR pattern of the $r$-th most probable output of the testing query be $\boldsymbol{\mu}'_r$. Recall that $F(\Psi; \boldsymbol{P}) = \sum_{i=1}^{M'}\lambda_i^{\boldsymbol{P}}\boldsymbol{\mu}'_i$, we then have that for some small $\epsilon > 0$,

$$
\lambda^{\boldsymbol{P}}_{r(q)} = \frac{|\{i \in [l^f_{ts}] : \boldsymbol{y}_i = \boldsymbol{\mu}'_r \text{ in } \boldsymbol{P}\}|}{l^f_{ts}} \pm \epsilon.
\tag{88}
$$

Then, by (81), the output of the query is $\boldsymbol{\mu}_{\arg\max_{r \in [M']}\lambda_r}$. Since that (85) does not hold, there exists at least a constant probability of the prompt $\boldsymbol{P}'$ with the same query as $\boldsymbol{P}$ such that

$$
\lambda^{\boldsymbol{P}'}_r = \frac{|\{i \in [l^f_{ts}] : \boldsymbol{y}_i = \boldsymbol{\mu}'_r \text{ in } \boldsymbol{P}'\}|}{l^f_{ts}} \pm \epsilon \neq \lambda^{\boldsymbol{P}}_r,
\tag{89}
$$

for some $r \in [M']$. Therefore, with a constant probability, the output for the same testing query and the same testing task $f$ varies. This leads to

$$
\bar{R}^f_{ICL, \boldsymbol{x} \in \mathcal{M}', f \in \mathcal{T}'}(\Psi) \geq \Omega(1).
\tag{90}
$$

$\square$

### E.4 PROOF OF PROPOSITION 1

*Proof.* This proposition is derived from the proof of Theorem 2. (22) comes from (77), while (23) comes from (78), both by induction. $\square$

# F   PROOF OF LEMMAS

## F.1   PROOF OF LEMMA 3

*Proof.*

$$
\eta \frac{1}{B} \sum_{n \in \mathcal{B}_b} \frac{\partial \ell(\Psi; \boldsymbol{P}^n, \boldsymbol{z}^n)}{\partial \boldsymbol{W}}
$$

$$
= \eta \frac{1}{B} \sum_{n \in \mathcal{B}_b} \frac{\partial \ell(\Psi; \boldsymbol{P}^n, \boldsymbol{z}^n)}{\partial F(\Psi; \boldsymbol{P})} \frac{\partial F(\Psi; \boldsymbol{P})}{\partial \boldsymbol{W}}
$$

$$
= \eta \frac{1}{B} \sum_{n \in \mathcal{B}_b} (F(\Psi; \boldsymbol{P}) - \boldsymbol{z}^n)^\top \sum_{i=1}^{l} \boldsymbol{W}_V \tilde{\boldsymbol{p}}_i \mathrm{softmax}(\tilde{\boldsymbol{p}}_i^\top \boldsymbol{W} \tilde{\boldsymbol{p}}_{query}) \tag{91}
$$

$$
\cdot (\tilde{\boldsymbol{p}}_i - \sum_{r=1}^{l} \mathrm{softmax}(\tilde{\boldsymbol{p}}_r^\top \boldsymbol{W} \tilde{\boldsymbol{p}}_i) \tilde{\boldsymbol{p}}_r) \tilde{\boldsymbol{p}}_{query}^\top.
$$

When $t = 0$, we know that each entry of $\boldsymbol{W}^{(0)}$ is generated from the Gaussian distribution $\mathcal{N}(0, \xi^2)$. Then,

$$
|\tilde{\boldsymbol{p}}_i^\top \boldsymbol{W}^{(0)} \tilde{\boldsymbol{p}}_{query}| = |\sum_{k,j} p_{i,k} p_{query,j} W_{k,j}^{(0)}| \lesssim \xi. \tag{92}
$$

Hence,

$$
\mathrm{softmax}(\tilde{\boldsymbol{p}}_i^\top \boldsymbol{W}^{(0)} \tilde{\boldsymbol{p}}_{query}) \geq \frac{e^{-\Theta(\xi)}}{l \cdot e^{\Theta(\xi)}} = \frac{1}{l} \cdot e^{-\Theta(\xi)}, \tag{93}
$$

$$
\mathrm{softmax}(\tilde{\boldsymbol{p}}_i^\top \boldsymbol{W}^{(0)} \tilde{\boldsymbol{p}}_{query}) \leq \frac{e^{-\Theta(\xi)}}{l \cdot e^{\Theta(\xi)}} = \frac{1}{l} \cdot e^{-\Theta(\xi)}. \tag{94}
$$

We can obtain

$$
F(\Psi; \boldsymbol{P}) = \sum_{i=1}^{l} \frac{e^{-\Theta(\xi)}}{l} \boldsymbol{W}_V \boldsymbol{p}_i. \tag{95}
$$

Since that $\mathrm{PE}(\cdot)$, and $\mathrm{TRR}(\cdot)$ denote the positional encoding, and the TSR pattern of the input, respectively, we have that for $\boldsymbol{p}$,

$$
\tilde{\boldsymbol{p}}^\top \tilde{\boldsymbol{p}}_{query} = \mathbb{1}[\mathrm{TRR}(\tilde{\boldsymbol{p}}) = \mathrm{TRR}(\tilde{\boldsymbol{p}}_{query})] + \mathbb{1}[\mathrm{PE}(\tilde{\boldsymbol{p}}) = \mathrm{PE}(\tilde{\tilde{\boldsymbol{p}}}_i)]. \tag{96}
$$

Given $\mathrm{lab}(\cdot)$ is the label embedding of the context as the input, we have that for $\boldsymbol{p}$,

$$
\tilde{\boldsymbol{p}}^\top \tilde{\boldsymbol{p}}_i = \mathbb{1}[\mathrm{TRR}(\tilde{\boldsymbol{p}}) = \mathrm{TRR}(\tilde{\boldsymbol{p}}_i)] + \mathbb{1}[\mathrm{lab}(\tilde{\boldsymbol{p}}) = \mathrm{lab}(\tilde{\boldsymbol{p}}_i)] + \mathbb{1}[\mathrm{PE}(\tilde{\boldsymbol{p}}) = \mathrm{PE}(\tilde{\boldsymbol{p}}_i)], \tag{97}
$$

$$
(\boldsymbol{W}_V \tilde{\boldsymbol{p}})^\top \boldsymbol{W}_V \tilde{\boldsymbol{p}}_i = \mathbb{1}[\mathrm{lab}(\tilde{\boldsymbol{p}}) = \mathrm{lab}(\tilde{\boldsymbol{p}}_i)]. \tag{98}
$$

When $t \geq 1$, we first consider the case where $\tilde{\boldsymbol{p}}$ shares the same TRR pattern and the positional encoding as $\tilde{\boldsymbol{p}}_{query}$. If $\tilde{\boldsymbol{p}}$ and $\tilde{\boldsymbol{p}}_{query}$ share the same TRR pattern, label pattern, and the positional encoding,

$$
\tilde{\boldsymbol{p}}^\top (\tilde{\boldsymbol{p}}_i - \sum_{r=1}^{l} \mathrm{softmax}(\tilde{\boldsymbol{p}}_r^\top \boldsymbol{W} \tilde{\boldsymbol{p}}_{query}) \tilde{\boldsymbol{p}}_r) \tilde{\boldsymbol{p}}_{query}^\top \tilde{\boldsymbol{p}} \geq 2 \cdot (3 - 3p_n(t) - (1 - p_n(t)))
$$
$$
= 4(1 - p_n(t)), \tag{99}
$$

and

$$
\tilde{\boldsymbol{p}}^\top (\tilde{\boldsymbol{p}}_i - \sum_{r=1}^{l} \mathrm{softmax}(\tilde{\boldsymbol{p}}_r^\top \boldsymbol{W} \tilde{\boldsymbol{p}}_{query}) \tilde{\boldsymbol{p}}_r) \tilde{\boldsymbol{p}}_{query}^\top \tilde{\boldsymbol{p}} \leq 2 \cdot (3 - 3p_n(t)) = 6(1 - p_n(t)). \tag{100}
$$

When $\tilde{\boldsymbol{p}}$ and $\tilde{\boldsymbol{p}}_{query}$ only share the same positional encoding or the same TRR pattern,

$$
2 - 6p_n(t) \geq \tilde{\boldsymbol{p}}^\top (\tilde{\boldsymbol{p}}_i - \sum_{r=1}^{l} \mathrm{softmax}(\tilde{\boldsymbol{p}}_r^\top \boldsymbol{W} \tilde{\boldsymbol{p}}_{query}) \tilde{\boldsymbol{p}}_r) \tilde{\boldsymbol{p}}_{query}^\top \tilde{\boldsymbol{p}} \geq -4p_n(t). \tag{101}
$$

When $\tilde{\boldsymbol{p}}$ and $\tilde{\boldsymbol{p}}_{query}$ share both different positional encodings and TRR patterns,

$$-6p_n(t) \geq \tilde{\boldsymbol{p}}^\top(\tilde{\boldsymbol{p}}_i - \sum_{r=1}^{l} \text{softmax}(\tilde{\boldsymbol{p}_r}^\top \boldsymbol{W}\tilde{\boldsymbol{p}}_{query})\tilde{\boldsymbol{p}}_r)\boldsymbol{p}_{q\tilde{e}ury}^\top\tilde{\boldsymbol{p}} \geq -2 - 4p_n(t). \tag{102}$$

Then, we consider the case where $\tilde{\boldsymbol{p}}$ only shares the same TRR pattern or the same positional encoding as $\tilde{\boldsymbol{p}}_i$. If $\tilde{\boldsymbol{p}}$ and $\tilde{\boldsymbol{p}}_{query}$ share the same TRR pattern, label pattern, and the positional encoding,

$$3 - p_n(t) \geq \tilde{\boldsymbol{p}}^\top(\tilde{\boldsymbol{p}}_i - \sum_{r=1}^{l} \text{softmax}(\tilde{\boldsymbol{p}_r}^\top \boldsymbol{W}\tilde{\boldsymbol{p}}_{query})\tilde{\boldsymbol{p}}_r)\tilde{\boldsymbol{p}}_{query}^\top\tilde{\boldsymbol{p}} \geq 1 \cdot (3 - p_n(t) - (1 - p_n(t)))$$
$$= 2. \tag{103}$$

When $\tilde{\boldsymbol{p}}$ and $\tilde{\boldsymbol{p}}_{query}$ only share the same positional encoding or the same TRR pattern,

$$1 - p_n(t) \geq \tilde{\boldsymbol{p}}^\top(\tilde{\boldsymbol{p}}_i - \sum_{r=1}^{l} \text{softmax}(\tilde{\boldsymbol{p}_r}^\top \boldsymbol{W}\tilde{\boldsymbol{p}}_{query})\tilde{\boldsymbol{p}}_r)\tilde{\boldsymbol{p}}_{query}^\top\tilde{\boldsymbol{p}} \geq 0. \tag{104}$$

When $\tilde{\boldsymbol{p}}$ and $\tilde{\boldsymbol{p}}_{query}$ only share both different positional encodings and TRR patterns,

$$-p_n(t) \geq \tilde{\boldsymbol{p}}^\top(\tilde{\boldsymbol{p}}_i - \sum_{r=1}^{l} \text{softmax}(\tilde{\boldsymbol{p}_r}^\top \boldsymbol{W}\tilde{\boldsymbol{p}}_{query})\tilde{\boldsymbol{p}}_r)\tilde{\boldsymbol{p}}_{query}^\top\tilde{\boldsymbol{p}} \geq -1. \tag{105}$$

Note that $-(1 - p_n(t))p_n(t) + (1 - p_n(t))^2 \alpha^2/K^2 < 0$ for $p_n(t) \in [\alpha/K, \alpha]$. Then, when $l \geq \Omega(\alpha^{-1})$ and $\tilde{\boldsymbol{p}}$ shares the same TRR pattern and the positional encoding as $\tilde{\boldsymbol{p}}_i$,

$$(\sum_{i=1}^{l} \text{softmax}(\tilde{\boldsymbol{p}_i}^\top \boldsymbol{W}\tilde{\boldsymbol{p}}_{query})\boldsymbol{W}_V\tilde{\boldsymbol{p}}_i - \boldsymbol{z}^n)^\top \sum_{i=1}^{l} \text{softmax}(\tilde{\boldsymbol{p}_i}^\top \boldsymbol{W}\tilde{\boldsymbol{p}}_{query})\boldsymbol{W}_V\tilde{\boldsymbol{p}}_i$$

$$\cdot \tilde{\boldsymbol{p}}^\top(\tilde{\boldsymbol{p}}_i - \sum_{r=1}^{l} \text{softmax}(\tilde{\boldsymbol{p}_r}^\top \boldsymbol{W}\tilde{\boldsymbol{p}}_{query})\tilde{\boldsymbol{p}}_r)\tilde{\boldsymbol{p}}_{query}^\top\tilde{\boldsymbol{p}}$$

$$\leq -4p_n(t)(1 - p_n(t))^2 - 4p_n(t)(1 - p_n(t))^2 \cdot \frac{\alpha^2}{K^2}$$

$$+ \frac{1}{l}(\frac{1}{K} - \frac{\alpha}{K})(-4p_n(t)) + \frac{1}{l}(\frac{1}{K} - \frac{\alpha}{K})(1 - p_n(t))(-2 - 4p_n(t))(K - 1)$$

$$= -4p_n(t)(1 - p_n(t))^2(1 + \frac{\alpha^2}{K^2}) + \frac{2}{lK}(1 - \alpha)(-(K - 1) - (K + 1)p_n(t) + 2p_n(t)^2(K - 1)). \tag{106}$$

We next consider the case where $\tilde{\boldsymbol{p}}$ shares the same TRR pattern and the different positional encoding as $\tilde{\boldsymbol{p}}_{query}$. Note that

$$\frac{2}{Kl} \cdot (1 - \alpha) \cdot K(1 - p_n(t)) \lesssim |(-(1 - p_n(t))p_n(t) + (1 - p_n(t))^2\frac{\alpha^2}{K^2})(1 - p_n(t))|, \tag{107}$$

if $l \geq \Omega(\alpha^{-1})$. Then,

$$(\sum_{i=1}^{l} \text{softmax}(\tilde{\boldsymbol{p}_i}^\top \boldsymbol{W}\tilde{\boldsymbol{p}}_{query})\boldsymbol{W}_V\tilde{\boldsymbol{p}}_i - \boldsymbol{z}^n)^\top \sum_{i=1}^{l} \text{softmax}(\tilde{\boldsymbol{p}_i}^\top \boldsymbol{W}\tilde{\boldsymbol{p}}_{query})\boldsymbol{W}_V\tilde{\boldsymbol{p}}_i$$

$$\cdot \tilde{\boldsymbol{p}}^\top(\tilde{\boldsymbol{p}}_i - \sum_{r=1}^{l} \text{softmax}(\tilde{\boldsymbol{p}_r}^\top \boldsymbol{W}\tilde{\boldsymbol{p}}_{query})\tilde{\boldsymbol{p}}_r)\tilde{\boldsymbol{p}}_{query}^\top\tilde{\boldsymbol{p}} \tag{108}$$

$$\leq -0 \cdot p_n(t)(1 - p_n(t)) + (1 - p_n(t))^2\frac{\alpha^2}{K^2} \cdot (+2) + \frac{1}{l}(\frac{1}{K} - \frac{\alpha}{K})(-(K - 1))$$

$$= 2(1 - p_n(t))^2\frac{\alpha^2}{K^2} - \frac{K - 1}{l}(\frac{1}{K} - \frac{\alpha}{K}).$$

We next consider the case where $\tilde{\boldsymbol{p}}$ shares the same positional encoding and the different TRR pattern as $\tilde{\boldsymbol{p}}_{query}$. Then,

$$
(\sum_{i=1}^{l} \mathrm{softmax}(\tilde{\boldsymbol{p}}_i{}^\top \boldsymbol{W}\tilde{\boldsymbol{p}}_{query})\boldsymbol{W}_V\tilde{\boldsymbol{p}}_i - \boldsymbol{z}^n)^\top \sum_{i=1}^{l} \mathrm{softmax}(\tilde{\boldsymbol{p}}_i{}^\top \boldsymbol{W}\tilde{\boldsymbol{p}}_{query})\boldsymbol{W}_V\tilde{\boldsymbol{p}}_i
$$

$$
\cdot \tilde{\boldsymbol{p}}^\top (\tilde{\boldsymbol{p}}_i - \sum_{r=1}^{l} \mathrm{softmax}(\tilde{\boldsymbol{p}}_r{}^\top \boldsymbol{W}\tilde{\boldsymbol{p}}_{query})\tilde{\boldsymbol{p}}_r)\tilde{\boldsymbol{p}}_i{}^\top \tilde{\boldsymbol{p}} \tag{109}
$$

$$
\leq 0 - (1 - p_n(t))^2 \frac{\alpha^2}{K^2} + \frac{1}{l}(\frac{1}{K} - \frac{\alpha}{K})(-(K-1))
$$

$$
= -(1 - p_n(t))^2 \frac{\alpha^2}{K^2} - \frac{K-1}{l}(\frac{1}{K} - \frac{\alpha}{K}).
$$

Therefore, as long as

$$
l \geq \Omega(\alpha^{-1}), \tag{110}
$$

we have

$$
\tilde{\boldsymbol{p}}^\top \eta \frac{1}{B}\sum_{n\in\mathcal{B}_b} \frac{\partial \ell(\Psi; \boldsymbol{P}^n, \boldsymbol{z}^n)}{\partial \boldsymbol{W}}\boldsymbol{p}
$$

$$
= \eta \frac{1}{B}\sum_{n\in\mathcal{B}_b} (F(\Psi; \boldsymbol{P}) - \boldsymbol{z}^n)^\top \sum_{i=1}^{l} \boldsymbol{W}_V\tilde{\boldsymbol{p}}_i \mathrm{softmax}(\tilde{\boldsymbol{p}}_i{}^\top \boldsymbol{W}\tilde{\boldsymbol{p}}_{query})
$$

$$
\cdot \tilde{\boldsymbol{p}}^\top (\tilde{\boldsymbol{p}}_i - \sum_{r=1}^{l} \mathrm{softmax}(\tilde{\boldsymbol{p}}_r{}^\top \boldsymbol{W}\tilde{\boldsymbol{p}}_{query})\tilde{\boldsymbol{p}}_r)\tilde{\boldsymbol{p}}_i{}^\top \tilde{\boldsymbol{p}}
$$

$$
\leq \eta \frac{1}{B}\sum_{n\in\mathcal{B}_b} (\frac{1}{KM}(1 - p_n(t))^2(-4p_n(t)(1 + \frac{\alpha^2}{K^2}) + \frac{2(K-1)\alpha^2}{K^2})
$$

$$
\cdot + (\frac{1}{K} - \frac{1}{M})(-(1 - p_n(t))^2 \frac{\alpha^2}{K^2}))
$$

$$
= \eta \cdot \frac{1}{B}\sum_{n\in\mathcal{B}_b} (\frac{1}{KM}(1 - p_n(t))^2(-4p_n(t)(1 + \frac{\alpha^2}{K^2}) + \frac{\alpha^2}{K}(1 + \frac{2(K-1)}{K})) - \frac{\alpha^2}{K^3}(1 - p_n(t))^2. \tag{111}
$$

We then consider the case where $\tilde{\boldsymbol{p}}'$ shares a different positional encoding and the same TRR pattern as $\tilde{\boldsymbol{p}}$. Let $\tilde{\boldsymbol{p}}$ share the same TRR pattern and the positional encoding as $\tilde{\boldsymbol{p}}_{query}$. If $\tilde{\boldsymbol{p}}'$ and $\tilde{\boldsymbol{p}}_i$ share the same TRR pattern, label pattern, and the positional encoding,

$$
2(3 - p_n(t)) \geq \tilde{\boldsymbol{p}}'{}^\top (\tilde{\boldsymbol{p}}_i - \sum_{r=1}^{l} \mathrm{softmax}(\tilde{\boldsymbol{p}}_r{}^\top \boldsymbol{W}\tilde{\boldsymbol{p}}_{query})\tilde{\boldsymbol{p}}_r)\tilde{\boldsymbol{p}}_{query}^\top \tilde{\boldsymbol{p}} \geq 2 \cdot (3 - p_n(t) - (1 - p_n(t)))
$$

$$
= 4. \tag{112}
$$

When $\tilde{\boldsymbol{p}}'$ and $\tilde{\boldsymbol{p}}_{query}$ only share the same positional encoding or the same TRR pattern,

$$
2(1 - p_n(t)) \geq \tilde{\boldsymbol{p}}'{}^\top (\tilde{\boldsymbol{p}}_i - \sum_{r=1}^{l} \mathrm{softmax}(\tilde{\boldsymbol{p}}_r{}^\top \boldsymbol{W}\tilde{\boldsymbol{p}}_{query})\tilde{\boldsymbol{p}}_r)\tilde{\boldsymbol{p}}_{query}^\top \tilde{\boldsymbol{p}} \geq 0. \tag{113}
$$

When $\tilde{\boldsymbol{p}}'$ and $\tilde{\boldsymbol{p}}_i$ only share both different positional encodings and TRR patterns,

$$
-2p_n(t) \geq \tilde{\boldsymbol{p}}'{}^\top (\tilde{\boldsymbol{p}}_i - \sum_{r=1}^{l} \mathrm{softmax}(\tilde{\boldsymbol{p}}_r{}^\top \boldsymbol{W}\tilde{\boldsymbol{p}}_{query})\tilde{\boldsymbol{p}}_r)\tilde{\boldsymbol{p}}_{query}^\top \tilde{\boldsymbol{p}} \geq -2. \tag{114}
$$

Then, we consider the case where $\tilde{\boldsymbol{p}}$ only shares the same TRR pattern as $\tilde{\boldsymbol{p}}_{query}$. If $\tilde{\boldsymbol{p}}'$ and $\tilde{\boldsymbol{p}}_i$ share the same TRR pattern, label pattern, and the positional encoding,

$$
3 - p_n(t)) \geq \tilde{\boldsymbol{p}}'{}^\top (\tilde{\boldsymbol{p}}_i - \sum_{r=1}^{l} \mathrm{softmax}(\tilde{\boldsymbol{p}}_r{}^\top \boldsymbol{W}\tilde{\boldsymbol{p}}_{query})\tilde{\boldsymbol{p}}_r)\tilde{\boldsymbol{p}}_{query}^\top \tilde{\boldsymbol{p}}
$$

$$
\geq 1 \cdot (3 - 3p_n(t) - (1 - p_n(t))) = 2(1 - p_n(t)). \tag{115}
$$

When $\tilde{\boldsymbol{p}}'$ and $\tilde{\boldsymbol{p}}_i$ only share the same positional encoding or the same TRR pattern,

$$1 - p_n(t) \geq \tilde{\boldsymbol{p}}'^\top (\tilde{\boldsymbol{p}}_i - \sum_{r=1}^l \text{softmax}(\tilde{\boldsymbol{p}_r}^\top \boldsymbol{W} \tilde{\boldsymbol{p}}_{query}) \tilde{\boldsymbol{p}_r}) \tilde{\boldsymbol{p}}_{query}^\top \tilde{\boldsymbol{p}} \geq -2p_n(t). \tag{116}$$

When $\tilde{\boldsymbol{p}}'$ and $\tilde{\boldsymbol{p}}_i$ only share both different positional encodings and TRR patterns,

$$-p_n(t) \geq \tilde{\boldsymbol{p}}'^\top (\tilde{\boldsymbol{p}}_i - \sum_{r=1}^l \text{softmax}(\tilde{\boldsymbol{p}_r}^\top \boldsymbol{W} \tilde{\boldsymbol{p}}_{query}) \tilde{\boldsymbol{p}_r}) \tilde{\boldsymbol{p}}_{query}^\top \tilde{\boldsymbol{p}} \geq -1 - 2p_n(t). \tag{117}$$

Next, we consider the case where $\tilde{\boldsymbol{p}}$ only shares the same positional encoding as $\tilde{\boldsymbol{p}}_{query}$. If $\tilde{\boldsymbol{p}}'$ and $\tilde{\boldsymbol{p}}_i$ share the same TRR pattern, label pattern, and the positional encoding,

$$3 \geq \tilde{\boldsymbol{p}}'^\top (\tilde{\boldsymbol{p}}_i - \sum_{r=1}^l \text{softmax}(\tilde{\boldsymbol{p}_r}^\top \boldsymbol{W} \tilde{\boldsymbol{p}}_{query}) \tilde{\boldsymbol{p}_r}) \tilde{\boldsymbol{p}}_{query}^\top \tilde{\boldsymbol{p}} \geq 1 \cdot (3 - (1 - p_n(t))) = 2 + p_n(t). \tag{118}$$

When $\tilde{\boldsymbol{p}}'$ and $\tilde{\boldsymbol{p}}_i$ only share the same positional encoding or the same TRR pattern,

$$1 \geq \tilde{\boldsymbol{p}}'^\top (\tilde{\boldsymbol{p}}_i - \sum_{r=1}^l \text{softmax}(\tilde{\boldsymbol{p}_r}^\top \boldsymbol{W} \tilde{\boldsymbol{p}}_{query}) \tilde{\boldsymbol{p}_r}) \tilde{\boldsymbol{p}}_{query}^\top \tilde{\boldsymbol{p}} \geq p_n(t). \tag{119}$$

When $\tilde{\boldsymbol{p}}'$ and $\tilde{\boldsymbol{p}}_i$ only share both different positional encodings and TRR patterns,

$$0 \geq \tilde{\boldsymbol{p}}'^\top (\tilde{\boldsymbol{p}}_i - \sum_{r=1}^l \text{softmax}(\tilde{\boldsymbol{p}_r}^\top \boldsymbol{W} \tilde{\boldsymbol{p}}_{query}) \tilde{\boldsymbol{p}_r}) \tilde{\boldsymbol{p}}_{query}^\top \tilde{\boldsymbol{p}} \geq -1 + p_n(t). \tag{120}$$

Then, when $l \geq \Omega(\alpha^{-1})$ and $\tilde{\boldsymbol{p}}$ shares the same TRR pattern and the positional encoding as $\tilde{\boldsymbol{p}}_{query}$,

$$\begin{aligned}
&(\sum_{i=1}^l \text{softmax}(\tilde{\boldsymbol{p}}_i^\top \boldsymbol{W} \tilde{\boldsymbol{p}}_{query}) \boldsymbol{W}_V \tilde{\boldsymbol{p}}_i - \boldsymbol{z}^n)^\top \sum_{i=1}^l \text{softmax}(\tilde{\boldsymbol{p}}_i^\top \boldsymbol{W} \tilde{\boldsymbol{p}}_{query}) \boldsymbol{W}_V \tilde{\boldsymbol{p}}_i \\
&\cdot \tilde{\boldsymbol{p}}'^\top (\tilde{\boldsymbol{p}}_i - \sum_{r=1}^l \text{softmax}(\tilde{\boldsymbol{p}_r}^\top \boldsymbol{W} \tilde{\boldsymbol{p}}_{query}) \tilde{\boldsymbol{p}_r}) \tilde{\boldsymbol{p}}_{query}^\top \tilde{\boldsymbol{p}} \\
&\leq -4p_n(t)(1 - p_n(t)) + \frac{1}{l}(\frac{1}{K} - \frac{\alpha}{K})(-2K).
\end{aligned} \tag{121}$$

We next consider the case where $\tilde{\boldsymbol{p}}$ shares the same TRR pattern and the different positional encoding as $\tilde{\boldsymbol{p}}_{query}$. Then, by (107),

$$\begin{aligned}
&(\sum_{i=1}^l \text{softmax}(\tilde{\boldsymbol{p}}_i^\top \boldsymbol{W} \tilde{\boldsymbol{p}}_{query}) \boldsymbol{W}_V \tilde{\boldsymbol{p}}_i - \boldsymbol{z}^n)^\top \sum_{i=1}^l \text{softmax}(\tilde{\boldsymbol{p}}_i^\top \boldsymbol{W} \tilde{\boldsymbol{p}}_{query}) \boldsymbol{W}_V \tilde{\boldsymbol{p}}_i \\
&\cdot \tilde{\boldsymbol{p}}'^\top (\tilde{\boldsymbol{p}}_i - \sum_{r=1}^l \text{softmax}(\tilde{\boldsymbol{p}_r}^\top \boldsymbol{W} \tilde{\boldsymbol{p}}_{query}) \tilde{\boldsymbol{p}_r}) \tilde{\boldsymbol{p}}_{query}^\top \tilde{\boldsymbol{p}} \\
&\leq -2p_n(t)(1 - p_n(t))^2 - 2p_n(t)(1 - p_n(t))^2 \cdot \frac{\alpha^2}{K^2} + \frac{1}{l}(\frac{1}{K} - \frac{\alpha}{K})((-1 - 2p_n(t))K) \\
&= -2p_n(t)(1 - p_n(t))^2(1 + \frac{\alpha^2}{K^2}) + \frac{1}{l}(1 - \alpha)(-1 - 2p_n(t)).
\end{aligned} \tag{122}$$

We next consider the case where $\tilde{\boldsymbol{p}}$ shares the same positional encoding and the different TRR pattern as $\tilde{\boldsymbol{p}}_{query}$. Then,

$$\begin{aligned}
&(\sum_{i=1}^l \text{softmax}(\tilde{\boldsymbol{p}}_i^\top \boldsymbol{W} \tilde{\boldsymbol{p}}_{query}) \boldsymbol{W}_V \tilde{\boldsymbol{p}}_i - \boldsymbol{z}^n)^\top \sum_{i=1}^l \text{softmax}(\tilde{\boldsymbol{p}}_i^\top \boldsymbol{W} \tilde{\boldsymbol{p}}_{query}) \boldsymbol{W}_V \tilde{\boldsymbol{p}}_i \\
&\cdot \tilde{\boldsymbol{p}}'^\top (\tilde{\boldsymbol{p}}_i - \sum_{r=1}^l \text{softmax}(\tilde{\boldsymbol{p}_r}^\top \boldsymbol{W} \tilde{\boldsymbol{p}}_{query}) \tilde{\boldsymbol{p}_r}) \tilde{\boldsymbol{p}}_{query}^\top \tilde{\boldsymbol{p}} \\
&\leq p_n(t)(1 - p_n(t))^2 + p_n(t)(1 - p_n(t))^2 \frac{\alpha^2}{K^2} + \frac{1}{l}(1 - \alpha)(-1 - 2p_n(t)) \\
&= p_n(t)(1 - p_n(t))^2(1 + \frac{\alpha^2}{K^2}) - \frac{1}{l}(1 - \alpha)(1 + 2p_n(t)).
\end{aligned} \tag{123}$$

Therefore, as long as

$$l \geq \Omega(\alpha^{-1}), \tag{124}$$

we have

$$\tilde{\boldsymbol{p}}'^\top \eta \frac{1}{B} \sum_{n \in \mathcal{B}_b} \frac{\partial \ell(\Psi; \boldsymbol{P}^n, \boldsymbol{z}^n)}{\partial \boldsymbol{W}} \tilde{\boldsymbol{p}}$$

$$= \eta \frac{1}{B} \sum_{n \in \mathcal{B}_b} (F(\Psi; \boldsymbol{P}) - \boldsymbol{z}^n)^\top \sum_{i=1}^{l} \boldsymbol{W}_V \tilde{\boldsymbol{p}}_i \mathrm{softmax}(\tilde{\boldsymbol{p}_i}^\top \boldsymbol{W} \tilde{\boldsymbol{p}}_{query}) \tilde{\boldsymbol{p}}^\top (\tilde{\boldsymbol{p}}_i$$

$$- \sum_{r=1}^{l} \mathrm{softmax}(\tilde{\boldsymbol{p}_r}^\top \boldsymbol{W} \tilde{\boldsymbol{p}}_{query}) \tilde{\boldsymbol{p}}_r) \tilde{\boldsymbol{p}}_{query}^\top \tilde{\boldsymbol{p}}$$

$$\leq \eta \frac{1}{B} \sum_{n \in \mathcal{B}_b} (\frac{1}{KM}(-4 - 2(K-1)(1-p_n(t))(1+\frac{\alpha^2}{K^2}))p_n(t)(1-p_n(t))$$

$$+ (\frac{1}{K} - \frac{1}{M})p_n(t)(1-p_n(t))^2(1+\frac{\alpha^2}{K^2}))$$

$$= \eta \cdot \frac{1}{B} \sum_{n \in \mathcal{B}_b} (\frac{1}{KM}(-4 - (3K-2)(1-p_n(t))(1+\frac{\alpha^2}{K^2}))p_n(t)(1-p_n(t))$$

$$+ \frac{1}{K} p_n(t)(1-p_n(t))^2(1+\frac{\alpha^2}{K^2})), \tag{125}$$

and

$$\tilde{\boldsymbol{p}}'^\top \eta \frac{1}{B} \sum_{n \in \mathcal{B}_b} \frac{\partial \ell(\Psi; \boldsymbol{P}^n, \boldsymbol{z}^n)}{\partial \boldsymbol{W}} \tilde{\boldsymbol{p}}$$

$$\geq \eta \frac{1}{B} \sum_{n \in \mathcal{B}_b} (\frac{1}{KM}(-4 - (3K-2)(1-p_n(t))(1+\frac{\alpha^2}{K^2}))p_n(t)(1-p_n(t))$$

$$+ \frac{1}{K} p_n(t)(1-p_n(t))^2(1+\frac{\alpha^2}{K^2}) + \frac{1}{K} \cdot (1-p_n(t))^2(-p_n(t) + (1-p_n(t))\frac{\alpha^2}{K^2}))$$

$$= \eta \cdot \frac{1}{B} \sum_{n \in \mathcal{B}_b} (\frac{1}{KM}(-4 - (3K-2)(1-p_n(t))(1+\frac{\alpha^2}{K^2}))p_n(t)(1-p_n(t)) + \frac{\alpha^2}{K^3}(1-p_n(t))^2).$$
$$\tag{126}$$

We next consider the case where $\tilde{\boldsymbol{p}}'$ shares a different TRR pattern and the same positional encoding as $\tilde{\boldsymbol{p}}$. Let $\tilde{\boldsymbol{p}}$ share the same TRR pattern and the positional encoding as $\tilde{\boldsymbol{p}}_{query}$. If $\tilde{\boldsymbol{p}}'$ and $\tilde{\boldsymbol{p}}_i$ share the same TRR pattern, label pattern, and positional encoding,

$$2(3 - p_n(t)) \geq \tilde{\boldsymbol{p}}'^\top (\tilde{\boldsymbol{p}}_i - \sum_{r=1}^{l} \mathrm{softmax}(\tilde{\boldsymbol{p}_r}^\top \boldsymbol{W} \tilde{\boldsymbol{p}}_{query}) \tilde{\boldsymbol{p}}_r) \tilde{\boldsymbol{p}}_{query}^\top \tilde{\boldsymbol{p}} \geq 2 \cdot (3 - p_n(t) - (1 - p_n(t)))$$

$$= 4. \tag{127}$$

When $\tilde{\boldsymbol{p}}'$ and $\tilde{\boldsymbol{p}}_i$ only share the same positional encoding or the same TRR pattern,

$$2(1 - p_n(t)) \geq \tilde{\boldsymbol{p}}'^\top (\tilde{\boldsymbol{p}}_i - \sum_{r=1}^{l} \mathrm{softmax}(\tilde{\boldsymbol{p}_r}^\top \boldsymbol{W} \tilde{\boldsymbol{p}}_{query}) \tilde{\boldsymbol{p}}_r) \tilde{\boldsymbol{p}}_{query}^\top \tilde{\boldsymbol{p}} \geq 0. \tag{128}$$

When $\tilde{\boldsymbol{p}}'$ and $\tilde{\boldsymbol{p}}_i$ only share both different positional encodings and TRR patterns,

$$-2p_n(t) \geq \tilde{\boldsymbol{p}}'^\top (\tilde{\boldsymbol{p}}_i - \sum_{r=1}^{l} \mathrm{softmax}(\tilde{\boldsymbol{p}_r}^\top \boldsymbol{W} \tilde{\boldsymbol{p}}_{query}) \tilde{\boldsymbol{p}}_r) \tilde{\boldsymbol{p}}_{query}^\top \tilde{\boldsymbol{p}} \geq -2. \tag{129}$$

Then, we consider the case where $\tilde{\boldsymbol{p}}$ only shares the same TRR pattern as $\tilde{\boldsymbol{p}}_{query}$. If $\tilde{\boldsymbol{p}}'$ and $\tilde{\boldsymbol{p}}_i$ share the same TRR pattern, label pattern, and the positional encoding,

$$3 \geq \tilde{\boldsymbol{p}}'^\top (\tilde{\boldsymbol{p}}_i - \sum_{r=1}^{l} \mathrm{softmax}(\tilde{\boldsymbol{p}_r}^\top \boldsymbol{W} \tilde{\boldsymbol{p}}_{query}) \tilde{\boldsymbol{p}}_r) \tilde{\boldsymbol{p}}_{query}^\top \tilde{\boldsymbol{p}} \geq 1 \cdot (3 - (1 - p_n(t))) = 2 + p_n(t). \tag{130}$$

When $\tilde{\boldsymbol{p}}'$ and $\tilde{\boldsymbol{p}}_i$ only share the same positional encoding or the same TRR pattern,

$$1 \geq \tilde{\boldsymbol{p}'}^{\top}(\tilde{\boldsymbol{p}}_i - \sum_{r=1}^{l} \text{softmax}(\tilde{\boldsymbol{p}}_r^{\top}\boldsymbol{W}\tilde{\boldsymbol{p}}_{query})\tilde{\boldsymbol{p}}_r)\tilde{\boldsymbol{p}}_{query}^{\top}\tilde{\boldsymbol{p}} \geq p_n(t). \tag{131}$$

When $\tilde{\boldsymbol{p}}'$ and $\tilde{\boldsymbol{p}}_i$ only share both different positional encodings and TRR patterns,

$$0 \geq \tilde{\boldsymbol{p}'}^{\top}(\tilde{\boldsymbol{p}}_i - \sum_{r=1}^{l} \text{softmax}(\tilde{\boldsymbol{p}}_r^{\top}\boldsymbol{W}\tilde{\boldsymbol{p}}_{query})\tilde{\boldsymbol{p}}_r)\tilde{\boldsymbol{p}}_{query}^{\top}\tilde{\boldsymbol{p}} \geq -1 + p_n(t). \tag{132}$$

Next, we consider the case where $\tilde{\boldsymbol{p}}$ only shares the same positional encoding as $\tilde{\boldsymbol{p}}_{query}$. If $\tilde{\boldsymbol{p}}'$ and $\tilde{\boldsymbol{p}}_i$ share the same TRR pattern, label pattern, and the positional encoding,

$$3 - p_n(t) \geq \tilde{\boldsymbol{p}'}^{\top}(\tilde{\boldsymbol{p}}_i - \sum_{r=1}^{l} \text{softmax}(\tilde{\boldsymbol{p}}_r^{\top}\boldsymbol{W}\tilde{\boldsymbol{p}}_{query})\tilde{\boldsymbol{p}}_r)\tilde{\boldsymbol{p}}_{query}^{\top}\tilde{\boldsymbol{p}} \geq 1 \cdot (3 - p_n(t) - (1 - p_n(t))) = 2. \tag{133}$$

When $\tilde{\boldsymbol{p}}'$ and $\tilde{\boldsymbol{p}}_i$ only share the same positional encoding or the same TRR pattern,

$$1 - p_n(t) \geq \tilde{\boldsymbol{p}'}^{\top}(\tilde{\boldsymbol{p}}_i - \sum_{r=1}^{l} \text{softmax}(\tilde{\boldsymbol{p}}_r^{\top}\boldsymbol{W}\tilde{\boldsymbol{p}}_{query})\tilde{\boldsymbol{p}}_r)\tilde{\boldsymbol{p}}_{query}^{\top}\tilde{\boldsymbol{p}} \geq 0. \tag{134}$$

When $\tilde{\boldsymbol{p}}'$ and $\tilde{\boldsymbol{p}}_i$ only share both different positional encodings and TRR patterns,

$$-p_n(t) \geq \tilde{\boldsymbol{p}'}^{\top}(\tilde{\boldsymbol{p}}_i - \sum_{r=1}^{l} \text{softmax}(\tilde{\boldsymbol{p}}_r^{\top}\boldsymbol{W}\tilde{\boldsymbol{p}}_{query})\tilde{\boldsymbol{p}}_r)\tilde{\boldsymbol{p}}_{query}^{\top}\tilde{\boldsymbol{p}} \geq -1. \tag{135}$$

Then, when $l \geq \Omega(\alpha^{-1})$, and when $\tilde{\boldsymbol{p}}$ shares the same TRR pattern and the positional encoding as $\tilde{\boldsymbol{p}}_{query}$, by (107),

$$(\sum_{i=1}^{l} \text{softmax}(\tilde{\boldsymbol{p}}_i^{\top}\boldsymbol{W}\tilde{\boldsymbol{p}}_{query})\boldsymbol{W}_V\tilde{\boldsymbol{p}}_i - \boldsymbol{z}^n)^{\top} \sum_{i=1}^{l} \text{softmax}(\tilde{\boldsymbol{p}}_i^{\top}\boldsymbol{W}\tilde{\boldsymbol{p}}_{query})\boldsymbol{W}_V\tilde{\boldsymbol{p}}_i$$

$$\cdot \tilde{\boldsymbol{p}'}^{\top}(\tilde{\boldsymbol{p}}_i - \sum_{r=1}^{l} \text{softmax}(\tilde{\boldsymbol{p}}_r^{\top}\boldsymbol{W}\tilde{\boldsymbol{p}}_{query})\tilde{\boldsymbol{p}}_r)\tilde{\boldsymbol{p}}_{query}^{\top}\tilde{\boldsymbol{p}} \tag{136}$$

$$\leq 0 - 2(1 - p_n(t))^2 \frac{\alpha^2}{K^2} + \frac{1}{l}(\frac{1}{K} - \frac{\alpha}{K})(-2(K - 1)).$$

We next consider the case where $\tilde{\boldsymbol{p}}$ shares the same TRR pattern and the different positional encoding as $\tilde{\boldsymbol{p}}_{query}$. Then,

$$(\sum_{i=1}^{l} \text{softmax}(\tilde{\boldsymbol{p}}_i^{\top}\boldsymbol{W}\tilde{\boldsymbol{p}}_{query})\boldsymbol{W}_V\tilde{\boldsymbol{p}}_i - \boldsymbol{z}^n)^{\top} \sum_{i=1}^{l} \text{softmax}(\tilde{\boldsymbol{p}}_i^{\top}\boldsymbol{W}\tilde{\boldsymbol{p}}_{query})\boldsymbol{W}_V\tilde{\boldsymbol{p}}_i$$

$$\cdot \tilde{\boldsymbol{p}'}^{\top}(\tilde{\boldsymbol{p}}_i - \sum_{r=1}^{l} \text{softmax}(\tilde{\boldsymbol{p}}_r^{\top}\boldsymbol{W}\tilde{\boldsymbol{p}}_{query})\tilde{\boldsymbol{p}}_r)\tilde{\boldsymbol{p}}_{query}^{\top}\tilde{\boldsymbol{p}}$$

$$\leq -p_n(t)(1 - p_n(t))(-1 + p_n(t)) + p_n(t)(1 - p_n(t))^2 \cdot \frac{\alpha^2}{K^2} + \frac{1}{l}(\frac{1}{K} - \frac{\alpha}{K})K(-1 + p_n(t))$$

$$= p_n(t)(1 - p_n(t))^2(\frac{\alpha^2}{K^2} + 1) + \frac{1}{l}(1 - \alpha)(-1 + p_n(t)). \tag{137}$$

We next consider the case where $\tilde{p}$ shares the same positional encoding and the different TRR pattern as $\tilde{p}_{query}$. Then,

$$
(\sum_{i=1}^{l} \text{softmax}(\tilde{p_i}^\top W \tilde{p}_{query}) W_V \tilde{p}_i - z^n)^\top \sum_{i=1}^{l} \text{softmax}(\tilde{p_i}^\top W \tilde{p}_{query}) W_V \tilde{p}_i
$$

$$
\cdot \tilde{p'}^\top (\tilde{p}_i - \sum_{r=1}^{l} \text{softmax}(\tilde{p_r}^\top W \tilde{p}_{query}) \tilde{p}_r) \tilde{p}_{query}^\top \tilde{p} \tag{138}
$$

$$
\leq -(1-p_n(t))^2 \frac{\alpha^2}{K^2} - 0 + \frac{1}{l}(\frac{1}{K} - \frac{\alpha}{K})(-K+1)
$$

$$
= -(1-p_n(t))^2 \frac{\alpha^2}{K^2} - \frac{K-1}{Kl}(1-\alpha).
$$

Therefore, as long as

$$
l \geq \Omega(\alpha^{-1}), \tag{139}
$$

we have

$$
\tilde{p'}^\top \eta \frac{1}{B} \sum_{n \in \mathcal{B}_b} \frac{\partial \ell(\Psi; P^n, z^n)}{\partial W} p
$$

$$
= \eta \frac{1}{B} \sum_{n \in \mathcal{B}_b} (F(\Psi; P) - z^n)^\top \sum_{i=1}^{l} W_V \tilde{p}_i \text{softmax}(\tilde{p_i}^\top W \tilde{p}_{query})
$$

$$
\cdot \tilde{p}^\top (\tilde{p}_i - \sum_{r=1}^{l} \text{softmax}(\tilde{p_r}^\top W \tilde{p}_{query}) \tilde{p}_r) \tilde{p}_{query}^\top \tilde{p}
$$

$$
\leq \eta \frac{1}{B} \sum_{n \in \mathcal{B}_b} (\frac{1}{KM}(-\frac{\alpha^2}{K^2} + (K-1)(1+\frac{\alpha^2}{K^2})p_n(t))(1-p_n(t))^2 - (\frac{1}{K}
$$

$$
- \frac{1}{M})(1-p_n(t))^2 \frac{\alpha^2}{K^2}))
$$

$$
= \eta \cdot \frac{1}{B} \sum_{n \in \mathcal{B}_b} (\frac{1}{KM}(-\frac{\alpha^2}{K^2} + (K-1+\frac{(2K-1)\alpha^2}{K^2})p_n(t))(1-p_n(t))^2 - (1-p_n(t))^2 \frac{\alpha^2}{K^3}). \tag{140}
$$

and

$$
\tilde{p'}^\top \eta \frac{1}{B} \sum_{n \in \mathcal{B}_b} \frac{\partial \ell(\Psi; P^n, z^n)}{\partial W} p
$$

$$
\geq \eta \cdot \frac{1}{B} \sum_{n \in \mathcal{B}_b} (\frac{1}{KM}(-\frac{\alpha^2}{K^2} + (K-1+\frac{(2K-1)\alpha^2}{K^2})p_n(t))(1-p_n(t))^2 - (1-p_n(t))^2 \frac{\alpha^2}{K^3}
$$

$$
+ \frac{1}{K} \cdot (1-p_n(t))^2(-p_n(t) + (1-p_n(t))\frac{\alpha^2}{K^2})). \tag{141}
$$

We next consider the case where $\tilde{p}'$ shares a different TRR pattern and a different positional encoding as $\tilde{p}$. Let $\tilde{p}$ share the same TRR pattern and the positional encoding as $\tilde{p}_{query}$. If $\tilde{p}'$ and $\tilde{p}_i$ share the same TRR pattern, label pattern, and the positional encoding,

$$
6 \geq \tilde{p'}^\top (\tilde{p}_i - \sum_{r=1}^{l} \text{softmax}(\tilde{p_r}^\top W \tilde{p}_{query}) \tilde{p}_r) \tilde{p}_{query}^\top \tilde{p} \geq 2 \cdot (3 - (1-p_n(t))) = 4 + 2p_n(t). \tag{142}
$$

When $\tilde{p}'$ and $\tilde{p}_i$ only share the same positional encoding or the same TRR pattern,

$$
2 \geq \tilde{p'}^\top (\tilde{p}_i - \sum_{r=1}^{l} \text{softmax}(\tilde{p_r}^\top W \tilde{p}_{query}) \tilde{p}_r) \tilde{p}_{query}^\top \tilde{p} \geq 2p_n(t). \tag{143}
$$

When $\tilde{p}'$ and $\tilde{p}_i$ only share both different positional encodings and TRR patterns,

$$0 \geq \tilde{p'}^\top (\tilde{p}_i - \sum_{r=1}^{l} \text{softmax}(\tilde{p_r}^\top W \tilde{p}_{query}) \tilde{p}_r) \tilde{p}_{query}^\top \tilde{p} \geq -2 + 2p_n(t). \tag{144}$$

Then, we consider the case where $\tilde{p}$ only shares the same TRR pattern as $\tilde{p}_{query}$. If $\tilde{p}'$ and $\tilde{p}_i$ share the same TRR pattern, label pattern, and the positional encoding,

$$3 \geq \tilde{p'}^\top (\tilde{p}_i - \sum_{r=1}^{l} \text{softmax}(\tilde{p_r}^\top W \tilde{p}_{query}) \tilde{p}_r) \tilde{p}_{query}^\top \tilde{p} \geq 1 \cdot (3 - p_n(t) - (1 - p_n(t))) = 2. \tag{145}$$

When $\tilde{p}'$ and $\tilde{p}_i$ only share the same positional encoding or the same TRR pattern,

$$1 \geq \tilde{p'}^\top (\tilde{p}_i - \sum_{r=1}^{l} \text{softmax}(\tilde{p_r}^\top W \tilde{p}_{query}) \tilde{p}_r) \tilde{p}_{query}^\top \tilde{p} \geq 0. \tag{146}$$

When $\tilde{p}'$ and $\tilde{p}_i$ only share both different positional encodings and TRR patterns,

$$0 \geq \tilde{p'}^\top (\tilde{p}_i - \sum_{r=1}^{l} \text{softmax}(\tilde{p_r}^\top W \tilde{p}_{query}) \tilde{p}_r) \tilde{p}_{query}^\top \tilde{p} \geq -1. \tag{147}$$

Next, we consider the case where $\tilde{p}$ only shares the same positional encoding as $\tilde{p}_{query}$. If $\tilde{p}'$ and $\tilde{p}_i$ share the same TRR pattern, label pattern, and the positional encoding,

$$3 \geq \tilde{p'}^\top (\tilde{p}_i - \sum_{r=1}^{l} \text{softmax}(\tilde{p_r}^\top W \tilde{p}_{query}) \tilde{p}_r) \tilde{p}_{query}^\top \tilde{p} \geq 1 \cdot (3 - (1 - p_n(t))) = 2 + p_n(t). \tag{148}$$

When $\tilde{p}'$ and $\tilde{p}_i$ only share the same positional encoding or the same TRR pattern,

$$1 \geq \tilde{p'}^\top (\tilde{p}_i - \sum_{r=1}^{l} \text{softmax}(\tilde{p_r}^\top W \tilde{p}_{query}) \tilde{p}_r) \tilde{p}_{query}^\top \tilde{p} \geq p_n(t). \tag{149}$$

When $\tilde{p}'$ and $\tilde{p}_i$ only share both different positional encodings and TRR patterns,

$$0 \geq \tilde{p'}^\top (\tilde{p}_i - \sum_{r=1}^{l} \text{softmax}(\tilde{p_r}^\top W \tilde{p}_{query}) \tilde{p}_r) \tilde{p}_{query}^\top \tilde{p} \geq -1 + p_n(t). \tag{150}$$

Then, when $l \geq \Omega(\alpha^{-1})$, and when $\tilde{p}$ shares the same TRR pattern and the positional encoding as $\tilde{p}_{query}$,

$$(\sum_{i=1}^{l} \text{softmax}(\tilde{p}_i^\top W \tilde{p}_{query}) W_V \tilde{p}_i - z^n)^\top \sum_{i=1}^{l} \text{softmax}(\tilde{p}_i^\top W \tilde{p}_{query}) W_V \tilde{p}_i$$

$$\cdot \tilde{p'}^\top (\tilde{p}_i - \sum_{r=1}^{l} \text{softmax}(\tilde{p_r}^\top W \tilde{p}_{query}) \tilde{p}_r) \tilde{p}_{query}^\top \tilde{p}$$

$$\leq - p_n(t)(1 - p_n(t))(-2 + 2p_n(t)) + (1 - p_n(t))^2 \frac{\alpha^2}{K^2} \cdot 2p_n(t) + \frac{1}{l}(1 - \alpha)(-2 + 2p_n(t)). \tag{151}$$

We next consider the case where $\tilde{p}$ shares the same TRR pattern and the different positional encoding as $\tilde{p}_{query}$. Then,

$$(\sum_{i=1}^{l} \text{softmax}(\tilde{p}_i^\top W \tilde{p}_{query}) W_V \tilde{p}_i - z^n)^\top \sum_{i=1}^{l} \text{softmax}(\tilde{p}_i^\top W \tilde{p}_{query}) W_V \tilde{p}_i$$

$$\cdot \tilde{p'}^\top (\tilde{p}_i - \sum_{r=1}^{l} \text{softmax}(\tilde{p_r}^\top W \tilde{p}_{query}) \tilde{p}_r) \tilde{p}_{query}^\top \tilde{p}$$

$$\leq 0 + p_n(t)(1 - p_n(t))^2 \cdot \frac{\alpha^2}{K^2} \cdot (-1) + \frac{1}{l}(\frac{1}{K} - \frac{\alpha}{K})(-K)$$

$$= - p_n(t)(1 - p_n(t))^2 \frac{\alpha^2}{K^2} + \frac{1}{l}(1 - \alpha)(-1). \tag{152}$$

We next consider the case where $\tilde{\boldsymbol{p}}$ shares the same positional encoding and the different TRR pattern as $\tilde{\boldsymbol{p}}_{query}$. Then,

$$
\begin{aligned}
&(\sum_{i=1}^{l} \text{softmax}(\tilde{\boldsymbol{p}_i}^\top \boldsymbol{W} \tilde{\boldsymbol{p}}_{query}) \boldsymbol{W}_V \tilde{\boldsymbol{p}}_i - \boldsymbol{z}^n)^\top \sum_{i=1}^{l} \text{softmax}(\tilde{\boldsymbol{p}_i}^\top \boldsymbol{W} \tilde{\boldsymbol{p}}_{query}) \boldsymbol{W}_V \tilde{\boldsymbol{p}}_i \\
&\quad \cdot \tilde{\boldsymbol{p}'}^\top (\tilde{\boldsymbol{p}}_i - \sum_{r=1}^{l} \text{softmax}(\tilde{\boldsymbol{p}_r}^\top \boldsymbol{W} \tilde{\boldsymbol{p}}_{query}) \tilde{\boldsymbol{p}_r}) \tilde{\boldsymbol{p}}_{query}^\top \tilde{\boldsymbol{p}} \\
&\leq - (1 - p_n(t)) p_n(t)(-1 + p_n(t)) + p_n(t)(1 - p_n(t))^2 \frac{\alpha^2}{K^2} + \frac{1}{l}(\frac{1}{K} - \frac{\alpha}{K})(-1 + p_n(t))K \\
&= (1 - p_n(t))^2 p_n(t)(1 + \frac{\alpha^2}{K^2}) + \frac{1}{l}(1 - \alpha)(-1 + p_n(t)).
\end{aligned}
\tag{153}
$$

Therefore, as long as

$$
l \geq \Omega(\alpha^{-1}),
\tag{154}
$$

we have

$$
\begin{aligned}
&\tilde{\boldsymbol{p}'}^\top \eta \frac{1}{B} \sum_{n \in \mathcal{B}_b} \frac{\partial \ell(\Psi; \boldsymbol{P}^n, \boldsymbol{z}^n)}{\partial \boldsymbol{W}} \boldsymbol{p} \\
&= \eta \frac{1}{B} \sum_{n \in \mathcal{B}_b} (F(\Psi; \boldsymbol{P}) - \boldsymbol{z}^n)^\top \sum_{i=1}^{l} \boldsymbol{W}_V \tilde{\boldsymbol{p}}_i \text{softmax}(\tilde{\boldsymbol{p}_i}^\top \boldsymbol{W} \tilde{\boldsymbol{p}}_{query}) \\
&\quad \cdot \tilde{\boldsymbol{p}}^\top (\tilde{\boldsymbol{p}}_i - \sum_{r=1}^{l} \text{softmax}(\tilde{\boldsymbol{p}_r}^\top \boldsymbol{W} \tilde{\boldsymbol{p}}_{query}) \tilde{\boldsymbol{p}_r}) \tilde{\boldsymbol{p}}_{query}^\top \tilde{\boldsymbol{p}} \\
&\leq \eta \frac{1}{B} \sum_{n \in \mathcal{B}_b} (\frac{1}{KM}(-p_n(t)(1 - p_n(t))(-2 + 2 p_n(t)) + (3 - K)(1 - p_n(t))^2 \frac{\alpha^2}{K^2} \cdot p_n(t)) \\
&\quad + (\frac{1}{K} - \frac{1}{M})(1 - p_n(t))^2 p_n(t)(1 + \frac{\alpha^2}{K^2})) \\
&= \eta \cdot \frac{1}{B} \sum_{n \in \mathcal{B}_b} (\frac{1}{KM} p_n(t)(1 - p_n(t))^2 (2 - K + \frac{(2 - K)\alpha^2}{K^2}) \\
&\quad + (1 - p_n(t))^2 p_n(t)(1 + \frac{\alpha^2}{K^2}) \cdot \frac{1}{K}),
\end{aligned}
\tag{155}
$$

and

$$
\begin{aligned}
&\tilde{\boldsymbol{p}'}^\top \eta \frac{1}{B} \sum_{n \in \mathcal{B}_b} \frac{\partial \ell(\Psi; \boldsymbol{P}^n, \boldsymbol{z}^n)}{\partial \boldsymbol{W}} \boldsymbol{p} \\
&\geq \eta \cdot \frac{1}{B} \sum_{n \in \mathcal{B}_b} (\frac{1}{KM} p_n(t)(1 - p_n(t))^2 (1 + \frac{(2 - K)\alpha^2}{K^2}) + (1 - p_n(t))^2 p_n(t)(1 + \frac{\alpha^2}{K^2}) \cdot \frac{1}{K} \\
&\quad + \frac{1}{K} \cdot (1 - p_n(t))^2 (-p_n(t) + (1 - p_n(t))\frac{\alpha^2}{K^2})) \\
&= \eta \cdot \frac{1}{B} \sum_{n \in \mathcal{B}_b} (\frac{1}{KM} p_n(t)(1 - p_n(t))^2 (1 + \frac{(2 - K)\alpha^2}{K^2}) + (1 - p_n(t))^2 \cdot \frac{\alpha^2}{K^3}).
\end{aligned}
\tag{156}
$$

$\square$

### F.2 PROOF OF LEMMA 4

*Proof.* We can derive that when $1 - p_n(t) \geq \Omega(1)$, $\tilde{\boldsymbol{p}'}^\top \boldsymbol{W}^{(t)} \tilde{\boldsymbol{p}}$ increases if $\tilde{\boldsymbol{p}}$ and $\tilde{\boldsymbol{p}'}$ share the same positional encoding. Otherwise, $\tilde{\boldsymbol{p}'}^\top \boldsymbol{W}^{(t)} \tilde{\boldsymbol{p}}$ decreases. We know that $p_n(t) \geq \frac{\alpha}{2}$. Combining the

results in Lemma 3, we can derive that when $t \geq 1$,

$$\boldsymbol{W}^{(t+1)} = \boldsymbol{W}^{(t)} - \eta \frac{1}{B} \sum_{n \in \mathcal{B}_b} \frac{\partial \ell(\Psi; \boldsymbol{P}^n, \boldsymbol{z}^n)}{\partial \boldsymbol{W}^{(t)}}. \tag{157}$$

Then, for $\tilde{\boldsymbol{p}_i}^n$ that share the same TRR pattern and the same positional encoding of $\tilde{\boldsymbol{p}}_{query}^n$,

$$\frac{p_n(t+1)}{|\mathcal{S}_1^n|} = \text{softmax}(\boldsymbol{p}_i^{n\top} \boldsymbol{W}^{(t+1)} \tilde{\boldsymbol{p}}_{query}^n)$$
$$\geq \frac{1}{l} \cdot \frac{1}{\frac{\alpha}{K} + \frac{(K-1)\alpha}{K} \cdot e^{-s_1} + (\frac{1}{K} - \frac{\alpha}{K})((K-1)e^{-s_2} + e^{-s_3})}, \tag{158}$$

where

$$s_1 \geq \eta \sum_{b=0}^{t} ((1 - p_n(b))^2 \frac{\alpha^2}{K^3} + \frac{\alpha^2}{K^3}(1 - p_n(b))^2) = \eta \sum_{b=0}^{t} (1 - p_n(b))^2 \frac{2\alpha^2}{K^3}, \tag{159}$$

$$s_2 \geq \sum_{b=0}^{t} (1 - p_n(b))^2 \cdot \frac{2\eta\alpha^2}{K^3}, \tag{160}$$

$$s_3 \geq -\frac{\eta}{KM} \sum_{b=0}^{t} (1 - p_n(b)^2(-4p_n(b)(1 + \frac{\alpha^2}{K^2}) + \frac{\alpha^2}{K}(1 + \frac{2(K-1)}{K}) + \frac{\alpha^2}{K^2}$$
$$- (K - 1 + \frac{2K-1}{K^2}\alpha^2)p_n(b)))$$
$$\geq \frac{\eta}{KM} \sum_{b=0}^{t} (1 - p_n(b))^2(p_n(b)(3 + \frac{\alpha^2}{K^2})(4 + \frac{2K-1}{K^2})), \tag{161}$$

where the last step is by $Kp_n(b) \geq 4\alpha^2/K^2$ when $p_n(b) \geq \alpha/K$. For $\tilde{\boldsymbol{p}_i}^n$ that share the same TRR pattern and a different positional encoding of $\tilde{\boldsymbol{p}}_{query}^n$,

$$\text{softmax}(\tilde{\boldsymbol{p}_i^n}^\top \boldsymbol{W}^{(t+1)} \tilde{\boldsymbol{p}}_{query}^n) = \frac{1}{l} \cdot \frac{1}{\frac{\alpha}{K}e^{s_1} + \frac{(K-1)\alpha}{K} + (\frac{1}{K} - \frac{\alpha}{K})((K-1)e^{-s_4} + e^{s_5})}, \tag{162}$$

where

$$s_4 \geq -\sum_{b=0}^{t} \frac{\eta}{M}((-4 - (3K - 2)(1 - p_n(b))(1 + \frac{\alpha^2}{K^2}))p_n(b)(1 - p_n(b))$$
$$- (2 - K)(1 + \frac{\alpha^2}{K^2})p_n(b)(1 - p_n(b))^2)$$
$$= \sum_{b=0}^{t} \frac{\eta}{M}(4 + 2K(1 - p_n(b))(1 + \frac{\alpha^2}{K^2}))p_n(b)(1 - p_n(b)), \tag{163}$$

$$s_5 \geq \sum_{b=0}^{t} (1 - p_n(b))^2 \cdot \frac{2\eta\alpha^2}{K^3}. \tag{164}$$

When $M \geq \Omega(K^4\alpha^{-1})$ and $t \geq \Omega(\eta^{-1}K^3 \log K\alpha^{-2})$,

$$(K - 1)e^{-s_4} + e^{s_5} > K. \tag{165}$$

If $M \geq \Omega(K^4\alpha^{-1})$ and $t \leq O(\eta^{-1}K^3 \log K\alpha^{-2})$, we cannot ensure

$$(K - 1)e^{-s_4} + e^{s_5} > K. \tag{166}$$

For $\tilde{\boldsymbol{p}_i^n}$ that share a different TRR pattern and the same positional encoding of $\tilde{\boldsymbol{p}}_{query}^n$,

$$\text{softmax}(\tilde{\boldsymbol{p}_i^n}^\top \boldsymbol{W}^{(t+1)} \tilde{\boldsymbol{p}}_{query}^n) = \frac{1}{l} \cdot \frac{1}{\frac{\alpha}{K}e^{s_3} + \frac{\alpha}{K} \cdot e^{-s_4} + (\frac{1}{K} - \frac{\alpha}{K})(1 + (K-1)e^{-s_6})}, \tag{167}$$

where

$$s_6 \geq \eta \sum_{b=0}^{t} \frac{2\alpha^2}{K^3}(1 - p_n(b))^2. \tag{168}$$

For $\tilde{\boldsymbol{p}}_i^n$ that share a different TRR pattern and a different positional encoding of $\tilde{\boldsymbol{p}}_{query}^n$,

$$\mathrm{softmax}(\tilde{\boldsymbol{p}}_i^{n\top}\boldsymbol{W}^{(t+1)}\tilde{\boldsymbol{p}}_{query}^n) = \frac{1}{l} \cdot \frac{1}{\frac{\alpha}{K}e^{s_2} + (\frac{1}{K} - \frac{\alpha}{K})(K - 1 + e^{s_6}) + \frac{\alpha}{K}e^{s_4}}. \tag{169}$$

Note that when $t \lesssim \eta^{-1}\alpha^{-2}K^3$, for $\boldsymbol{p}_{query}^n$ in the $k$-th step, we have

$$\sum_{i \in \mathcal{S}_{[K]\setminus\{k\}}} \mathrm{softmax}(\tilde{\boldsymbol{p}}_i^{n\top}\boldsymbol{W}^{(t+1)}\tilde{\boldsymbol{p}}_{query}^n) \geq \Omega(1), \tag{170}$$

for $\tilde{\boldsymbol{p}}_i^n$ that share a different positional encoding from $\tilde{\boldsymbol{p}}_{query}^n$. To make the total softmax values on contexts that share a different positional encoding and a different TRR pattern from the query smaller than $\epsilon$, we need

$$s_1, s_2, s_6 \gtrsim \log \frac{K}{\epsilon}. \tag{171}$$

When $t$ further increases to be larger than $\Omega(\eta^{-1}\alpha^{-2}K^3 \log \frac{K}{\epsilon})$, we also have that the total softmax values on contexts that share a different positional encoding and the same TRR pattern from the query smaller than $\epsilon$. Therefore,

$$t \gtrsim T_1 := \eta^{-1}\alpha^{-2}K^3 \log \frac{K}{\epsilon}. \tag{172}$$

$\square$

### F.3 PROOF OF LEMMA 5

*Proof.* We consider the case when $t \geq T_1$ given Lemma 4. When $l \geq \Omega(\alpha^{-1})$, and when $\tilde{\boldsymbol{p}}$ shares the same TRR pattern and the positional encoding as $\tilde{\boldsymbol{p}}_{query}$,

$$\begin{aligned}
&(\sum_{i=1}^{l} \mathrm{softmax}(\tilde{\boldsymbol{p}}_i^{\top}\boldsymbol{W}\tilde{\boldsymbol{p}}_{query})\boldsymbol{W}_V\tilde{\boldsymbol{p}}_i - \boldsymbol{z}^n)^{\top} \sum_{i=1}^{l} \mathrm{softmax}(\tilde{\boldsymbol{p}}_i^{\top}\boldsymbol{W}\tilde{\boldsymbol{p}}_{query})\boldsymbol{W}_V\tilde{\boldsymbol{p}}_i \\
&\cdot \tilde{\boldsymbol{p}}^{\top}(\tilde{\boldsymbol{p}}_i - \sum_{r=1}^{l} \mathrm{softmax}(\tilde{\boldsymbol{p}}_r^{\top}\boldsymbol{W}\tilde{\boldsymbol{p}}_{query})\tilde{\boldsymbol{p}}_r)\tilde{\boldsymbol{p}}_{query}^{\top}\tilde{\boldsymbol{p}} \\
&\leq -4p_n(t)(1 - p_n(t))^2 + \epsilon \\
&\lesssim -4p_n(t)(1 - p_n(t))^2.
\end{aligned} \tag{173}$$

We next consider the case where $\tilde{\boldsymbol{p}}$ shares the same TRR pattern and the different positional encoding as $\tilde{\boldsymbol{p}}_{query}$. Then,

$$\begin{aligned}
&(\sum_{i=1}^{l} \mathrm{softmax}(\tilde{\boldsymbol{p}}_i^{\top}\boldsymbol{W}\tilde{\boldsymbol{p}}_{query})\boldsymbol{W}_V\tilde{\boldsymbol{p}}_i - \boldsymbol{z}^n)^{\top} \sum_{i=1}^{l} \mathrm{softmax}(\tilde{\boldsymbol{p}}_i^{\top}\boldsymbol{W}\tilde{\boldsymbol{p}}_{query})\boldsymbol{W}_V\tilde{\boldsymbol{p}}_i \\
&\cdot \tilde{\boldsymbol{p}}^{\top}(\tilde{\boldsymbol{p}}_i - \sum_{r=1}^{l} \mathrm{softmax}(\tilde{\boldsymbol{p}}_r^{\top}\boldsymbol{W}\tilde{\boldsymbol{p}}_{query})\tilde{\boldsymbol{p}}_r)\tilde{\boldsymbol{p}}_{query}^{\top}\tilde{\boldsymbol{p}} \\
&\lesssim -0 \cdot p_n(t)(1 - p_n(t)) + \epsilon \\
&\lesssim \epsilon.
\end{aligned} \tag{174}$$

We next consider the case where $\tilde{\boldsymbol{p}}$ shares the same positional encoding and the different TRR pattern as $\tilde{\boldsymbol{p}}_{query}$. Then,

$$\begin{aligned}
&(\sum_{i=1}^{l} \mathrm{softmax}(\tilde{\boldsymbol{p}}_i^{\top}\boldsymbol{W}\tilde{\boldsymbol{p}}_{query})\boldsymbol{W}_V\tilde{\boldsymbol{p}}_i - \boldsymbol{z}^n)^{\top} \sum_{i=1}^{l} \mathrm{softmax}(\tilde{\boldsymbol{p}}_i^{\top}\boldsymbol{W}\tilde{\boldsymbol{p}}_{query})\boldsymbol{W}_V\tilde{\boldsymbol{p}}_i \\
&\cdot \tilde{\boldsymbol{p}}^{\top}(\tilde{\boldsymbol{p}}_i - \sum_{r=1}^{l} \mathrm{softmax}(\tilde{\boldsymbol{p}}_r^{\top}\boldsymbol{W}\tilde{\boldsymbol{p}}_{query})\tilde{\boldsymbol{p}}_r)\tilde{\boldsymbol{p}}_{query}^{\top}\tilde{\boldsymbol{p}} \\
&\lesssim \epsilon.
\end{aligned} \tag{175}$$

Therefore,

$$
\tilde{\boldsymbol{p}}^\top \eta \frac{1}{B} \sum_{n \in \mathcal{B}_b} \frac{\partial \ell(\Psi; \boldsymbol{P}^n, \boldsymbol{z}^n)}{\partial \boldsymbol{W}} \boldsymbol{p}
$$

$$
= \eta \frac{1}{B} \sum_{n \in \mathcal{B}_b} (F(\Psi; \boldsymbol{P}) - \boldsymbol{z}^n)^\top \sum_{i=1}^{l} \boldsymbol{W}_V \tilde{\boldsymbol{p}}_i \mathrm{softmax}(\tilde{\boldsymbol{p}_i}^\top \boldsymbol{W} \tilde{\boldsymbol{p}}_{query})
$$

$$
\cdot \tilde{\boldsymbol{p}}^\top (\tilde{\boldsymbol{p}}_i - \sum_{r=1}^{l} \mathrm{softmax}(\tilde{\boldsymbol{p}_r}^\top \boldsymbol{W} \tilde{\boldsymbol{p}}_{query}) \tilde{\boldsymbol{p}}_r) \tilde{\boldsymbol{p}}_{query}^\top \tilde{\boldsymbol{p}} \tag{176}
$$

$$
\lesssim \eta \frac{1}{B} \sum_{n \in \mathcal{B}_b} (\frac{1}{2M}(-4p_n(t)(1-p_n(t))^2) + (\frac{1}{2} - \frac{1}{M}) \cdot \epsilon
$$

$$
= -\eta \cdot \frac{1}{2M} \cdot \frac{1}{B} \sum_{n \in \mathcal{B}_b} 4p_n(t)(1-p_n(t))^2.
$$

We then discuss if $\tilde{\boldsymbol{p}}$ and $\tilde{\boldsymbol{p}}'$ only share the same TRR pattern. When $l \geq \Omega(\alpha^{-1})$, and when $\tilde{\boldsymbol{p}}$ shares the same TRR pattern and the positional encoding as $\tilde{\boldsymbol{p}}_{query}$, we can obtain

$$
(\sum_{i=1}^{l} \mathrm{softmax}(\tilde{\boldsymbol{p}_i}^\top \boldsymbol{W} \tilde{\boldsymbol{p}}_{query}) \boldsymbol{W}_V \tilde{\boldsymbol{p}}_i - \boldsymbol{z}^n)^\top \sum_{i=1}^{l} \mathrm{softmax}(\tilde{\boldsymbol{p}_i}^\top \boldsymbol{W} \tilde{\boldsymbol{p}}_{query}) \boldsymbol{W}_V \tilde{\boldsymbol{p}}_i
$$

$$
\cdot \tilde{\boldsymbol{p}}'^\top (\tilde{\boldsymbol{p}}_i - \sum_{r=1}^{l} \mathrm{softmax}(\tilde{\boldsymbol{p}_r}^\top \boldsymbol{W} \tilde{\boldsymbol{p}}_{query}) \tilde{\boldsymbol{p}}_r) \tilde{\boldsymbol{p}}_{query}^\top \tilde{\boldsymbol{p}} \tag{177}
$$

$$
\gtrsim -2(1-p_n(t))^2 p_n(t).
$$

We next consider the case where $\tilde{\boldsymbol{p}}$ shares the same TRR pattern and the different positional encoding as $\tilde{\boldsymbol{p}}_{query}$. Then,

$$
(\sum_{i=1}^{l} \mathrm{softmax}(\tilde{\boldsymbol{p}_i}^\top \boldsymbol{W} \tilde{\boldsymbol{p}}_{query}) \boldsymbol{W}_V \tilde{\boldsymbol{p}}_i - \boldsymbol{z}^n)^\top \sum_{i=1}^{l} \mathrm{softmax}(\tilde{\boldsymbol{p}_i}^\top \boldsymbol{W} \tilde{\boldsymbol{p}}_{query}) \boldsymbol{W}_V \tilde{\boldsymbol{p}}_i
$$

$$
\cdot \tilde{\boldsymbol{p}}'^\top (\tilde{\boldsymbol{p}}_i - \sum_{r=1}^{l} \mathrm{softmax}(\tilde{\boldsymbol{p}_r}^\top \boldsymbol{W} \tilde{\boldsymbol{p}}_{query}) \tilde{\boldsymbol{p}}_r) \tilde{\boldsymbol{p}}_{query}^\top \tilde{\boldsymbol{p}} \tag{178}
$$

$$
\gtrsim -(1-p_n(t))(1-p_n(t))p_n(t).
$$

We next consider the case where $\tilde{\boldsymbol{p}}$ shares the same positional encoding and the different TRR pattern as $\tilde{\boldsymbol{p}}_{query}$. Then,

$$
\left| (\sum_{i=1}^{l} \mathrm{softmax}(\tilde{\boldsymbol{p}_i}^\top \boldsymbol{W} \tilde{\boldsymbol{p}}_{query}) \boldsymbol{W}_V \tilde{\boldsymbol{p}}_i - \boldsymbol{z}^n)^\top \sum_{i=1}^{l} \mathrm{softmax}(\tilde{\boldsymbol{p}_i}^\top \boldsymbol{W} \tilde{\boldsymbol{p}}_{query}) \boldsymbol{W}_V \tilde{\boldsymbol{p}}_i \right.
$$

$$
\left. \cdot \tilde{\boldsymbol{p}}'^\top (\tilde{\boldsymbol{p}}_i - \sum_{r=1}^{l} \mathrm{softmax}(\tilde{\boldsymbol{p}_r}^\top \boldsymbol{W} \tilde{\boldsymbol{p}}_{query}) \tilde{\boldsymbol{p}}_r) \tilde{\boldsymbol{p}}_{query}^\top \tilde{\boldsymbol{p}} \right| \tag{179}
$$

$$
\lesssim \epsilon.
$$

Therefore,

$$
\left| \tilde{\boldsymbol{p}}'^\top \eta \frac{1}{B} \sum_{n \in \mathcal{B}_b} \frac{\partial \ell(\Psi; \boldsymbol{P}^n, \boldsymbol{z}^n)}{\partial \boldsymbol{W}} \boldsymbol{p} \right|
$$

$$
= \left| \eta \frac{1}{B} \sum_{n \in \mathcal{B}_b} (F(\Psi; \boldsymbol{P}) - \boldsymbol{z}^n)^\top \sum_{i=1}^{l} \boldsymbol{W}_V \tilde{\boldsymbol{p}}_i \mathrm{softmax}(\tilde{\boldsymbol{p}_i}^\top \boldsymbol{W} \tilde{\boldsymbol{p}}_{query}) \tilde{\boldsymbol{p}}^\top \right.
$$

$$
\left. \cdot (\tilde{\boldsymbol{p}}_i - \sum_{r=1}^{l} \mathrm{softmax}(\tilde{\boldsymbol{p}_r}^\top \boldsymbol{W} \tilde{\boldsymbol{p}}_{query}) \tilde{\boldsymbol{p}}_r) \tilde{\boldsymbol{p}}_{query}^\top \tilde{\boldsymbol{p}} \right| \tag{180}
$$

$$
\leq \eta \epsilon.
$$

We next discuss when $\tilde{p}$ only shares the same positional encoding as $\tilde{p}'$. When $l \geq \Omega(\alpha^{-1})$, and when $\tilde{p}$ shares the same TRR pattern and the positional encoding as $\tilde{p}_{query}$,

$$
\begin{aligned}
&(\sum_{i=1}^{l} \text{softmax}(\tilde{p_i}^\top W \tilde{p}_{query}) W_V \tilde{p_i} - z^n)^\top \sum_{i=1}^{l} \text{softmax}(\tilde{p_i}^\top W \tilde{p}_{query}) W_V \tilde{p_i} \\
&\quad \cdot \tilde{p}'^\top (\tilde{p_i} - \sum_{r=1}^{l} \text{softmax}(\tilde{p_r}^\top W \tilde{p}_{query}) \tilde{p_r}) \tilde{p}_{query}^\top \tilde{p} \\
&\lesssim \epsilon.
\end{aligned}
\tag{181}
$$

We next consider the case where $\tilde{p}$ shares the same TRR pattern and the different positional encoding as $\tilde{p}_{query}$. Then,

$$
\begin{aligned}
&(\sum_{i=1}^{l} \text{softmax}(\tilde{p_i}^\top W \tilde{p}_{query}) W_V \tilde{p_i} - z^n)^\top \sum_{i=1}^{l} \text{softmax}(\tilde{p_i}^\top W \tilde{p}_{query}) W_V \tilde{p_i} \\
&\quad \cdot \tilde{p}'^\top (\tilde{p_i} - \sum_{r=1}^{l} \text{softmax}(\tilde{p_r}^\top W \tilde{p}_{query}) \tilde{p_r}) \tilde{p}_{query}^\top \tilde{p} \\
&\lesssim -p_n(t)(1 - p_n(t))(-1 + p_n(t)) + \frac{1}{M} \\
&\lesssim p_n(t)(1 - p_n(t))^2.
\end{aligned}
\tag{182}
$$

We next consider the case where $\tilde{p}$ shares the same positional encoding and the different TRR pattern as $\tilde{p}_{query}$. Then,

$$
\begin{aligned}
&(\sum_{i=1}^{l} \text{softmax}(\tilde{p_i}^\top W \tilde{p}_{query}) W_V \tilde{p_i} - z^n)^\top \sum_{i=1}^{l} \text{softmax}(\tilde{p_i}^\top W \tilde{p}_{query}) W_V \tilde{p_i} \\
&\quad \cdot \tilde{p}'^\top (\tilde{p_i} - \sum_{r=1}^{l} \text{softmax}(\tilde{p_r}^\top W \tilde{p}_{query}) \tilde{p_r}) \tilde{p}_{query}^\top \tilde{p} \\
&\lesssim \epsilon.
\end{aligned}
\tag{183}
$$

Therefore,

$$
\begin{aligned}
&\tilde{p}'^\top \eta \frac{1}{B} \sum_{n \in \mathcal{B}_b} \frac{\partial \ell(\Psi; P^n, z^n)}{\partial W} p \\
&= \eta \frac{1}{B} \sum_{n \in \mathcal{B}_b} (F(\Psi; P) - z^n)^\top \sum_{i=1}^{l} W_V \tilde{p_i} \text{softmax}(\tilde{p_i}^\top W \tilde{p}_{query}) \tilde{p}^\top \\
&\quad \cdot (\tilde{p_i} - \sum_{r=1}^{l} \text{softmax}(\tilde{p_r}^\top W \tilde{p}_{query}) \tilde{p_r}) \tilde{p}_{query}^\top \tilde{p} \\
&\lesssim \eta \frac{1}{B} \sum_{n \in \mathcal{B}_b} \frac{1}{2M} \cdot p_n(b)(1 - p_n(b))^2.
\end{aligned}
\tag{184}
$$

We then consider if $\tilde{p}$ shares a different TRR pattern and a different positional encoding as $\tilde{p}'$. When $l \geq \Omega(\alpha^{-1})$, and when $\tilde{p}$ shares the same TRR pattern and the positional encoding as $\tilde{p}_{query}$,

$$
\begin{aligned}
&(\sum_{i=1}^{l} \text{softmax}(\tilde{p_i}^\top W \tilde{p}_{query}) W_V \tilde{p_i} - z^n)^\top \sum_{i=1}^{l} \text{softmax}(\tilde{p_i}^\top W \tilde{p}_{query}) W_V \tilde{p_i} \\
&\quad \cdot \tilde{p}'^\top (\tilde{p_i} - \sum_{r=1}^{l} \text{softmax}(\tilde{p_r}^\top W \tilde{p}_{query}) \tilde{p_r}) \tilde{p}_{query}^\top \tilde{p} \\
&\gtrsim \epsilon.
\end{aligned}
\tag{185}
$$

We next consider the case where $\tilde{p}$ shares the same TRR pattern and the different positional encoding as $\tilde{p}_{query}$. Then,

$$(\sum_{i=1}^{l} \text{softmax}(\tilde{p_i}^\top W \tilde{p}_{query}) W_V \tilde{p}_i - z^n)^\top \sum_{i=1}^{l} \text{softmax}(\tilde{p_i}^\top W \tilde{p}_{query}) W_V \tilde{p}_i$$
$$\cdot \tilde{p}'^\top (\tilde{p}_i - \sum_{r=1}^{l} \text{softmax}(\tilde{p_r}^\top W \tilde{p}_{query}) \tilde{p_r}) \tilde{p}_{query}^\top \tilde{p} \tag{186}$$
$$\gtrsim -(1 - p_n(t)) p_n(t).$$

We next consider the case where $\tilde{p}$ shares the same positional encoding and the different TRR pattern as $\tilde{p}_{query}$. Then,

$$\left| (\sum_{i=1}^{l} \text{softmax}(\tilde{p_i}^\top W \tilde{p}_{query}) W_V \tilde{p}_i - z^n)^\top \sum_{i=1}^{l} \text{softmax}(\tilde{p_i}^\top W \tilde{p}_{query}) W_V \tilde{p}_i \right.$$
$$\left. \cdot \tilde{p}'^\top (\tilde{p}_i - \sum_{r=1}^{l} \text{softmax}(\tilde{p_r}^\top W \tilde{p}_{query}) \tilde{p_r}) \tilde{p}_{query}^\top \tilde{p} \right| \tag{187}$$
$$\lesssim \epsilon.$$

Therefore,

$$\left| \tilde{p}'^\top \eta \frac{1}{B} \sum_{n \in \mathcal{B}_b} \frac{\partial \ell(\Psi; P^n, z^n)}{\partial W} p \right|$$
$$= \left| \eta \frac{1}{B} \sum_{n \in \mathcal{B}_b} (F(\Psi; P) - z^n)^\top \sum_{i=1}^{l} W_V \tilde{p}_i \text{softmax}(\tilde{p_i}^\top W \tilde{p}_{query}) \tilde{p}^\top \right.$$
$$\left. (\tilde{p}_i - \sum_{r=1}^{l} \text{softmax}(\tilde{p_r}^\top W \tilde{p}_{query}) p_r) \tilde{p}_{query}^\top \tilde{p} \right| \tag{188}$$
$$\lesssim \eta \epsilon.$$

$\square$