# OpenReview forum: "Training Nonlinear Transformers for Chain-of-Thought Inference:  A Theoretical Generalization Analysis"
_ICLR.cc/2025/Conference — ICLR 2025 Poster_

### Official Review · Reviewer_6Hy5 · 2024-11-02

**Soundness:** 3
**Presentation:** 3
**Contribution:** 3
**Rating:** 6
**Confidence:** 2

**Summary:**

This paper presents a theoretical analysis of training Transformers with nonlinear attention mechanisms to develop Chain-of-Thought (CoT) reasoning abilities. The authors focus on quantifying the requirements (such as sample complexity and iterations) for training a Transformer model capable of CoT generalization on unseen tasks, even under conditions where examples might contain noise and inaccuracies. The study also compares CoT with In-Context Learning (ICL), showing that CoT can outperform ICL when intermediate reasoning steps are essential. The paper provides both theoretical insights and empirical evidence supporting the model's behavior during training and testing.

**Strengths:**

● This paper is one of the first to theoretically explore training dynamics of nonlinear Transformers with CoT capabilities, providing important insights into sample complexity, required iterations, and generalization guarantees.

● The comparison between CoT and ICL is theoretically grounded, with a clear quantitative explanation of why CoT can succeed in contexts where ICL may fail.

● This work addresses an understudied area in Transformer models, specifically how they generalize CoT reasoning abilities across unseen tasks, making it a relevant contribution to understanding advanced prompting methods in LLMs.

**Weaknesses:**

● The authors analyze attention-only models, aligning with previous work by [1] Olsson et al. (2022). Both studies observe that trained models’ attention values tend to focus on context examples with input patterns similar to the test query. While Olsson et al. attribute this to ICL (In-Context Learning) ability, the authors link it to CoT ability and explore how training may facilitate this behavior. A more detailed comparison of contributions between this study and Olsson et al. would be valuable.

  ○ [1] In-context Learning and Induction Heads.  2022. Olsson et al.

● The paper acknowledges the gap between the one-layer model studied and multi-layer Transformer models. As practical LLMs are complex, the findings might not directly extend to real-world Transformer architectures. Please discuss and analyze the applicability of these conclusions in more complex scenarios.

● The theoretical results assume a linear span between training-relevant (TRR) and testing-relevant (TSR) patterns, limiting applicability to cases with greater diversity in test distributions.

**Questions:**

No further issues.

---

> ### Author Response · Authors · 2024-11-24
> **Response to Reviewer 6Hy5 Part I**
>
> We thank the reviewer for the valuable time and effort in the evaluation. We have updated Theorem 2 and Remark 1 according to the suggestions.
>
> **Q1 (Weakness 1)**: A more detailed comparison of contributions between this study and Olsson et al. would be valuable
>
> **A1**: This is a great question. First, [Olsson et al., 2022] is an **empirical work** that proposes the induction head mechanism only for ICL. In contrast, **we provide a theoretical analysis of how the CoT mechanism is achieved by characterizing the gradient update process of the training and analyzing the resulting trained model theoretically, which is our technical contribution**.
> Second, despite the similarity between the concluded mechanisms in attending to similar tokens as the query, there is still a difference in handling tokens with different steps. The induction head for ICL in [Olsson et al., 2022] refers to that the attention layers first attend to previous tokens that are similar to the query and then copy the next token of the attended-to token for generation. Based on our data and task formulation, our CoT mechanism states that Transformers select previous tokens with the same TRR/TSR pattern and the same step as the current query step to make generation with the next tokens of the selected ones. Therefore, our CoT mechanism also requires that the attended-to tokens should be of the same step as the query. This is because tokens corresponding to different steps but with the same pattern may have different next tokens.

---

> > ### Author Response · Authors · 2024-11-24
> > **Response to Reviewer 6Hy5 Part II**
> >
> > **Q2 (Weakness 2)**: As practical LLMs are complex, the findings might not directly extend to real-world Transformer architectures. Please discuss and analyze the applicability of these conclusions in more complex scenarios.
> >
> > **A2**: This is a good question. We have added an experiment on arithmetic tasks using GPT-4 in Section A of Appendix. Our results show that when the number of testing examples is fixed, with the number of examples with presentation error increases, the testing accuracy decreases. This is consistent with Remark 1 for Theorem 2. We also show that when two examples with implementation error exist, where Condition 1 may not hold, CoT performs better than ICL, which justifies Remark 2 for Theorem 3.
> >
> > **Note that our conclusions of one-layer single-head Transformers are empirically verified by multi-layer multi-head Transformers to some extent.** We justify the attention mechanism of CoT on a three-layer two-head Transformer in Figure 7 of the submitted manuscript on page 10. The findings show that at least one head of each layer is implementing the CoT mechanism of Proposition 1 for the single-layer single-head case. This implies that our theoretical analysis of the training dynamics of the individual head of matching tokens with the same TSR pattern and the same step as the query can potentially be extended to the multi-layer multi-head case.
> >
> > We leave the detailed theoretical analysis of the multi-layer multi-head case as future works for the following reasons. **First, the state-of-the-art theoretical works [Li et al., 2023; Huang et al., 2023 & 2024; Makkuva et al., 2024] recently published at top ML conferences like ICLR, Neurips and ICML on the theoretical generalization and/or learning dynamics of Transformers also focus on one-layer single-head Transformers only.** That is because the loss landscape for multi-layer/head Transformers is highly nonlinear and non-convex due to the interactions between multiple nonlinear functions. The extension to multi-layer multi-head Transformers requires a more complicated characterization of the gradient updates, while the simplified model architecture can benefit the theoretical analysis as mentioned by the reviewer.
> > For the theoretical generalization guarantees of Transformers beyond one-layer single-head, to the best of our knowledge, only [Chen et al., 2024; Yang et al., 2024]  theoretically study ICL with multi-head Transformers, and [Gatmiry et al., 2024] study ICL with looped Transformers, which are multi-layer Transformers but the weights for all layers are the same. To the best of our knowledge, no existing works study the optimization and generalization of CoT theoretically at all,  even for one-layer single-head Transformers. Therefore, we focus on the one-layer analysis to obtain some initial theoretical insights.
> >
> > **Second, the simplification of one-layer single-head Transformers enables us to make contributions under our CoT settings.** Our work is the first one to investigate the optimization and generalization of CoT and characterize the conditions when CoT is better than ICL. We establish the required number of context examples for a successful CoT in terms of how informative and erroneous the prompt is.
> >
> > Li et al., 2023a. A Theoretical Understanding of Shallow Vision Transformers: Learning, Generalization, and Sample Complexity. At ICLR.
> >
> > Huang et al., 2024. In-context convergence of transformers. At ICML.
> >
> > Makkuva et al., 2024. Local to Global: Learning Dynamics and Effect of Initialization for Transformers. At Neurips.
> >
> > Chen et al., 2024. Training dynamics of multi-head softmax attention for in-context learning: Emergence, convergence, and optimality. Preprint.
> >
> > Yang et al., 2024. In-context learning with representations: Contextual generalization of trained transformers. Neurips 2024.
> >
> > Gatmiry et al., 2024. Can Looped Transformers Learn to Implement Multi-step Gradient Descent for In-context Learning? At ICML.

---

> ### Author Response · Authors · 2024-11-24
> **Response to Reviewer 6Hy5 Part III**
>
> **Q3 (Weakness 3)**: The theoretical results assume a linear span between training-relevant (TRR) and testing-relevant (TSR) patterns, limiting applicability to cases with greater diversity in test distributions.
>
> **A3**: We thank the reviewer for this question. We first want to clarify that this is a sufficient condition for a successful CoT generalization with distribution-shifted data, rather than an assumption or a specific setting for the theoretical analysis. To the best of our knowledge, no existing works study the optimization and generalization of CoT theoretically at all,  even on pattern transition tasks we study. Therefore, we focus on sufficient conditions under which we can obtain some initial theoretical insights.
>
> We realize that theoretical results can be extended to more general cases where TSR patterns can have components outside the linear span of TRR patterns. Our Theorem 2, Remark 1, and the proof have been updated accordingly. Specifically, each TSR pattern $\mu_j'$ in the orthonormal set $\\{\mu_j'\\}_{j=1}^{M'}$ can be revised into $\mu_j' =\lambda_j+\tilde{\mu}_j$, where $\lambda_j\perp\text{span}(\mu_1, \cdots,\mu_M)$, $\tilde{\mu}_j\in\text{span}(\mu_1,\cdots,\mu_M)$,  and
> $\|\tilde{\mu}_j\|\geq \Theta((\log\epsilon^{-1})^{-1})$. This means that the TSR patterns do not necessarily be linear combinations of merely TRR patterns, but should contain enough amount of TRR patterns.
>
> The reason why we can handle this case comes from the training dynamics analysis. Based on our formulation of training data and tasks, we can derive strong conclusions about the properties of the trained model in Section 4, where the attention layer learns the TRR pattern and the positional encoding. Then, during CoT inference, the essential is to ensure enough TRR patterns in the TSR patterns, so that the attention layer selects tokens with the same TSR pattern and the same step as the query. Having $\lambda_j$ that is orthogonal to TRR patterns does not affect the CoT mechanism as long as the norm of each $\tilde{\mu}_j$ is not too small.

---

> ### Author Response · Authors · 2024-12-01
>
> Dear Reviewer 6Hy5,
>
> Since the reviewer-author discussion period is closing within less than two days, we would appreciate it if you could check our response and see whether your concern has been addressed. If so, we kindly hope that you could consider updating the evaluation of the paper. We are pleased to answer your further questions. Thank you very much for your time and effort!
>
> Authors

---

### Official Review · Reviewer_txG6 · 2024-11-03

**Soundness:** 3
**Presentation:** 3
**Contribution:** 3
**Rating:** 6
**Confidence:** 3

**Summary:**

The paper provides a theoretical study on training Transformers with nonlinear attention to obtain CoT generalization capability, allowing the model to infer unseen tasks when the input is augmented with examples of the new task. They quantify the required training samples and iterations needed to train a Transformer model towards CoT ability and prove the success of its CoT generalization on unseen tasks with distribution-shifted testing data. The paper's major contributions include a quantitative analysis of training dynamics on a one-layer single-head attention-only Transformer, characterizing the conditions for accurate reasoning output by CoT even in the presence of noisy and inaccurate examples. It also provides a theoretical explanation for why CoT outperforms in-context learning (ICL) in certain cases, highlighting the robustness of CoT against noise in reasoning examples. Through experiments, the authors justify their theoretical findings, demonstrating the practical relevance of their theoretical analysis. This work advances the understanding of Transformer models' reasoning abilities and their potential applications in solving complex reasoning tasks.

**Strengths:**

1. Theoretical Advancement: The paper makes a significant theoretical contribution by providing the first analysis of training dynamics for Transformers aimed at achieving chain-of-thought (CoT) generalization capability. This addresses a critical gap in the literature, as prior works have not fully explored how Transformers can be trained to generalize on multi-step reasoning tasks via CoT.
﻿2. Deep Quantitative Analysis: The authors offer a detailed quantitative analysis of the training process, including the number of context examples required for successful CoT reasoning and the conditions under which CoT outperforms in-context learning (ICL). This level of detail is commendable and provides actionable insights for practitioners looking to implement CoT in Transformers.

**Weaknesses:**

1. **Limitations in Model Complexity**: The paper primarily analyzes a single-head attention Transformer, which may not encapsulate the full complexity and performance characteristics of multi-layer and multi-head attention models. The validation was confined to binary classification tasks, thereby restricting the generalizability of the theoretical findings.
2. **Formula Accuracy**: The paper requires a meticulous review of its mathematical expressions. For instance, in Example 1, the function should be denoted as \( f_2(\mu_2') = \mu_1' \) instead of \( f_1(\mu_2') = \mu_1' \), and the second matrix \( A_1^f \) should be corrected to \( A_2^f \). It is essential to verify all formulas throughout the text to ensure accuracy.
3. **Theorem Validity and Clarification**: The theorems presented in the article should be scrutinized for their validity, particularly the sections that substantiate reasoning, as they may induce some ambiguity. Reading the appendix is mandatory for a comprehensive understanding of the article; otherwise, it might lead to misconceptions.
4. **Originality Concerns**: The article's reasoning and writing logic bear similarities to those found in "How Do Nonlinear Transformers Learn and Generalize in In-Context Learning." It raises the question of whether this work is merely an extension of the previous study or if it introduces novel contributions.

**Questions:**

See the section on weakness. We will increase the score based on the answer to the question.

---

> ### Author Response · Authors · 2024-11-24
> **Response to Reviewer txG6 Part I**
>
> We thank the reviewer for the valuable time and effort in the evaluation. We have made revisions to Example 1 in Section 3.3, Definition 1 and Remark 1 in Section 3.4, Condition 1 and Theorem 3 in Section 3.5, and Proof Sketch in Section 4.2  according to the suggestions.
>
> **Q1 (Weakness 1)**: Limitations in Model Complexity. The validation was confined to binary classification tasks, thereby restricting the generalizability of the theoretical findings.
>
> **A1**: This is a good question. **First, the state-of-the-art theoretical works [Li et al., 2023; Huang et al., 2023 & 2024; Makkuva et al., 2024] recently published at top ML conferences like ICLR, Neurips, and ICML on the theoretical generalization and/or learning dynamics of Transformers also focus on one-layer single-head Transformers only**. That is because the loss landscape for multi-layer/head Transformers is highly nonlinear and non-convex due to the interactions between multiple nonlinear functions. The extension to multi-layer multi-head Transformers requires a more complicated characterization of the gradient updates, while the simplified model architecture can benefit the theoretical analysis as mentioned by the reviewer.
> For the theoretical generalization guarantees of Transformers beyond one-layer single-head, to the best of our knowledge, only [Chen et al., 2024; Yang et al; 2024]  theoretically study ICL with multi-head Transformers, and [Gatmiry et al., 2024] studies ICL with looped Transformers, which are multi-layer Transformers but the weights for all layers are the same. To the best of our knowledge, no existing works study the optimization and generalization of CoT theoretically at all,  even for one-layer single-head Transformers. Therefore, we focus on the one-layer analysis to obtain some initial theoretical insights.
>
> **Second, the simplification of one-layer single-head Transformers enables us to make contributions under our CoT settings**. Our work is the first one to investigate the optimization and generalization of CoT and characterize the conditions when CoT is better than ICL. We establish the required number of context examples for a successful CoT in terms of how informative and erroneous the prompt is.
>
>
> **Third, the theoretical conclusions of one-layer single-head Transformers are empirically verified by multi-layer multi-head Transformers to some extent**. We justify the attention mechanism of CoT on a three-layer two-head Transformer in Figure 7 of the submitted manuscript on page 10. The findings show that at least one head of each layer is implementing the CoT mechanism of Proposition 1 for the single-layer single-head case. We leave the detailed theoretical analysis of the multi-layer multi-head case as future works.
>
>
> We also would like to kindly clarify that **the inference task we theoretically study and empirically verify is a transition task between orthonormal patterns, instead of binary classification**. Specifically, the $k$-step task function $f_k$ maps the input $z_{k-1}$ to $z_k$, $k\in[K]$, where both $z_{k-1}$ and $z_k$ are from a set of orthonormal patterns. This is introduced in lines 254-262 in Section 3.2. We use two patterns in Example 1. However, our theoretical analysis can handle any $K$ patterns, and we use $M=20$ TRR patterns in our experiment.
>
> Li et al., 2023a. A Theoretical Understanding of Shallow Vision Transformers: Learning, Generalization, and Sample Complexity. At ICLR.
>
> Huang et al., 2024. In-context convergence of transformers. At ICML.
>
> Makkuva et al., 2024. Local to Global: Learning Dynamics and Effect of Initialization for Transformers. At Neurips.
>
> Chen et al., 2024. Training dynamics of multi-head softmax attention for in-context learning: Emergence, convergence, and optimality. Preprint.
>
> Yang et al., 2024. In-context learning with representations: Contextual generalization of trained transformers. Neurips 2024.
>
> Gatmiry et al., 2024. Can Looped Transformers Learn to Implement Multi-step Gradient Descent for In-context Learning? At ICML.

---

> > ### Author Response · Authors · 2024-11-24
> > **Response to txG6 Part II**
> >
> > **Q2 (Weakness 2)**: Formula Accuracy: The paper requires a meticulous review of its mathematical expressions.
> >
> > **A2**: We are sorry for the typos and thank the reviewer for catching this. Yes, in Example 1, it should be $f_2(\mu_2') = \mu_1'$ instead of $f_1(\mu_2') = \mu_1'$ at the end of line 308. It should be $A_2^f=\begin{pmatrix} 0.4 & 0.6 \\\\ 0.8 & 0.2 \end{pmatrix}$ instead of $A_1^f=\begin{pmatrix} 0.4 & 0.6 \\\\ 0.8 & 0.2 \end{pmatrix}$. We have reviewed the accuracy of the paper and made corresponding revisions according to the reviewers' suggestions.
> >
> > ----------------
> >
> > **Q3 (Weakness 3)**: Theorem Validity and Clarification: The theorems presented in the article should be scrutinized for their validity, particularly the sections that substantiate reasoning, as they may induce some ambiguity.
> >
> > **A3**: Thank the reviewer for the suggestion. We have improved the presentation of the sections related to reasoning and the proof sketch to support theorems to reduce ambiguity. The revision in the updated version is marked in red. In Definition 1, we restate that $\tau_f$ measures the minimum probability, over all the initial TSR patterns, of the $K$-step  reasoning trajectory that has the highest probability over all $K$-step trajectories. $\tau_o^f$ measures the minimum probability, over all the initial TSR patterns, of the output that has the highest probability over outputs. The proof sketch is also improved in Section 4.2, where we add the illustration of the two stages of training dynamics in lines 450-452. We update several remarks to help illustrate Theorems. In the revision of Remark 1, we emphasize extra conditions for successful CoT. In the remark after Condition 1, we add a more detailed explanation of what Condition 1 states and why Example 1 does not satisfy Condition 1.
> >
> > We would like to restate the main results of our theorems to avoid confusion. Theorem 1 shows that a one-layer Transformer can achieve a diminishing loss of data following the same distribution as training examples with sufficient training samples and iterations. Theorem 2 states that if TSR patterns contain a non-trivial component in the span of TRR patterns, CoT generalization achieves zero error if the number of testing examples is sufficient. If the contexts are more informative of the current task (larger $\alpha’$) and more accurate in the reasoning (larger $\tau^f$, $\rho^f$), the required number of testing examples can be reduced. Theorem 3 states that when Condition 1 holds, i.e., the fraction of correct input-label examples is dominant in the testing prompt, a desired ICL generalization can be obtained by enough testing examples. If Condition 1 does not hold, more examples cannot help the performance, and CoT outperforms ICL.

---

> ### Author Response · Authors · 2024-11-24
> **Response to txG6 Part III**
>
> **Q4 (Weaknes 4)**: Originality Concerns. It raises the question of whether this work is merely an extension of the previous study “How Do Nonlinear Transformers Learn and Generalize in In-Context Learning” or if it introduces novel contributions.
>
> **A4**: This is a good question. **There are many key differences between our work and the mentioned work**. First, the studied tasks in these two works are different. **[Li et al., 2024a] study binary classification using in-context learning, while our work studies multi-step inference as a transition between patterns, which is different from binary classification, using Chain-of-Thought**. We also allow the existence of incorrect steps in the transition for each testing task, while [Li et al., 2024a] only consider the correct label given inputs that determine the task. To the best of our knowledge, no similar formulation has appeared in the CoT works to theoretically study the training dynamics and the generalization performance.
>
> **Second, we analyze a different training dynamics and mechanism of the trained model than [Li et al., 2024a]**. By adding positional encoding to distinguish different reasoning steps in the formulation, we theoretically derive two stages of the training dynamics. In the first stage, the attention layer prioritizes tokens in the same reasoning step as that of  the query. In the second stage, among those tokens in the same reasoning step, the model further selects tokens with the same TSR pattern and the same step as the query.  Such a mechanism was not analyzed in [Li et al., 2024a].
>
> Specifically,  Lemmas 3 and 4 in this paper show that if a training prompt $P$ includes the first $k$ steps of the reasoning query, then the attention weights on tokens of $P$ with a different step from the query decrease to be close to zero in the first stage. Lemma 5 computes the gradient updates in the second stage, where the attention weights on columns in $P$ that correspond to the same step and have the same TRR pattern as the query gradually becomes dominant.
>
> **In contrast, no positional encoding is considered in [Li et al., 2024a]**. The ICL mechanism in [Li et al., 2024a] only selects examples with the same pattern as the query, and there is no phase change in the training dynamics analysis. This implies that the convergence analysis of [Li et al., 2024a] cannot be extended to the CoT setting.
>
> **Third, the analysis in our work makes extra contributions to the comparison between CoT and ICL on the data and tasks formulated in Section 3.2**. Previous works [Feng et al., 2024; Li et al., 2023; Li et al., 2024b] only study this problem by comparing the expressive power of using CoT and ICL, and none of them provides any theoretical comparison of the generalization performance,  to the best of our knowledge.
>
> Li et al., 2024a. How Do Nonlinear Transformers Learn and Generalize in In-Context Learning? At ICML.
>
> Feng et al., 2023. Towards revealing the mystery behind chain of thought: a theoretical perspective. At Neurips.
>
> Li et al., 2023. Dissecting chain-of-thought: Compositionality through in-context filtering and learning. At Neurips.
>
> Li et al.,2024b. Chain of thought empowers transformers to solve inherently serial problems. At ICLR.

---

> ### Author Response · Authors · 2024-12-01
>
> Dear Reviewer txG6,
>
> Since the reviewer-author discussion period is closing within less than two days, we would appreciate it if you could check our response and see whether your concern has been addressed. If so, we kindly hope that you could consider updating the evaluation of the paper. We are pleased to answer your further questions. Thank you very much for your time and effort!
>
> Authors

---

> > ### Author Response · Authors · 2024-12-02
> > **A Friendly Reminder**
> >
> > Dear Reviewer txG6,
> >
> > Since today is the last day for reviewers to ask questions, we would appreciate it if you could check whether your concerns have been addressed by our responses. We are willing to answer your further questions. Thank you!
> >
> > Authors

---

### Official Review · Reviewer_y9x5 · 2024-11-03

**Soundness:** 4
**Presentation:** 3
**Contribution:** 3
**Rating:** 8
**Confidence:** 2

**Summary:**

The paper provides a theoretical characterization of three aspects of in-context learning with examples consisting of chain-of-thought using a simplified data model and a simplified single-layer transformer with non-linear attention. The central result of the paper is a quantitative characterization of the training dynamics that leads to CoT ability. Using this, the authors characterize how the noise and relevance of the in-context examples affect the CoT performance. The final result compares the error rates for ICL with CoT and ICL without CoT with the main conclusion that while the rates are similar, ICL without CoT needs the examples to be dominantly correct, while CoT does not need this. Experiments on synthetic data support the theoretical results.

**Strengths:**

**Originality and Significance**: While previous papers analyze ICL (with and without CoT), their assumptions do not match this paper's. Specifically, this paper does the analysis using a transformer with non-linear attention, with test data coming from a different distribution than the training data, making the setting more general and useful.

**Quality and Clarity**: The theoretical results are presented clearly and are supported by relevant experiments. The discussion and remarks help with understanding the implications of the results.

**Weaknesses:**

It would increase the impact of the paper if the authors could include experiments on some real datasets.

**Questions:**

Are the results applicable, or can they be easily extended to the case where the reasoning steps $z_i, f(z_i)$ are not necessarily of the same sequence length? In other words, $z_1 \in R^{1\times d}$, $f(z_1) \in \mathbb R^{2\times d}$, $z_2 \in \mathbb R^{2 \times d}$, $f(z_2) \in \mathbb R^{1\times d}$. Such a setting might be of greater interest because the steps are typically multi-token in applications involving natural language.

---

> ### Author Response · Authors · 2024-11-24
> **Response to Reviewer y9x5**
>
> We thank the reviewer for the valuable time and effort in the evaluation. We have added Appendix A according to the suggestions.
>
> **Q1 (Weaknesses)**: It would increase the impact of the paper if the authors could include experiments on some real datasets.
>
> **A1**: Thank you for the question. We have added an experiment on GPT-4  in Section A of Appendix. We consider a simple arithmetic task that outputs $((A\_1 o\_1 A\_2)o\_2 A\_3) o\_3 A\_4$ given $A\_1, A\_2, A\_3, A\_4$ chosen from integers from $0$ to $9$ as the input, where $o_1, o_2, o_3\in O=\\{+,-,\times\\}$. The CoT output follows the format of $A_1 o_1 A_2=S_1, S_1 o_2 A_3= S_2, S_2 o_3 A_4=S_3$ and will be evaluated by whether all the three steps are correct for the query as Equation 7. ICL directly outputs $S_3$, and the performance is evaluated by the prediction accuracy of $S_3$ as Equation 9. We use $50$ prompts for testing.
>
> We first consider the error that $o_3$’s are replaced by one operation $\hat{o_3}$ from $O\backslash o_3$ in the presentation of some testing examples. Note that $S_3$’s are still correctly computed by  $S_3=S_2 o_3A_4$. Our results show that when the number of testing examples is fixed, with the number of incorrect examples increasing, the testing accuracy decreases. This is consistent with Remark 1 for Theorem 2. We next consider the setting, where $o_1$’s are replaced with one operation $\hat{o_1}$ randomly and independently selected from $O\backslash o_1$. Hence, $S_1=A_1\hat{o_1}A_2$, and the successive computations are based on the wrongly computed $S_1$. Our results show that when two incorrect examples exist, CoT performs better than ICL, which justifies Remark 2 for Theorem 3.
>
> -----------------
>
> **Q2 (Questions)**: Are the results applicable, or can they be easily extended to the case where the reasoning steps $z_i$, $f(z_i)$ are not necessarily of the same sequence length?
>
> **A2**: This is an interesting question. One potential extension from our current framework is the setup that the number of output tokens increases through every reasoning step. Similarly, another extension is the setup that the number of output tokens decreases through each reasoning step. For example, for any $k\in[K]$, $f_k(z_i)\in\mathbb{R}^{H\times d\_\mathcal{X}}$, $H>1$, given $z_i\in\mathbb{R}^{d\_\mathcal{X}}$ as the input. Such an output in each step can be achieved by using a one-layer $H$-head Transformer, which can be a concatenation of $H$ heads of the Transformer model in our Equation 2.  Such a case can be handled by our setup with simple modification to be $H$ heads. We can formulate the prompt as Equation 1 with each $x_i$ and $y_{i,k}$ as $KHd\_\mathcal{X}$ dimensional for $k\in[K]$. The first token of $x_i$ is a TRR/TSR pattern, while the remaining tokens are zero. $y_{i,k}$ has the first $kH$ tokens of TRR/TSR patterns with zero paddings as the remaining tokens. Then, the learned model can implement the CoT mechanism characterzied in Section 4.1, i.e., Transformers attend to tokens with the same step and the same patterns as the query. We leave the detailed analysis of one layer $H$ heads as a future work.

---

> > ### Comment · Reviewer_y9x5 · 2024-11-26
> > **Thanks for the clarifications**
> >
> > I have no further questions.

---

> > > ### Author Response · Authors · 2024-11-26
> > > **Thank you**
> > >
> > > Thank you again for your time and effort in the evaluation and the positive score.

---

### Official Review · Reviewer_LGXd · 2024-11-04

**Soundness:** 2
**Presentation:** 3
**Contribution:** 3
**Rating:** 6
**Confidence:** 3

**Summary:**

This work presented a theoretical study on training Transformers to achieve Chain-of-Thought (CoT) reasoning capabilities. The authors explored the training dynamics of nonlinear Transformers and provided a quantitative analysis of the required training samples and iterations for developing CoT ability. They analyzed the model's success in generalizing CoT on new tasks with O.O.D. test data, even in noisy and inaccurate reasoning examples, furthermore, contrasted CoT with in-context learning (ICL) and theoretically characterizes scenarios where CoT outperforms ICL.

**Strengths:**

1. The paper provided a novel theoretical framework for understanding how a simplified Transformer can be trained to perform CoT reasoning, which is important for Transformers-based model reasoning.

2. The authors offered a detailed quantitative analysis of the training process, including the number of context examples, training samples, and iterations required for successful CoT training. This level of detail helps in understanding the complexity and feasibility of training similar models.

3. The paper addresses the generalization of CoT ability to new tasks with distribution shifts, which is a practical concern for real-world applications where training and testing data may differ.

**Weaknesses:**

1. The paper idealized the Transformer as a simplified one-layer single-head architecture. While such a simplification can aid in theoretical analysis, it may limit the generalizability of more complex Transformer architectures.

2. The assumptions made about the data and tasks, such as the orthonormality of training-relevant and testing-relevant patterns, may not hold in all real-world scenarios, which could affect the applicability of the results.

**Questions:**

1. For the CoT training set, this work constructed it with examples + query + (K-1)-reasoning steps, and the K-th step as the label. Why did not adopted more later steps as the label or more flexible label strategies?

2. In the 3rd line of Example 1, should the last affinity function be ?

3. How well do the theoretical results extend to more complex Transformer models with multiple layers and heads?

4. Have the authors considered validating their theoretical findings with real-world datasets, and if so, what are the challenges and potential modifications needed?

5. Could the authors elaborate on how the model's performance is affected by different levels of noise and inaccuracy in the context examples?

---

> ### Author Response · Authors · 2024-11-24
> **Response to Reviewer LGXd Part I**
>
> We thank the reviewer for the valuable time and effort in the evaluation. We have made revisions to Example 1 and added Appendix A according to the suggestions.
>
> **Q1 (Weakness 1 & Question 3)**: The paper idealized the Transformer as a simplified one-layer single-head architecture. While such a simplification can aid in theoretical analysis, it may limit the generalizability of more complex Transformer architectures. How well do the theoretical results extend to more complex Transformer models with multiple layers and heads?
>
>
> **A1**: This is a good question.
>
> **First, the state-of-the-art theoretical works [Li et al., 2023; Huang et al., 2023 & 2024; Makkuva et al., 2024] recently published at top ML conferences like ICLR, ICML, and Neurips on the theoretical generalization and/or learning dynamics of Transformers also focus on one-layer single-head Transformers only.** That is because the loss landscape for multi-layer/head Transformers is highly nonlinear and non-convex due to the interactions between multiple nonlinear functions. The extension to multi-layer multi-head Transformers requires a more complicated characterization of the gradient updates, while the simplified model architecture can benefit the theoretical analysis as mentioned by the reviewer.
>
> For the theoretical generalization guarantees of Transformers beyond one-layer single-head, to the best of our knowledge, only [Chen et al., 2024; Yang et al., 2024]  theoretically study ICL with multi-head Transformers, and [Gatmiry et al., 2024] studies ICL with looped Transformers, which are multi-layer Transformers but the weights for all layers are the same. To the best of our knowledge, no existing works study the optimization and generalization of CoT theoretically at all,  even for one-layer single-head Transformers. Therefore, we focus on the one-layer analysis to obtain some initial theoretical insights.
>
>
> **Second, the simplification of one-layer single-head Transformers enables us to make contributions under our CoT settings.** Our work is the first one to investigate the optimization and generalization of CoT and characterize the conditions when CoT is better than ICL. We establish the required number of context examples for a successful CoT in terms of how informative and erroneous the prompt is.
>
>
> **Third, the theoretical conclusions of one-layer single-head Transformers are empirically verified by multi-layer multi-head Transformers to some extent.** We justify the attention mechanism of CoT on a three-layer two-head Transformer in Figure 7 of the submitted manuscript on page 10. The findings show that at least one head of each layer is implementing the CoT mechanism of Proposition 1 for the single-layer single-head case. This implies that our training dynamics analysis of the individual head of matching tokens with the same TSR pattern and the same step as the query can potentially be extended to multi-layer multi-head case. We leave the detailed theoretical analysis of the extension as future works.
>
> Li et al., 2023. A Theoretical Understanding of Shallow Vision Transformers: Learning, Generalization, and Sample Complexity. At ICLR.
>
> Huang et al., 2024. In-context convergence of transformers. At ICML.
>
> Makkuva et al., 2024. Local to Global: Learning Dynamics and Effect of Initialization for Transformers. At Neurips.
>
> Chen et al., 2024. Training dynamics of multi-head softmax attention for in-context learning: Emergence, convergence, and optimality. Preprint.
>
> Yang et al., 2024. In-context learning with representations: Contextual generalization of trained transformers. Neurips 2024.
>
> Gatmiry et al., 2024. Can Looped Transformers Learn to Implement Multi-step Gradient Descent for In-context Learning? At ICML.

---

> ### Author Response · Authors · 2024-11-24
> **Response to Reviewer LGXd Part II**
>
> **Q2 (Weakness 2)**: The assumptions made about the data and tasks, such as the orthonormality of training-relevant and testing-relevant patterns, may not hold in all real-world scenarios, which could affect the applicability of the results.
>
> **A2**: We agree that real-world data can be more complicated, and there is a gap between theory and experiments. There are several reasons for using such data formulation. **First, our data formulation of orthogonal patterns is widely used in the state-of-the-art theoretical study of model training or ICL on language and sequential data [Huang et al., 2024; Tian et al., 2023; Chen et al., 2024] published at top ML conferences ICML, ICLR, Neurips**. For example, Sections 2.1 and 2.2 in [Chen et al., 2024] consider learning n-gram data in ICL by formulating transition between orthogonal patterns. [Huang et al., 2024; Li et al., 2024] study ICL on regression or classification tasks, which also use orthogonal patterns as data. Section 3.1 of [Li et al., 2023] and Section 3 of [Tian et al., 2023] also assume orthogonal patterns in Transformer model training. The data formulation we use is consistent with the existing theoretical works.
>
> **Second, one can characterize the gradient updates in different directions for different patterns and steps**. This facilitates us to distinguish the impact of different patterns and steps in the convergence analysis of CoT using Transformers. Moreover, we would like to mention that during the inference, the tokens in testing prompts contain noises as defined in Equation 10. This makes the tokens of different TSR patterns not orthogonal to each other and relaxes our orthogonality condition to some degree.
> Therefore, we believe our formulation could be an appropriate framework for potential theoretical research on LLM reasoning in theory.
>
>
> Huang et al., 2024. In-context convergence of transformers. At ICML.
>
> Chen et al., 2024. Unveiling Induction Heads: Provable Training Dynamics and Feature Learning in Transformers. Preprint.
>
> Tian et al., 2023. Scan and Snap: Understanding Training Dynamics and Token Composition in 1-layer Transformer. At Neurips.
>
> Li et al., 2024. How Do Nonlinear Transformers Learn and Generalize in In-Context Learning? At ICML.
>
> Li et al., 2023. A Theoretical Understanding of Shallow Vision Transformers: Learning, Generalization, and Sample Complexity. At ICLR.
>
> -----------------
>
> **Q3 (Question 1)**: For the CoT training set, this work constructed it with examples + query + (K-1)-reasoning steps, and the K-th step as the label. Why did not adopted more later steps as the label or more flexible label strategies?
>
> **A3**: This is a great question. For CoT training, we use the first $k-1$ steps in the reasoning query, $k\in[K]$, where $K$ is the total number of steps for the task, and the $k$-th step output is the label for the prompt. This means the training data include using the first $k-1$ steps to predict the $k$th step for all possible choices of $k\in[K]$. One motivation for this setting of training is to simulate the Next Token Prediction training process in practice. In our CoT training, given the input prompt that may end at any possible reasoning step, we aim to predict the next token. This is close to the setting of Next Token Prediction [Nichani et al., 2024; Ildiz et al., 2024], where we use one word in the text as the label and the prefix of this word as the input sequence.  Moreover, using all the reasoning steps of the query as training labels is a similar setting to ours that satisfies the requirements for our CoT training. This is because training with all the steps as labels can ensure that the query input is uniformly sampled from all the steps as required by lines 168-169 in Section 2.2. Both training settings enable the model to fully learn all the positional encodings to distinguish different reasoning steps.
>
> Nichani et al., 2024. How Transformers Learn Causal Structure with Gradient Descent. At ICML.
>
> Ildiz et al., 2024. From Self-Attention to Markov Models: Unveiling the Dynamics of Generative Transformers. At ICML.
>
> -----------------
>
> **Q4 (Question 2)**: In the 3rd line of Example 1, should the last affinity function be?
>
> **A4**: We think the reviewer refers to, the last function should be $f_2(\mu_2’)=\mu_1’$ in the 3rd line of Example 1. Thank you for catching this typo. We have fixed it in the updated version.

---

> ### Author Response · Authors · 2024-11-24
> **Response to Reviewer LGXd Part III**
>
> **Q5 (Question 3)**: Have the authors considered validating their theoretical findings with real-world datasets, and if so, what are the challenges and potential modifications needed?
>
> **A5**: Thank you for the question. **We have added an experiment on GPT-4 in Section A of Appendix**. We consider a simple arithmetic task that outputs $((A\_1 o\_1 A\_2)o\_2 A\_3) o\_3 A\_4$ given $A\_1, A\_2, A\_3, A\_4$ chosen from integers from $0$ to $9$ as the input, where $o_1, o_2, o_3\in O=\\{+,-,\times\\}$. The CoT output follows the format of $A_1 o_1 A_2=S_1, S_1 o_2 A_3= S_2, S_2 o_3 A_4=S_3$ and will be evaluated by whether all the three steps are correct for the query as Equation 7. ICL directly outputs $S_3$, and the performance is evaluated by the prediction accuracy of $S_3$ as Equation 9. We use $50$ prompts for testing.
>
> We first consider the error that $o_3$’s are replaced by one operation $\hat{o_3}$ from $O\backslash o_3$ in the presentation of some testing examples. Note that $S_3$’s are still correctly computed by  $S_3=S_2 o_3A_4$. Our results show that when the number of testing examples is fixed, with the number of incorrect examples increasing, the testing accuracy decreases. This is consistent with Remark 1 for Theorem 2. We next consider the setting, where $o_1$’s are replaced with one operation $\hat{o_1}$ randomly and independently selected from $O\backslash o_1$. Hence, $S_1=A_1\hat{o_1}A_2$, and the successive computations are based on the wrongly computed $S_1$. Our results show that when two incorrect examples exist, CoT performs better than ICL, which justifies Remark 2 for Theorem 3.
>
> --------------------
>
> **Q6 (Question 4)**: Could the authors elaborate on how the model's performance is affected by different levels of noise and inaccuracy in the context examples?
>
> **A5**: Our theoretical results in Theorems 2 and 3 characterize sufficient conditions for successful CoT and ICL generalization. These conditions are based on the setting that, **first, the magnitude of the noise is smaller than $\sqrt{2}/2$ in Equation 10**. This is to ensure that the noise level is not too large to confuse two distinct TSR patterns, which guarantees that the mechanism of selecting tokens with the same TSR pattern as the query is not affected by the noise. **Second, the inaccuracy in testing examples is upper bounded so that Equation 11 holds, i.e., the true label of each step is still the most probable by the step-wise transition matrices**. This guarantees that the testing prompt contains the dominant fraction of correct examples generated by the transition matrices. Therefore, the testing examples are overall informative of the testing task.
> If conditions in Equations 10 or 11 do not hold, our theoretical analysis cannot ensure a successful CoT or ICL on the multi-step inference task we formulate in Section 3.2.

---

> > ### Comment · Reviewer_LGXd · 2024-11-26
> >
> > Thanks for the responses. My concerns have been addressed, and I have increased my score.

---

> > > ### Author Response · Authors · 2024-11-26
> > > **Thank you!**
> > >
> > > We are delighted that your concerns have been addressed. Thank you for increasing the score to positive!

---

### Author Response · Authors · 2024-11-26
**A Friendly Reminder**

Dear Reviewers txG6 and 6Hy5,

Since the deadline for resubmitting the updated manuscript is approaching within less than two days, we would appreciate it if you could check our response and see whether your concern has been addressed. If so, we kindly hope that you could consider updating the evaluation of the paper. We are pleased to answer your further questions and update our submission. Thank you very much for your time and effort!

Authors

---

### Author Response · Authors · 2024-12-04
**Summary of Revisions**

Dear Reviewers/AC/SAC/PC,

We appreciate the evaluation and suggestions. Here is a summary of revisions in the uploaded manuscript according to the review.

1. We have corrected typos in Example 1 and other places as suggested by Reviewers LGXd and txG6.

2. We have added Appendix A as a real-world experiment to support our analysis as suggested by Reviewers LGXd and y9x5.

3. We have revised Definition 1 and Remark 1 in Section 3.4, Condition 1 and Theorem 3 in Section 3.5, and Proof Sketch in Section 4.2 for a better readability of the discussion of CoT reasoning as suggested by Reviewer txG6.

4. We have updated Theorem 2, Remark 1, and related proof to extend our analysis to a wider out-of-domain CoT generalization as suggested by Reviewer 6Hy5.

Thanks,

Authors

---

### Meta-Review · Area_Chair_4FSQ · 2024-12-22

**Metareview:**

Summary: The paper investigates how nonlinear Transformers can be trained to perform Chain-of-Thought (CoT) reasoning, providing a theoretical foundation for this capability. It examines the training dynamics required to develop CoT abilities and demonstrates that Transformers trained in this way can generalize robustly to unseen tasks. Additionally, a comparison with In-Context Learning (ICL) highlights CoT's advantages, particularly in scenarios where ICL struggles due to its dependency on accurate input-label pairs. Simple experiments are used to corroborate theoretical results.

Strengths:
- All the reviewers leaning positive
- Analysis of training dynamics for CoT capability in transformers - an important problem
- Detailed quantitative analysis of sample complexity, iterations, and generalization bounds
- Theoretical framework comparing CoT vs ICL performance

Weakness:
- Analysis limited to simplified one-layer single-head transformer architecture
- Assumptions about orthogonal patterns may not fully represent real-world scenarios
- Experiments primarily on synthetic data rather than real-world applications
- Some presentation issues including typos and formula accuracy that needed correction

Decision:
All the reviewers were in consensus that the paper represents an important step forward in understanding transformer capabilities and limitations, providing both theoretical insights and practical implications for future work. It merits publication at ICLR 2025, with a few minor edits: the contribution section can softened and simplified: currently repitive and over-claiming.

**Additional Comments On Reviewer Discussion:**

We thank the authors and reviewers for engaging during the discussion phase towards improving the paper. Below are some of the highlights:

1. Model Complexity Limitations:
- Reviewers questioned applicability to real transformers
- Authors justified focus on simplified model by citing precedent in theoretical works
- Added experiments on GPT-4 for arithmetic tasks
- Demonstrated empirical verification on multi-layer transformers

2. Data Formulation:
- Concerns about orthogonal pattern assumptions
- Authors expanded analysis to more general case where test patterns can have out-of-span components
- Updated theorems and proofs accordingly

3. Formula Accuracy:
- Several typos identified in mathematical expressions
- Authors corrected all issues in revised version
- Improved clarity of definitions and theorem statements

4. Originality vs Prior Work:
- Questions about overlap with previous ICL work
- Authors clearly differentiated their novel contributions
- Explained key technical differences in analysis approach

---

### Decision · Program_Chairs · 2025-01-22

Accept (Poster)